# Adaptations for stealth in the wing-like flippers of a large ichthyosaur

Johan Lindgren[1 ✉], Dean R. Lomax[2,3], Robert-Zoltán Szász[4], Miguel Marx[1], Johan Revstedt[4], Georg Göltz[16], Sven Sachs[5], Randolph G. De La Garza[1], Miriam Heingård[1], Martin Jarenmark[1], Kristina Ydström[6], Peter Sjövall[7], Frank Osbæck[8], Stephen A. Hall[9], Michiel Op de Beeck[10], Mats E. Eriksson[1], Carl Alwmark[1], Federica Marone[11], Alexander Liptak[12], Robert Atwood[12], Genoveva Burca[12,13], Per Uvdal[14], Per Persson[10] & Dan-Eric Nilsson[15]

With their superficially shark-like appearance, the Mesozoic ichthyosaurs provide a classic illustration of major morphological adaptations in an ancestrally terrestrial tetrapod lineage following the invasion of marine habitats[1–3]. Much of what is known about ichthyosaur soft tissues derives from specimens with body outlines[4–6]. However, despite offering insights into aspects of biology that are otherwise difficult to envisage from skeletal evidence alone (such as the presence of a crescentic fluke), information on their soft parts has hitherto been limited to a taxonomically narrow sample of small- to dolphin-sized animals[2,4–6]. Here we report the discovery of a metre-long front flipper of the large-bodied Jurassic ichthyosaur *Temnodontosaurus*, including unique details of its soft-tissue anatomy. In addition to revealing a wing-like planform, the fossil preserves a serrated trailing edge that is reinforced by novel cartilaginous integumental elements, herein denominated chondroderms. We also document chordwise-parallel skin ornamentations and a protracted fleshy distal tip that presumably acted like a flexible winglet in life. By integrating morphological and numerical data, we show that the observed features probably provided hydroacoustic benefits, and conclude that the visually guided[7,8] *Temnodontosaurus* relied on stealth while hunting in dim-lit pelagic environments. This unexpected combination of control surface modifications represents a previously unrecognized mode of concealment, and underscores the importance of soft-tissue fossils when inferring aspects of palaeoethology and predator–prey palaeoecology.

Of the multiple lineages of secondarily adapted aquatic tetrapods that have colonized the oceans over the past 300 million years, the Mesozoic ichthyosaurs (Ichthyopterygia) rank among the most successful[3]. During their evolutionary transition from land to sea, these iconic reptiles profoundly transformed their bodies as an adaptive response to an increasingly pelagic existence, gradually attaining more streamlined profiles in the process[1,2]. Discoveries of articulated skeletons with associated body outlines have offered key insights into aspects of the biology, physiology and ecology of both early branching and derived (parvipelvian) forms[6]. However, whereas a wealth of information has been gathered on a range of soft tissues in some smaller-bodied, piscivorous and teuthophagous species[6], no comparable evidence exists for those ichthyosaurs that occupied higher trophic levels in the marine ecosystems.

Here we describe an excellently preserved, approximately 183- to 181-million-year-old (Early Jurassic Epoch) front flipper (forefin) that includes extensive portions of integument (SSN8DOR11; Paläontologisches Museum Nierstein, Nierstein, Germany) of the megapredator *Temnodontosaurus*[9–11]. The notably wing-like fin sheds light on the unique hunting strategy of this large-bodied parvipelvian, revealing secondary control structures that probably served to minimize self-generated noise during foraging activities in low-light habitats—in effect, a novel form of stealth (silent swimming) in an ancient marine reptile.

## Description

SSN8DOR11 was collected from a temporary exposure of dark, laminated limestone belonging to the εII$_5$ ('Unterer Stein') part of the Toarcian Posidonia Shale[12] in south-western Germany (Supplementary Information). The fossil was discovered during construction blasting work, and was consequently retrieved as a three-dimensional jigsaw

[1]Department of Geology, Lund University, Lund, Sweden. [2]Palaeobiology Research Group, School of Earth Sciences, University of Bristol, Bristol, UK. [3]Department of Earth and Environmental Sciences, The University of Manchester, Manchester, UK. [4]Department of Energy Sciences, Lund University, Lund, Sweden. [5]Abteilung Geowissenschaften, Naturkunde-Museum Bielefeld, Bielefeld, Germany. [6]Department of Medical Radiation Physics, Lund University, Lund, Sweden. [7]Materials and Production, RISE Research Institutes of Sweden, Borås, Sweden. [8]Fur Museum, Museum Salling, Fur, Denmark. [9]Division of Solid Mechanics, Lund University, Lund, Sweden. [10]Centre for Environmental and Climate Science, Lund University, Lund, Sweden. [11]Swiss Light Source, Paul Scherrer Institut, Villigen, Switzerland. [12]Diamond Light Source, Harwell Science and Innovation Campus, Didcot, UK. [13]ISIS Neutron and Muon Source, STFC Rutherford Appleton Laboratory, Didcot, UK. [14]Chemical Physics, Department of Chemistry, Lund University, Lund, Sweden. [15]Department of Biology, Lund University, Lund, Sweden. [16]Unaffiliated: Georg Göltz, Schorndorf, Germany. ✉e-mail: johan.lindgren@geol.lu.se

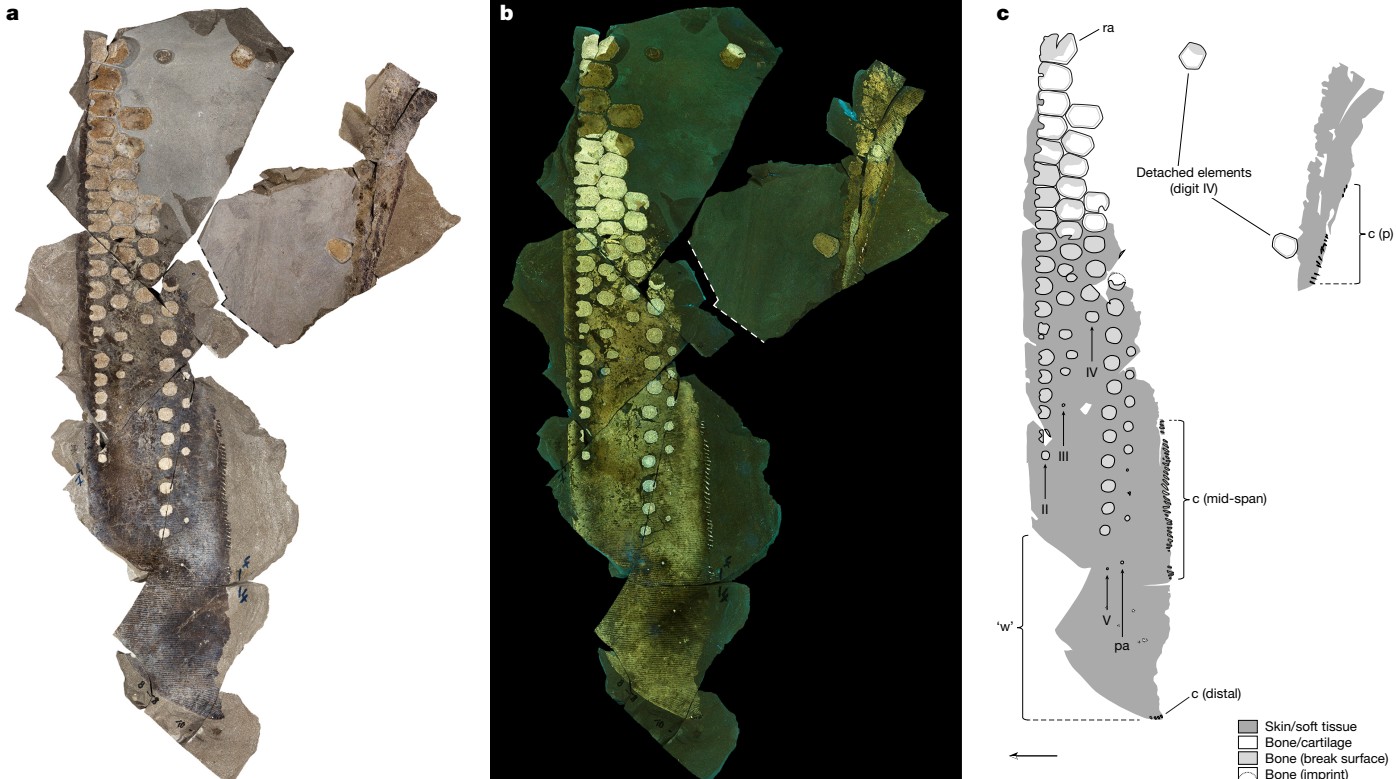

**Fig. 1 | Front flipper of *Temnodontosaurus trigonodon* with soft tissues.**
**a**–**c**, Photographs of the part section of SSN8DOR11 under polarized (**a**) and ultraviolet (longpass cut-off 455 nm) (**b**) light, respectively, together with a diagrammatic representation of the forelimb in planform view (**c**). Note that the individual blocks have been re-assembled in their original position (the stippled line delineates the end of sediment that has been digitally removed to show underlying bones). Arrow indicates anterior. Extended Data Figs. 1 and 2 depict the counterpart section. II–V, digit II–V; c, chondroderms; p, proximal; pa, postaxial accessory digit; ra, radiale; 'w', winglet-like distal segment. Scale bar, 10 cm.

puzzle of odd-sized rock slabs, collectively forming opposing part (Fig. 1) and counterpart (displayed on two main blocks; Extended Data Figs. 1 and 2) sections of a flipper that has been cleaved along the sagittal plane.

The metre-long forelimb is tetradactyl (digits II–V)[13] and equipped with a postaxial accessory digit (Fig. 1, Extended Data Fig. 1 and Supplementary Information). Whereas the radiale, distal carpal 2, metacarpal II, and all phalanges occur in near-perfect articulation, the humerus, epipodials (except for a dislocated fragment of the presumed radius)—as well as the remaining carpals and metacarpals—are missing. In addition, two proximal elements of digit IV, along with a patch of residual soft tissue, have shifted from their original position to lie scattered a short distance from the main section of the fossil (Fig. 1 and Extended Data Figs. 1 and 2). On the basis of its general proportions, digit arrangement and consistent anterior notching of the leading edge elements, SSN8DOR11 can be confidently identified as a forefin of the large-sized temnodontosaurid *Temnodontosaurus trigonodon* (Extended Data Fig. 3 and Supplementary Information).

Surrounding and adhering to the outside of the bones are remnant soft tissues (Fig. 2 and Extended Data Fig. 4). These are preserved as a bedding-parallel coating of dark matter that defines in great fidelity the high aspect ratio planform of the flipper (Fig. 1 and Extended Data Fig. 1). With the possible exception of a blackish film on a few phalanges (Extended Data Fig. 4a–c), the fossilized material appears to derive exclusively from one side of the limb, which is presumed to be the surface that originally was in contact with the seafloor[4]. The compressed matter shows that the soft parts extended well beyond the skeleton distally to produce an extended fleshy fin tip (Fig. 2a). As in other marine tetrapods with hydrofoil-shaped extremities, the bony

support is concentrated towards the leading edge, thereby probably contributing to a broadly teardrop-shaped cross-sectional profile in life. Notably, the digits do not converge distally as otherwise typically seen in historical, slab-mounted skeletal specimens of *T. trigonodon* (Extended Data Fig. 3a); instead, they remain almost parallel to one another over their full length. The leading edge of the flipper is gently curved proximally, but gradually becomes more posteriorly inclined when approaching the tip (Figs. 1 and 2a and Extended Data Fig. 1). Although rather smooth near the base, the trailing edge is modified into prominent sinusoidal serrations mid-distally along the span (Fig. 2b). No scales are apparent; instead, evenly spaced stripes, set approximately 2 mm apart, run chordwise across the entire surface of the fin blade (Fig. 2a–c and Extended Data Fig. 4d,e). These bands do not intersect at any point, although some of them can be seen to bifurcate (Fig. 2a).

Histological and microscopic examination of the mineralized residue revealed a vertically laminated organization that compares favourably with the stratified integument of living amniotes (Extended Data Fig. 4d). The topographically outermost layer (corresponding to the animal) is semi-transparent to pale yellow in colour with interspersed dark brown dots that occasionally exhibit external projections (Fig. 2c,d). When visualized under field emission gun scanning electron microscopy (FEG-SEM) (Extended Data Fig. 4f), these spots were resolved as clusters of carbonaceous microbodies in an otherwise predominantly phosphatized matrix that most probably represents part of the epidermis[5]. An electron-dense interior (Fig. 2e) and intimate association with remnant eumelanin (Extended Data Fig. 5a–c) allow identification of the aggregated granules and their cell-like casings as fossilized melanosomes and melanophores, respectively[5].

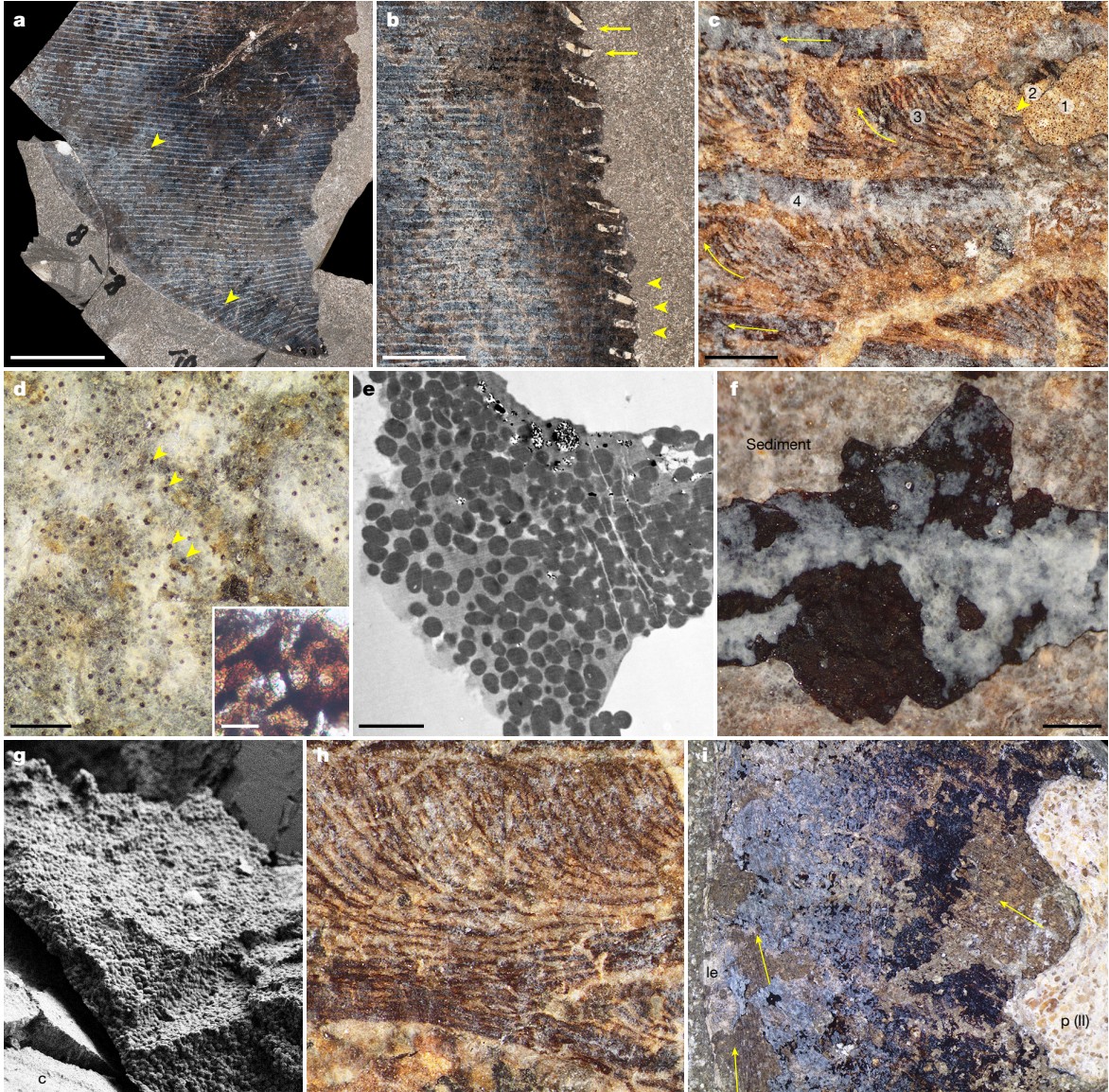

**Fig. 2 | Structure of soft tissues of *Temnodontosaurus* specimen SSN8DOR11.**
**a**, Distal tip. Note chordwise-parallel skin ornamentations. Arrowheads indicate bifurcating lines. **b**, Mid-distal portion of the trailing edge. Originally, each serration was strengthened by a chondroderm (arrows); however, some elements have been taphonomically displaced relative to the soft-tissue margin (arrowheads). **c**, Internal view of soft tissues. Topographically, the preserved matter includes (from exterior to interior): (1) phosphatized epidermis with melanophores; (2) densely packed melanosomes (arrowhead); (3) structural fibre bundles; and (4) a phosphatic crust. Arrows denote main fibre bundle directions. **d**, Pigment cells (arrowheads) embedded in phosphatized epidermis. Inset, melanophores with dendritic processes after demineralization (*n* = 3 samples). **e**, Transmission electron microscopy (TEM) micrograph of a melanophore containing melanosomes (*n* = 2 samples). **f**, Aggregated melanosomes (black layer) representing the taphonomically condensed epidermal–dermal interface and superficial dermis (EDISD). The phosphatic crust marks the bottom of the preserved sequence. **g**, FEG-SEM micrograph depicting EDISD melanosomes enveloping a chondroderm (c) (*n* = 5 specimens). **h**, Structural fibre bundle architecture. **i**, Leading edge (le) of SSN8DOR11. Arrows indicate principal fibre bundle directions. p (II), phalanx of digit II. Scale bars, 5 cm (**a**), 2 cm (**b**), 3 mm (**i**), 1 mm (**c**), 500 μm (**h**), 200 μm (**d**,**f**), 20 μm (**d**, inset), 10 μm (**g**), 2 μm (**e**).

Subjacent to the relict pigment cells is a layer comprising densely packed melanosomes that can be reasonably interpreted as the condensed remains of the juxtaposed epidermal–dermal interface and superficial dermis following extensive decay and subsequent compactional flattening[5] (Fig. 2c,f,g and Extended Data Figs. 4d,e and 5d–g). Despite the current, taphonomically induced low relief, recurring spatial differences in the development of this blackish material contribute to both a patchy 'dashed line' pattern (Extended Data Fig. 4g) and the overall striped appearance of the flipper blade (Fig. 2c and Extended Data Fig. 4d,e). The linear configuration is further augmented by strand-like microstructures organized into a regular meshwork that is likely to represent the fossilized remnants of structural fibre bundles from the dermis, hypodermis and/or underlying connective tissue (Fig. 2c,h and Extended Data Fig. 4d,e). Chordwise-parallel filaments traverse the fin in straight paths beneath the darker bands (Fig. 2c,h), and alternate with bundles that initially run in a near-parallel fashion; however, from this narrow base, they then bend, diverge and intermittently coalesce to form branching, wavy tufts (Fig. 2c,h). Additional fibrous elements, together with a thin phosphatic crust (Fig. 2f and Extended Data Fig. 4d), constitute the innermost portion of the preserved soft-tissue sequence, and include longitudinally oriented strands adjacent to the leading and trailing edges (Fig. 2i and

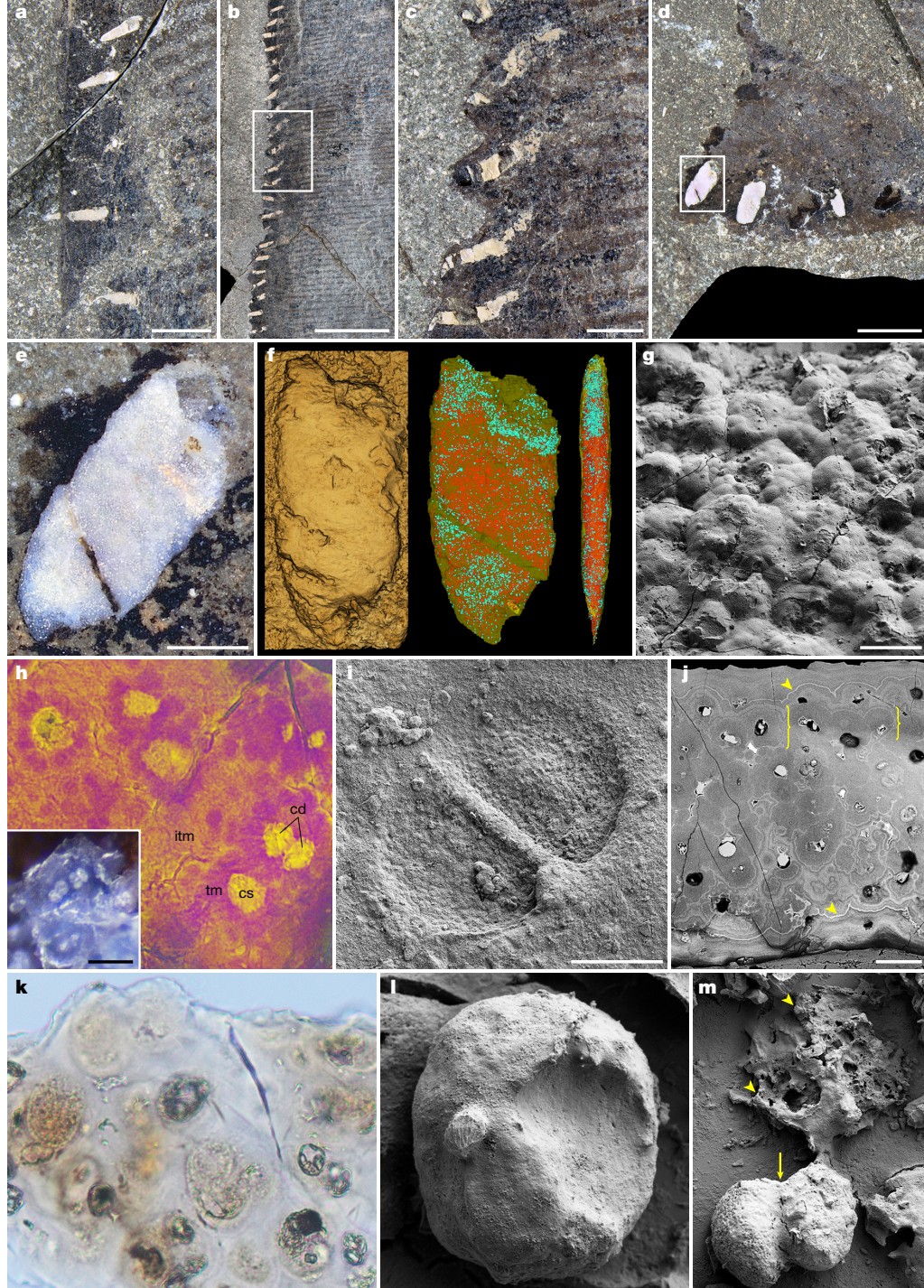

**Fig. 3 | Cartilaginous integumentary deposits. a**, Chondroderms near the base of the trailing edge. **b**, Chondroderms supporting the mid-distal portion of the trailing edge. The region in the box is enlarged in **c. c**, Chondroderms reinforcing the sinusoidal trailing edge. **d**, Chondroderms at the tip of the flipper. The framed region is shown in **e. e**, Magnification of a distal chondroderm. **f**, Micro-computed tomography (μCT) visualization of the same element. 3D rendering of the exposed surface (left) and semi-transparent 3D reconstruction of the chondroderm in planform (centre) and anterior (right) views (see also Supplementary Video 1). Chondrocyte lacunae occur as turquoise and red dots (*n* = 2 specimens). **g**, FEG-SEM micrograph of the granular chondroderm exterior (*n* = 5 specimens). **h**, Synchrotron radiation X-ray tomographic microscopy (SRXTM) rendering of chondroderm cartilage (see also Supplementary Video 2). Artificial colouring reflects density variances (yellow, higher density; magenta, lower density) related to recurring differences in the degree of phosphatization between the territorial and interterritorial matrix, indicating both biological and diagenetic mineralization[49] (*n* = 4 samples). Inset, isogenous groups of cell-like structures embedded in phosphatic matrix. cd, cell doublet; cs, cellular structure; itm, interterritorial matrix; tm, territorial matrix. **i**, Mouldic preservation of a cell doublet (FEG-SEM micrograph) (*n* = 5 samples). **j**, Ground section of chondroderm calcified cartilage (FEG-SEM micrograph) (*n* = 3 samples). Note globular organization (brackets) and Liesegang banding patterns (arrowheads). **k**, Chondrocyte lacunae containing pigmented matter consistent in appearance with cartilage cells. **l**, FEG-SEM micrograph of a carbonaceous chondrocyte after demineralization (*n* = 2 samples). **m**, Cell doublet with mitotic junction (arrow) and associated extracellular matrix (arrowheads) liberated from demineralized chondroderm cartilage (FEG-SEM micrograph) (*n* = 2 samples). Scale bars, 3 cm (**b**), 5 mm (**a,c,d**), 1 mm (**e**), 50 μm (**g,h,j,k**), 10 μm (**i,m**), 5 μm (**l**).

Extended Data Fig. 4i), and antero-proximally directed bundles that appear to insert or originate from the notched anterior margin of the bones in digit II (Fig. 2i and Extended Data Fig. 4k).

## Cartilaginous integumentary structures

Supporting the trailing edge are conspicuous rod-like mineralizations (Fig. 3 and Extended Data Fig. 6a–c). These reside within the melanosome layer that is inferred to represent the condensed epidermal–dermal interface and superficial dermis, and were presumably chordwise oriented in life. The deposits vary in size, shape and geometry along the fin span. Near the base, they are slender, spicular and measure approximately 4–6 mm in length (Fig. 3a). Further distally, they become noticeably larger (ranging in size from about 9 to 13 mm) and attain a pointed, elongate conical shape (the distorted appearance of some elements is likely to reflect preservational artefacts; Fig. 3b,c). At the distal tip, the objects occur in the form of ellipsoid plates, measuring between 2 and 4 mm in length (Fig. 3d–f and Supplementary Video 1).

The morphology and anatomical localization of the mineralizations are reminiscent of osteoderms—integumental skeletal structures that are widely distributed among both extant and extinct tetrapods[14]. However, contrary to these intradermal ossifications, which are composed principally of bone[14], the enigmatic skin features in SSN8DOR11 are made of a material that is morphologically consistent with globular calcified cartilage—a tissue that is typically associated with the endoskeleton of ancestral vertebrates and chondrichthyan fishes[15,16], but also occurs in tetrapods[17].

At the microscopic level, each element is resolved as densely aggregated granular bodies containing centrally located chambers (lacunae) and lunate to spherical cellular structures that are frequently aligned in pairs or small clusters (Fig. 3g–i). The size, shape and organization of the cell-like microstructures conform to extant chondrocytes (cartilage cells), to suggest a common origin. Furthermore, each 'cell nest' (isogenous group) is surrounded by a phosphatized territorial matrix[18] in the shape of a spheroid (Supplementary Video 2). The individual globules are either loosely attached by their walls (Fig. 3h, inset) or held together by a similarly mineralized intervening matrix (Fig. 3h). Concentric growth rings and contour lines (Liesegang banding patterns) were observed in petrographic ground sections prepared from three elements (Fig. 3j), and are likely to represent waves of successive accretional calcification[15,19]. Demineralization of the fossilized tissue liberated cellular bodies with a heterogeneous chemical composition (Fig. 3k–m, Extended Data Fig. 6e–h and Supplementary Information), together with residues of the extracellular matrix that are preserved as stringy fibrous to vesicular organic matter enriched in aliphatic and aromatic hydrocarbons (Fig. 3m and Extended Data Fig. 6d–h). Because osteoderm development does not involve any cartilaginous precursor[14], the trailing edge elements in SSN8DOR11 most probably represent a unique derivative of the tetrapod dermal skeletal system (Supplementary Information). We therefore propose to use the term 'chondroderm'—from the Greek words χόνδρος (*chondros*, meaning cartilage) and δέρμα (*dérma*, meaning skin)—as the name for these novel integumentary reinforcements.

## Functional implications

SSN8DOR11 exhibits a fourfold combination of eidonomic and anatomical features not previously seen in any aquatic vertebrate, living or extinct: a long and narrow planform, an extended distal region without skeletal support, chordwise-parallel surface ornamentations, and a serrated trailing edge strengthened by chondroderms. Given the proposed adaptations for dim light vision in *Temnodontosaurus*[7,8], we hypothesize that the fleshy tip and above listed passive flow control devices contributed to a reduced acoustic and hydrodynamic signature, thereby allowing a stealthy approach to unwitting prey under the cover of darkness.

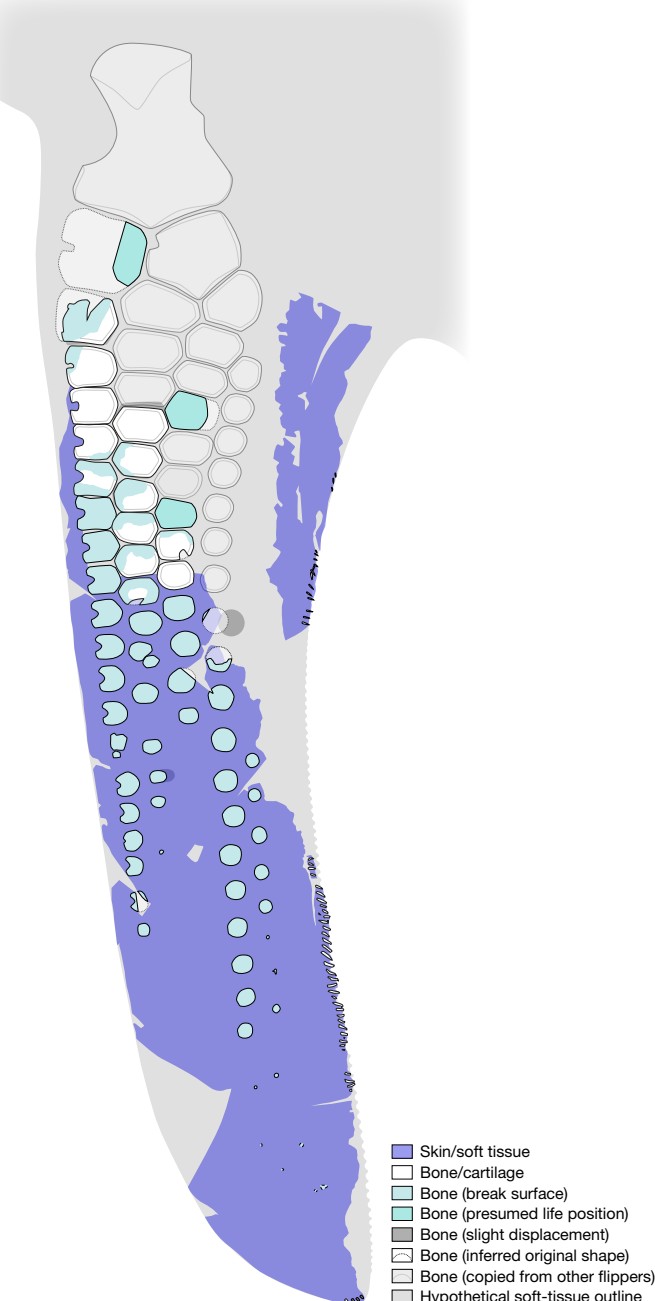

**Fig. 4 | Skeletal reconstruction and hypothetical soft-tissue outline of SSN8DOR11.** Missing elements (shaded light grey) are based on several individuals of *T. trigonodon*, all scaled to the same size. The extent of soft tissue along the trailing edge was estimated by interpolation of dimensional data and comparisons with front flippers of other parvipelvians (see Supplementary Information).

Legend:
- Skin/soft tissue
- Bone/cartilage
- Bone (break surface)
- Bone (presumed life position)
- Bone (slight displacement)
- Bone (inferred original shape)
- Bone (copied from other flippers)
- Hypothetical soft-tissue outline

As reconstructed (Fig. 4), SSN8DOR11 has a substantially higher aspect ratio than hitherto reported parvipelvian forelimbs[1,6] (Supplementary Information), in effect forming a wing-like appendage with high lift to drag properties to manipulate flow around the body[20,21]. In similarity with extant cetaceans[22], the skeletal elements were probably organized into digital rays with non-mineralized cartilaginous junctions (Extended Data Fig. 7), and further held together by encasing connective tissue to produce a semi-rigid, hydrofoil-shaped structure[13]. Although the interlocking carpals, metacarpals and proximal phalanges presumably only permitted limited mobility in life[1], the more widely separated distal phalanges probably facilitated a higher

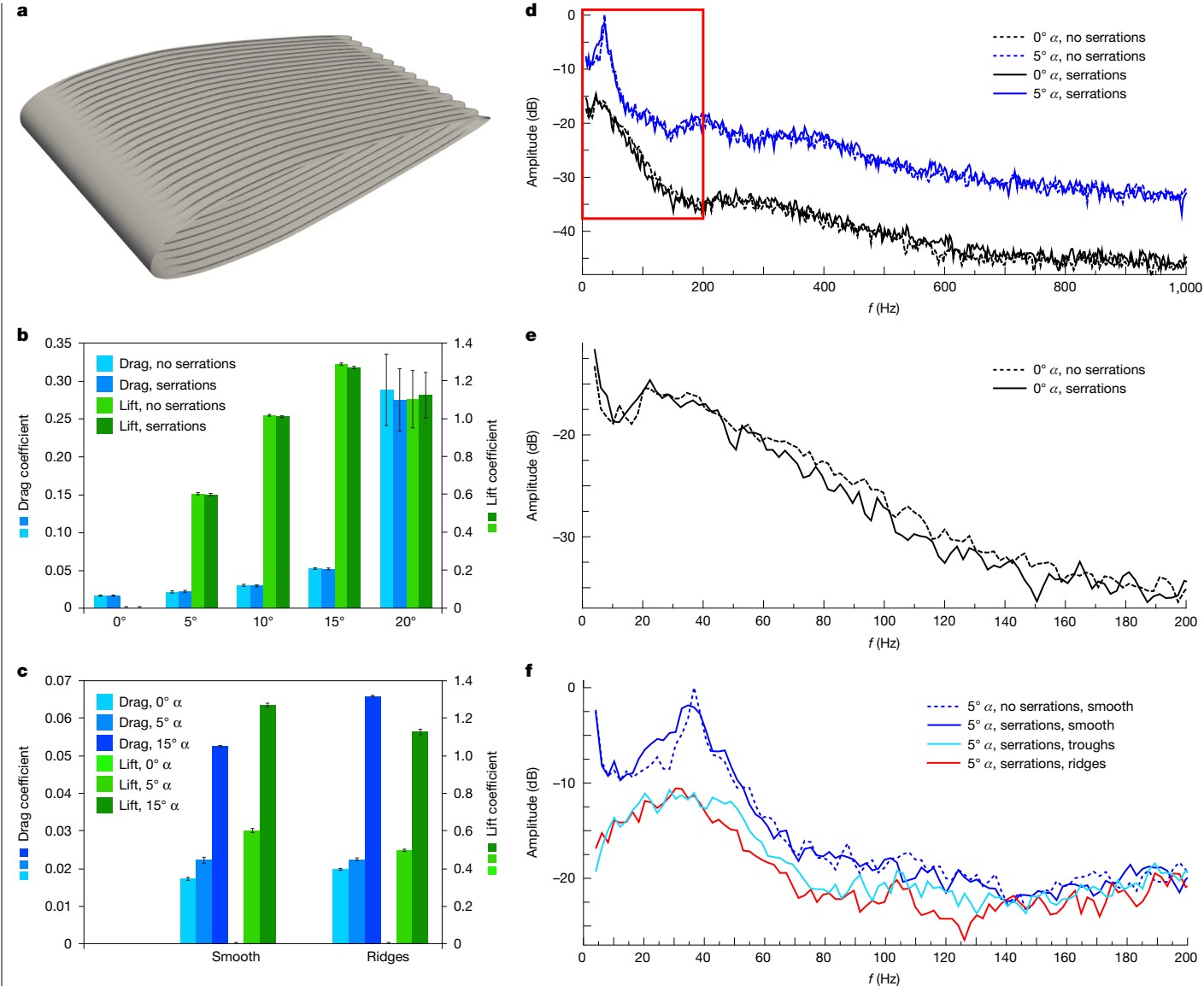

**Fig. 5 | Computational fluid dynamics simulations. a**, Flipper section used in the analyses, equipped with trailing edge serrations and ridges. **b**, Drag (blue) and lift (green) coefficients for geometries without and with trailing edge serrations as a function of angle of attack, α (0, 5, 10, 15 and 20°). Error bars represent s.d. **c**, Drag (blue) and lift (green) coefficients for a serrated fin section without surface treatments together with a serrated geometry featuring ridges at α of 0, 5 and 15°. Error bars represent s.d. **d**, Sound pressure levels (SPL) 50 m upstream of models with (solid lines) and without (dashed lines) serrations at α of 0 and 5°. The region in the red box is enlarged in **e** and **f**. **e**, Magnification of

the 0–200 Hz regime (St = 0–26.7) at α = 0°. Note that the addition of trailing edge serrations causes noise suppression over most of the depicted frequency range. **f**, SPL 50 m upstream of a serrated flipper section covered with either troughs (light blue) or ridges (red), together with a serrated geometry without these surface treatments (deep blue) and a fin section lacking passive flow control devices altogether (dashed curve) at α = 5°. Both surface embellishments result in noise attenuation, which is particularly noticeable below approximately 70 Hz (St = 9.3). Data were collected for 67.5 convection times. Spectra in **d**–**f** are based on 65,536 wide Hanning windows, 50% overlap and 33 samples.

degree of dorsoventral flexure (a condition possibly augmented by the subadult age of the individual to which the flipper belonged; Supplementary Information).

Bones are notably absent in the distal termination, and although fin ray-like elements have been previously documented in a few ichthyosaur fossils[13,23] (Extended Data Fig. 8a), no comparable spiny structures exist in SSN8DOR11. Instead, the tip presumably was supported chiefly by non-mineralized cartilage, an inference corroborated by incipient ossifications in the distal fin blade (Extended Data Fig. 7d,e). A predominantly soft, and thus rather flexible, flipper apex could potentially induce changes to the flow around the outer extremity. For instance, spanwise bending of the tip would improve the hydrodynamic efficiency while simultaneously reducing noise emissions by acting like a winglet to modify vortex generation[20,24].

Minute ridges are variably present on the skin of living odontocetes, as well as in other pelagic swimmers[25]. Moreover, loss of tension due to post mortem decomposition can result in additional creasing of the integument[5]. Although there is some resemblance between the stripes in SSN8DOR11 and wrinkles that deform decaying soft tissues (Extended Data Fig. 8b), their consistent chordwise arrangement, uniform spacing and even distribution across the fin blade (Figs. 1 and 2a,b), as well as highly repetitive nature of the constituent fibrous matter (Fig. 2c and Extended Data Fig. 4e), strongly suggest that they were present already when the animal was alive—a supposition that is further supported by the occurrence of virtually identical lines in a forelimb (SMNS 81842; Staatliches Museum für Naturkunde Stuttgart, Stuttgart, Germany) of the closely related *Eurhinosaurus* (Extended Data Fig. 8c–e). Certain surface treatments, such as riblets, finlets and troughs, can improve the

aerodynamic or hydrodynamic and acoustic performance of aerofoils and hydrofoils, respectively[26–28].

Although tetrapod osteoderms generally have a protective function[14], we propose that the chondroderms instead served to reinforce the otherwise fleshy trailing edge while concurrently ensuring enough flexibility to maintain efficient manoeuvrability during locomotion. The size and shape of the cartilaginous elements change over the length of the fin blade, a morphological transition that is also reflected in the soft-tissue outline of the trailing edge (Fig. 3a–d). Interestingly, the serrations seem to be most prominent at around 75% of the span (Fig. 3c), which corresponds to measurements obtained from the fringed trailing edge of owl wings and crenulated caudal fluke of the humpback whale, *Megaptera novaeangliae*[24]. Notably, trailing edge serrations have been shown to reduce noise at low frequency ranges[24,29].

## Computational fluid dynamics

We performed computational fluid dynamics simulations[30] to numerically examine the hydroacoustic effects of trailing edge serrations and surface treatments on a virtual section of SSN8DOR11 (Fig. 5a). The chord length, serration size and spacing of the ornamentations were based on direct measurements from the flipper at about 75% of the span, and the hydrofoil profile was constructed from the cross-section of the forelimb of the living minke whale, *Balaenoptera acutorostrata*[22]. The relative flow speed was fixed at 1.5 ms$^{-1}$—close to the estimated optimal cruising speed for *Stenopterygius*[21,31]—and the angle of attack ($\alpha$) was set to 0, 5, 10, 15 and 20°. Different arrangements were explored, consisting of fin geometries with and without serrations and/or surface embellishments in the form of chordwise ridges and troughs (the flattened nature of the fossil precludes a confident determination whether the ornamentations were originally raised above or sunken into the flipper surface). Further details about the setup are provided in Supplementary Information.

Despite its inherent limitations[32], our numerical approach indicates that both trailing edge serrations and surface treatments can have important roles in suppressing hydrodynamic self-noise (Fig. 5 and Extended Data Figs. 9 and 10), which is consistent with the findings of previous investigations[20,24,26–29]. Notably, for cruising and gliding conditions[22,33] at zero $\alpha$, the addition of serrations to our virtual replica resulted in an upstream mitigation (amounting to approximately 1–3 dB) of the emitted noise over a range of low (below 200 Hz, Strouhal number (St) = 26.7) frequencies (Fig. 5e), with negligible impact on the hydrodynamic performance (Fig. 5b). An accompanying reduction (5–10 dB) of the acoustic signature, particularly noticeable below around 70 Hz (St = 9.3), was further observed at zero and small (5°) $\alpha$ when either ridges or troughs covered the exterior of our serrated model (Fig. 5c,f), suggesting that the combined effect of these passive flow control devices can cause noise attenuation over a broad range of low frequencies (Supplementary Information).

## Mode of life

Judging by their often enormous proportions (skeletally mature individuals occasionally surpassed 10 m in total body length)[34,35] and inferred opportunistic dietary preferences[11,36], members of the genus *Temnodontosaurus* occupied the highest trophic levels in the marine ecosystems of the Early Jurassic[9–11]. Available osteological and morphometric data indicate that all species had elongate and relatively flexible bodies by parvipelvian standards[37]. Whereas lateral oscillatory motions generated by a long, but presumably rather low aspect ratio, fluke provided the primary means of propulsion, the large flippers have been interpreted as being involved in steering and possibly also thrust production at slow swimming speeds[23,37,38].

A conspicuous feature of *Temnodontosaurus* is its huge eyeballs; these are the largest of any vertebrate known[1,2,7], rivalling those of the

giant and colossal squid (of the genera *Architeuthis* and *Mesocychoteuthis*) in absolute size[8]. Although *Temnodontosaurus* must have relied on a keen sense of sight for survival, conflicting views exist whether its massive eyes developed as a response to needs for high visual acuity or more low-resolution tasks[2,8]. Regardless, there is broad consensus that the eyes conferred advantages at low light levels, and thus were well suited either for nocturnal life or deep diving habits[7,8]. Following this premise, it is conceivable that *Temnodontosaurus* operated primarily under dimly lit conditions using its unique flippers to reduce acoustic and hydrodynamic disturbances that could lead to early detection by prey and adversaries alike. High lift properties of the extended forefins probably facilitated calm gliding motions through the water with occasional sideways excursions of the posterior body and tail. Pressure and displacement fluctuations that might cause audible and/or mechanical sensations were likewise presumably kept at a minimum by the fleshy tip and passive flow control devices to allow stealthy searches and pursuits[24]. In addition, a reduction of perceivable water movements could have limited interferences with the animals' own sensory systems[39]. Even though ichthyosaurs are thought to have lacked sophisticated directional hearing[40], they were likely to have had means for low-frequency underwater sound detection[41]. On the basis of ultrastructural similarities between fossilized 'pores' (fig. 2i,j in ref. 5) and scale organs of extant reptiles (fig. 1c,d in ref. 42), it is even possible (albeit more speculative) that they possessed cutaneous receptors to sense water-borne mechanical stimuli.

The results of our computational fluid dynamics simulations indicate that suppression of self-induced noise by trailing edge serrations and surface treatments occurred primarily at low frequencies (Fig. 5). However, to be useful for a hunting ichthyosaur, this dampening should be within the detectable frequency range of its intended prey. Besides other ichthyosaurs, coleoid cephalopods are known from *Temnodontosaurus* bromalites[11,36], and they additionally are sensitive to low-frequency sounds[43,44]. By minimizing the emission of perceivable acoustic and hydrodynamic motions through optimization of its flippers, the large-bodied *Temnodontosaurus* could have mitigated the manoeuvrability advantage of these comparatively small-sized but agile invertebrates, thereby enabling greater flexibility in forage choice. It is thus plausible that a coevolutionary arms race in detection range between Jurassic parvipelvians and their prey resulted in a simultaneous selection for stealth and dim light vision in some forms (Supplementary Information).

Auditory cues are important sensory stimuli to seagoing animals not only in the distant past but also today[45,46]. Increased ambient noise from shipping activity, military sonar and offshore wind farms is therefore a growing concern because of its negative impact on aquatic life[45–47]. To reduce human-induced noise pollution (that is, anthropogenic masking of biotic sounds), the effectiveness of passive flow control devices, such as trailing edge serrations and surface treatments, on hydrofoils and aerofoils is currently being explored[24,26,48]. Our findings show that such features already existed in at least one lineage of ichthyosaurs 183 million years ago.

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

## Methods

### Fossil material

Initial efforts to re-assemble and mechanically prepare SSN8DOR11 were made at the time of its discovery in 2009. However, an investigative computed tomography (CT) scan in 2021 revealed the presence of additional bones that remained hidden within the matrix, and these were uncovered using a combination of chisels, dental picks and a microscribe.

An approximately 5-mm-thick section of a distal phalanx, an incipient ossification and three partial chondroderms were immersed in polyester resin (Araldite DBF, ABIC Kemi) to prevent shattering during slide preparation. Once embedded, approximately 1-mm-thick sections were cut from the blocks using a slow-speed diamond saw. Each section was attached to a petrographic slide with polyester resin and then ground to optical translucency. All sections were imaged using an Olympus BX53 system microscope equipped with an Olympus UC30 camera and an Olympus SZX16 stereo microscope fitted with an Olympus SC30 camera.

Two distal chondroderms were sacrificed for in-depth ultrastructural and molecular analysis. They were collected using a low-vibrational saw, μCT-scanned, washed multiple times with ultrapure (Milli-Q) water and ethanol (VWR, 96% rectapur), and stored loosely wrapped in aluminium foil prior to being treated with 8 ml 0.5 M ethylenediaminetetraacetic acid (EDTA, Panreac Applichem) at pH 8.0 in tissue culture plates (VWR). The buffer solution was exchanged on a daily basis for five consecutive days, and the soft, semi-transparent debris liberated during the demineralization process was then transferred in 50 μl portions to separate glass vials (VWR). Adhering skin was also isolated from the sediment and placed in separate vials. All samples were washed 8 times by removing all but 150 μl of the buffer solution (after all of the remaining solids had settled) and then adding 1.35 ml washing solution. Three different washing solutions were used depending on sample and subsequent analysis: for the chondroderm samples, aqueous ammonium formate (0.25 M, 5 times, followed by 0.15 M, 3 times, pH 7.0; Bioultra Sigma Aldrich) was used for time-of-flight secondary ion mass spectrometry (ToF-SIMS) and aqueous sodium chloride (0.25 M, 5 times, followed by 0.15 M, 3 times, pH 6.3; Bioxtra Sigma Aldrich) for infrared (IR) microspectroscopy, while Milli-Q water was used for the skin samples. After washing, all samples were transferred in 50 μl aliquots to silicon wafers or CaF$_2$ windows (12 × 1 mm, Eksma Optics) depending on analysis (ToF-SIMS or IR microspectroscopy), and left to air dry in a semi-closed box.

### Modern reference materials

Three deceased female harbour porpoise (*Phocoena phocoena*) calves (specimens 22-VLT000946, 22-VLT000947 and 22-VLT000981) were photographed and dissected at Statens veterinärmedicinska anstalt (SVA) in Uppsala, Sweden. These animals were provided as incidental fisherman bycatch in Swedish waters, and received in near-perfect condition. The flippers were removed with scalpels and knives before being transported to Lund University in containers filled with ice. Tissue samples from one of the flippers were immersed in a freshly prepared fixative solution, 2% paraformaldehyde and 2.5% glutaraldehyde in 0.1 M cacodylate buffer (pH 7.4) for 24 h at 4 °C. The samples were then dehydrated in a graded ethanol series and embedded in epoxy resin (Agar 100, Resin kit R1031) via acetone, which was left to polymerize for 48 h at 60 °C. Semi-thin (1.5 μm) light microscopic sections were then cut with a glass knife using a Leica EM UC7 Ultramicrotome, and mounted on objective glasses. Every second section was stained with Richardson's solution prior to examination using an Olympus BX53 system microscope equipped with an Olympus UC30 camera.

### Polarized and ultraviolet light photography

Photography was performed using a Canon EOS 600D camera equipped with 18–55 mm standard lenses. The camera was mounted onto an overhead rig, and both the focal length and aperture were kept at the same settings throughout the study. Photographs were taken in a dark room using specialized (polarized and ultraviolet) lighting to illuminate the fossil. Multiple photographs were taken at a distance of ~40 cm to the specimen and then merged into a single image and distortion corrected using Adobe Photoshop (v.CC 22.3.0).

Polarized light photography was done following the recommendations by Crabb[50]. Two LED camera lights (Neewer) equipped with custom-fitted polarizer sheets provided continuous light. A circular polarizer filter was affixed onto the camera lens and rotated until maximum cross-polarization was achieved. Photographs were taken at an ISO setting of 100 with an exposure time of 1 to 2 s.

Ultraviolet-induced visible fluorescence photography[51] was conducted using two adjacent facing 25 W Eurolite ultraviolet spotlights to illuminate the fossil. A Schott 455-nm longpass filter was taped onto a step-up ring and attached to the camera lens. Photographs were taken with an exposure time of 30 s and an ISO setting of 200.

### FEG-SEM and EDX

FEG-SEM analyses were performed using two different instruments. At RISE, both untreated and demineralized samples previously analysed by ToF-SIMS were coated with a 15-nm-thick film of gold/palladium and examined in a Zeiss Supra 40VP FEG-SEM instrument at an electron energy of 2.0 keV and a working distance of ~6 mm using the standard Everhardt-Thornley type detector (SE2) and the Zeiss SmartSEM v6 software. Elemental analyses and mappings were done using an energy-dispersive X-ray microanalysis (EDX) detector from Oxford Instruments (X-Max 50, 50 mm$^2$) at an electron energy of 15 keV and a working distance of ~8.5 mm. Collection and analysis of the EDX data were done using the Aztec software, v.3.3 and v.6.1 (Oxford Instruments Nanotechnology Tools Ltd).

At Lund University, both modern and fossil samples were coated with a 6-nm-thick layer of platinum/palladium and examined in a Tescan Mira3 High Resolution Schottky FEG-SEM fitted with both standard and in-lens secondary electron, as well as back-scattered electron, detectors at an acceleration voltage varying between 1 and 15 kV at a working distance of 3–15 mm. Elemental analyses and mappings were performed with a linked energy-dispersive spectrometer (X-MaxN 80, 124 eV, 80 mm$^2$) from Oxford Instruments. The EDX data were processed and analysed using Aztec (v.6.1) from Oxford Instruments Nanotechnology Tools Ltd.

### Transmission electron microscopy

Demineralized fossil skin was immersed in epoxy resin (AGAR 100, Resin kit R1031), which was left to polymerize at room temperature for 72 h, followed by 48 h at 60 °C. Ultra-thin (50 nm) sections were cut using a Leica EM UC7 Ultramicrotome equipped with a diamond knife, and mounted on pioloform-coated copper grids without further treatment or staining. All sections were examined in a JEOL JEM-1400 PLUS TEM at 100 kV. Micrographs were recorded with a JEOL Matataki CMOS camera using TEM Centre for JEM-1400 Plus software.

### X-ray computed tomography

The blocks containing the proximal portion of SSN8DOR11 were assembled and embedded in sand, and then scanned in a Siemens Definition Flash CT scanner (Siemens Healthineers). The rock slabs were examined at 140 kV, using a dual tube (flash scan) with a tube current of 950 mA and a pitch of 0.35 to enable sufficient signal through the entombing sedimentary matrix. Images were then reconstructed using a medium soft filter and reviewed as 1 mm slices.

### X-ray computed microtomography

X-ray computed microtomography was performed on two distal chondroderms (see 'Fossil material') using a ZEISS Xradia 520 Versa 3D X-ray microscope (4D Imaging Lab, Division of Solid Mechanics,

Lund University, Sweden). The chondroderms were scanned with a source voltage of 80 kV, and the manufacturer-supplied Le4 source filter was applied to reduce beam hardening effects. The subsequent tomographic reconstructions, using the ZEISS reconstructor software (XradiaReconstructorApp V11.0) with correction for the centre of rotation, provided the 3D image volume of cubic voxels with side lengths of 2.875 μm output as 16-bit tiff slices. The chondroderms were then segmented and virtually reconstructed from the scan slice data without down-sampling using the 3D Slicer 4.6.2[52] and Drishti 3.0[53] software packages.

**Synchrotron radiation X-ray tomographic microscopy (SRXTM)**
SRXTM was performed at the TOMCAT beamline X02DA of the Swiss Light Source (Paul Scherrer Institut, Villigen, Switzerland). Four samples collected from a single chondroderm were immersed in water to enhance the image quality, and then scanned with a beam energy of 12 keV (2–3% bandwidth). The transmitted X-ray radiation was converted into visible light using a 20-μm-thick GGG:Eu scintillator (the distance between the sample and scintillator was a few mm) and magnified with a ×20 objective. The projections were recorded with a sCMOS camera (PCO.edge 5.5). Two sets of tomographic scans were acquired: 'slow' scans (aimed at higher image quality) had 1,000 equiangularly distributed projections recorded during the rotation of the sample over 180°. For 'fast' scans (which lead to higher stability of the sample during acquisition), the number of projections was reduced to 500. The exposure time per projection was 100 ms. The data illustrated in Fig. 3h and Supplementary Video 2 were acquired using the 'fast' configuration. The reconstruction was made using an in-house version of gridrec[54], a software based on a gridding procedure. The cubic voxel side length of the resulting tomograms was 0.33 μm. The tomographs (as 16-bit .tiff images) were processed and analysed using the Voxler 3 software.

In addition, two skin samples were measured on the I12-JEEP beamline[55] at the Diamond Light Source using a 90 keV monochromatic beam with a high-resolution imaging camera equipped with a scintillator (Crytur), a custom radiation resistant visible light optical module (SILL Optics) with a resolution of 3.24 μm × 3.24 μm per pixel, and the commercial visible light sCMOS sensor, PCO.edge 5.5 (PCO imaging, now Excellitas). The samples were scanned at 2,400 angles with an angular resolution of 0.075 degrees per step. The tomographic reconstructions were performed using the SAVU system[56]. The reconstructed data were segmented and analysed with the 3D Slicer software.

**Time-of-flight secondary ion mass spectrometry**
ToF-SIMS analyses were carried out in three instruments (all by IONTOF): a TOFSIMSIV and M6 located at RISE in Borås, Sweden, and a TOFSIMS 5 at Chalmers Materials Analysis Laboratory (CMAL), Chalmers University of Technology, Sweden. Negative- and positive-ion data were acquired using $Bi_3^+$ primary ions (25–30 keV) and low energy electron flooding for charge compensation. High-mass-resolution data were obtained in the bunched mode ($m/\Delta m$ = 5,000–10,000; lateral resolution, 2–5 μm; 0.1–0.2 pA pulsed current) and high-image-resolution data were measured in the fast-imaging mode (lateral resolution, 0.1–0.5 μm; $m/\Delta m$ = 300; approximately 0.04 pA pulsed current). The fossil spectra were compared against spectra acquired for various reference materials, including *Sepia officinalis* eumelanin, calcium carbonate and hydroxyapatite (all from Sigma-Aldrich). Collection and analysis of the ToF-SIMS data were done using the SurfaceLab software v.6.7, v.7.1 and v.7.3 (IONTOF).

**IR microspectroscopy**
Hyperspectral images were recorded at the SMIS beamline at Synchrotron SOLEIL, France, with an Agilent Cary 620 microscope coupled to a Cary 670 FTIR spectrometer (Agilent Resolutions Pro 5.3.0 software, Agilent Technologies), using the internal thermal source. The microscope was equipped with a 128 × 128 pixels Lancer MCT Focal Plane Array detector. All images were recorded in reflection and high magnification mode with a ×15 objective, giving a field of view of 141 × 141 μm and a projected pixel size of 1.1 × 1.1 μm². A total of 512 scans were collected per pixel at 8 cm⁻¹ spectral resolution, and processed using Kramers–Kronig transforms to extract absorption coefficients.

Hyperspectral images were also recorded at the Centre for Environmental and Climate Science, Lund University, Sweden, with a Hyperion 3000 IR microscope (operated in transmission mode) coupled to a Tensor 27 spectrometer (Bruker OPUS 8.5 software). The images were collected using a 64 × 64 pixel MCT Focal Plane Array detector. The distance between the detector elements was 2.3 μm. In total, 1,024 scans were collected per pixel at 4 cm⁻¹ spectral resolution.

The chemical images were generated using the Quasar 1.7.0 package[57] and superimposed onto visible microscopy images. Heat maps were produced using the baseline corrected area of peaks of interest as indicated in Extended Data Fig. 6c,d.

**Computational fluid dynamics**
Owing to the very low Mach number (M = 0.001) and because acoustic fluctuations scale with the square of M, direct computation of the noise by solving a compressible set of Navier–Stokes equations is not possible. Therefore, we used a hybrid computational hydroacoustic approach, where the flow was calculated by solving an incompressible set of Navier–Stokes equations, and the sound propagation estimated from an acoustic analogy. All computations were done using the pimpleFoam solver, which is part of the OpenFOAM package (an open-source computational fluid dynamics software). The cfMesh utility was employed to generate a hex-dominant, unstructured mesh with multiple levels of local refinements in the vicinity of our virtual flipper section, as well as in the wake region of this geometry. The need to compute acoustic fluctuations inhibits the use of steady flow solvers. As a consequence, flow was resolved in time using large eddy simulations, with the wall-adapting local eddy-viscosity sub-grid scale model to account for turbulent fluctuations. The time evolution of the acoustic pressure was computed using the Curle acoustic analogy[58], which is an extension of the Lighthill acoustic analogy to account for the presence of solid surfaces. Because the acoustic wave propagation is not numerically resolved but instead analytically integrated at desired virtual microphone locations, these can be placed outside of the region covered by the flow solver. Post-processing was done using ParaView 5.7.0, Grace 5.1.25 and a custom tool[59] based on the open-source fftw3 library.

**Reporting summary**
Further information on research design is available in the Nature Portfolio Reporting Summary linked to this article.

## Data availability
The ichthyosaur fossil examined in this study (SSN8DOR11) is permanently accessioned into the collections of Paläontologisches Museum Nierstein, Nierstein, Germany, while the comparative porpoise samples are housed at the Department of Geology, Lund University, Sweden. All data required for assessing the conclusions are contained in the Article, Extended Data Figs. 1–10 and Supplementary Information. The X-ray computed microtomographic and SRXTM data (Fig. 3f,h and Supplementary Videos 1 and 2) can be downloaded via MorphoSource at https://www.morphosource.org/concern/biological_specimens/000744564 (ref. 60).

## Code availability
The custom codes used to generate the flipper profile and to evaluate the acoustic spectra are available at Figshare (https://doi.org/10.6084/m9.figshare.29136740 (ref. 59)).

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

**Acknowledgements** The authors thank O. Gustafsson for preparing the comparative porpoise samples and for assistance during the TEM analysis; A. Lindskog for technical advice; S. Svanberg and A.-L. Sahlberg for information on fluorescence photography and loan of ultraviolet filters; and C. Sandt and F. Borondics for assistance when using the SMIS beamline at SOLEIL. A. Neimane accommodated a visit by M.M. to SVA. R. and F. Hauff, and E. Maxwell provided access to comparative fossil specimens. The computational fluid dynamics computations were enabled by resources provided by the National Academic Infrastructure for Supercomputing in Sweden (NAISS) and Swedish National Infrastructure for Computing (SNIC) at LUNARC, which are partially funded by the Swedish Research Council (SRC) through grant agreements 2018-05973 and 2022-06725. We acknowledge the Paul Scherrer Institut, Villigen, Switzerland, for provision of synchrotron radiation beamtime at the TOMCAT beamline X02DA of the Swiss Light Source, the Diamond Light Source, Didcot, UK, for beamtime at the I12-JEEP beamline (experiment MG33954), and SOLEIL, Saint Aubin, France, for provision of synchrotron radiation facilities. Financial support was provided by a project grant (2020-03542) from the SRC and a research grant (20220563) from The Crafoord Foundation to J.L., a project grant (42011) from the Royal Physiographic Society of Lund to M.M., a project grant (SRC, 2019-03516) to M.E.E., and a research fellowship from The Royal Commission for the Exhibition of 1851, a project grant from UKRI STFC (MG33954) and a travel grant from the Western Interior Paleontological Society to D.R.L.

**Author contributions** J.L. conceived the project. J.L. and D.R.L. wrote the paper with contributions from R.-Z.S., M.M., J.R., S.S., P.S. and D.-E.N., and feedback from all authors. J.L. assembled the figures with input from all authors. G.G. collected SSN8DOR11 and F.O. prepared parts of the fossil. The taxonomic identification was made by D.R.L. M.M. performed the porpoise dissection. R.-Z.S. and J.R. conducted the computational fluid dynamics simulations, R.G.D.L.G. and M.J. undertook the polarized and ultraviolet light investigation, M.J. and J.L. demineralized the fossil samples, and P.S., C.A., M.J. and J.L. performed the light microscopy, FEG-SEM and TEM analyses, K.Y., S.A.H., F.M., D.R.L., R.A. and G.B. collected the computed tomography scans and M.H., M.E.E. and A.L. segmented the imaging data, P.S. and J.L. executed the ToF-SIMS experiments, and M.O.d.B., M.M., J.L., M.J. P.U., and P.P. recorded the IR microspectroscopic measurements.

**Funding** Open access funding provided by Lund University.

**Competing interests** The authors declare no competing interests.

**Additional information**
**Correspondence and requests for materials** should be addressed to Johan Lindgren.

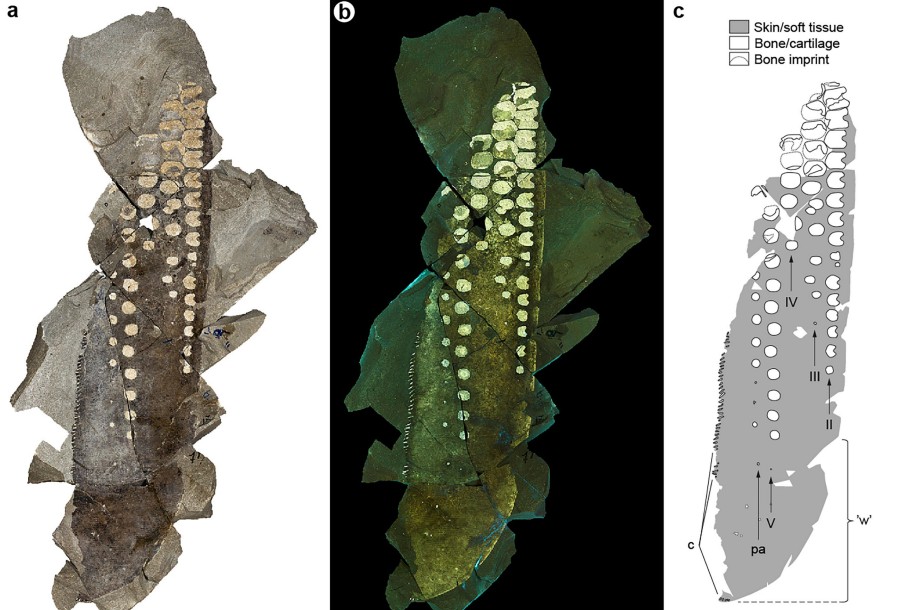

**Extended Data Fig. 1 | Front flipper of *Temnodontosaurus trigonodon* with soft tissues. a–c**, Photographs of the largest re-assembled slab containing the counterpart section of SSN8DOR11 under (**a**) polarized and (**b**) ultraviolet (longpass cut-off 455 nm) light, respectively, together with (**c**) an interpretative drawing of the limb in planform view. Arrow denotes anterior. II–V, digit II–V; c, chondroderms; pa, postaxial accessory digit; 'w', winglet-like distal segment. Scale bar, 10 cm.

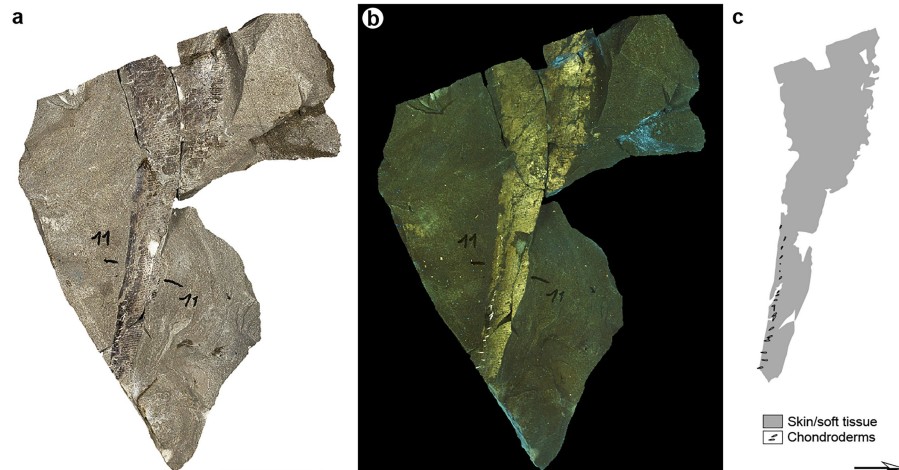

**Extended Data Fig. 2 | *Temnodontosaurus trigonodon*, patch of displaced soft tissue with associated chondroderms. a**–**c**, Photographs of the second (smaller) re-assembled slab containing the counterpart section of SSN8DOR11 under (**a**) polarized and (**b**) ultraviolet (longpass cut-off 455 nm) light, respectively, together with (**c**) a sketch of the detached structures in planform view. Arrow indicates anterior. Scale bar, 10 cm.

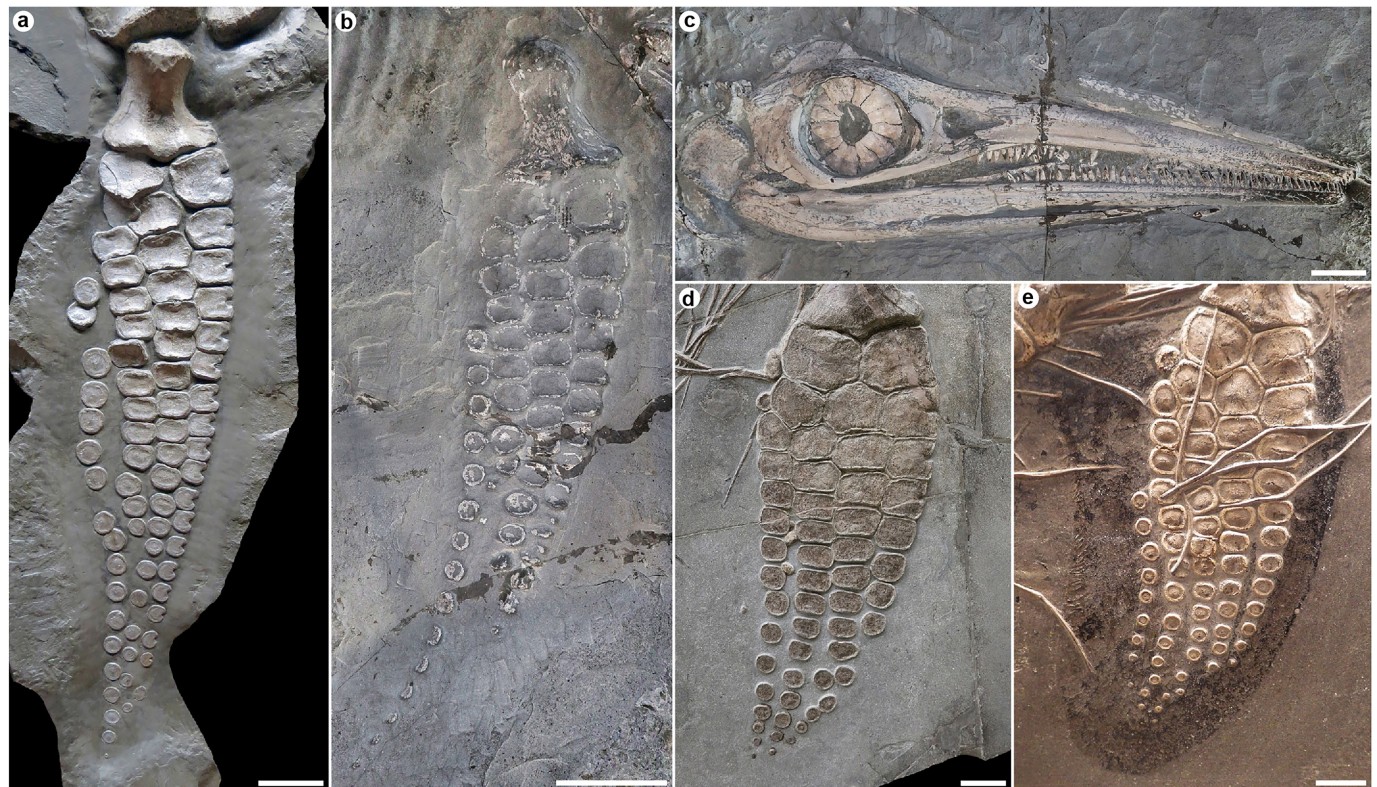

**Extended Data Fig. 3 | Parvipelvian forefins. a**,**b**, *Temnodontosaurus trigonodon*, specimens (**a**) SMNS 50000 and (**b**) UMH-0020 (Urwelt-Museum Hauff, Holzmaden, Germany), anterior to the right. **c**, Skull of UMH-0020.

**d**, **e**, *Stenopterygius quadriscissus*, specimens (**d**) SMNS 14846 and (**e**) GPIT-PV-30017 (Paläontologische Sammlung der Universität Tübingen, Tübingen, Germany), anterior to the right. Scale bars, 10 cm (**a**–**c**), 3 cm (**d**), 1 cm (**e**).

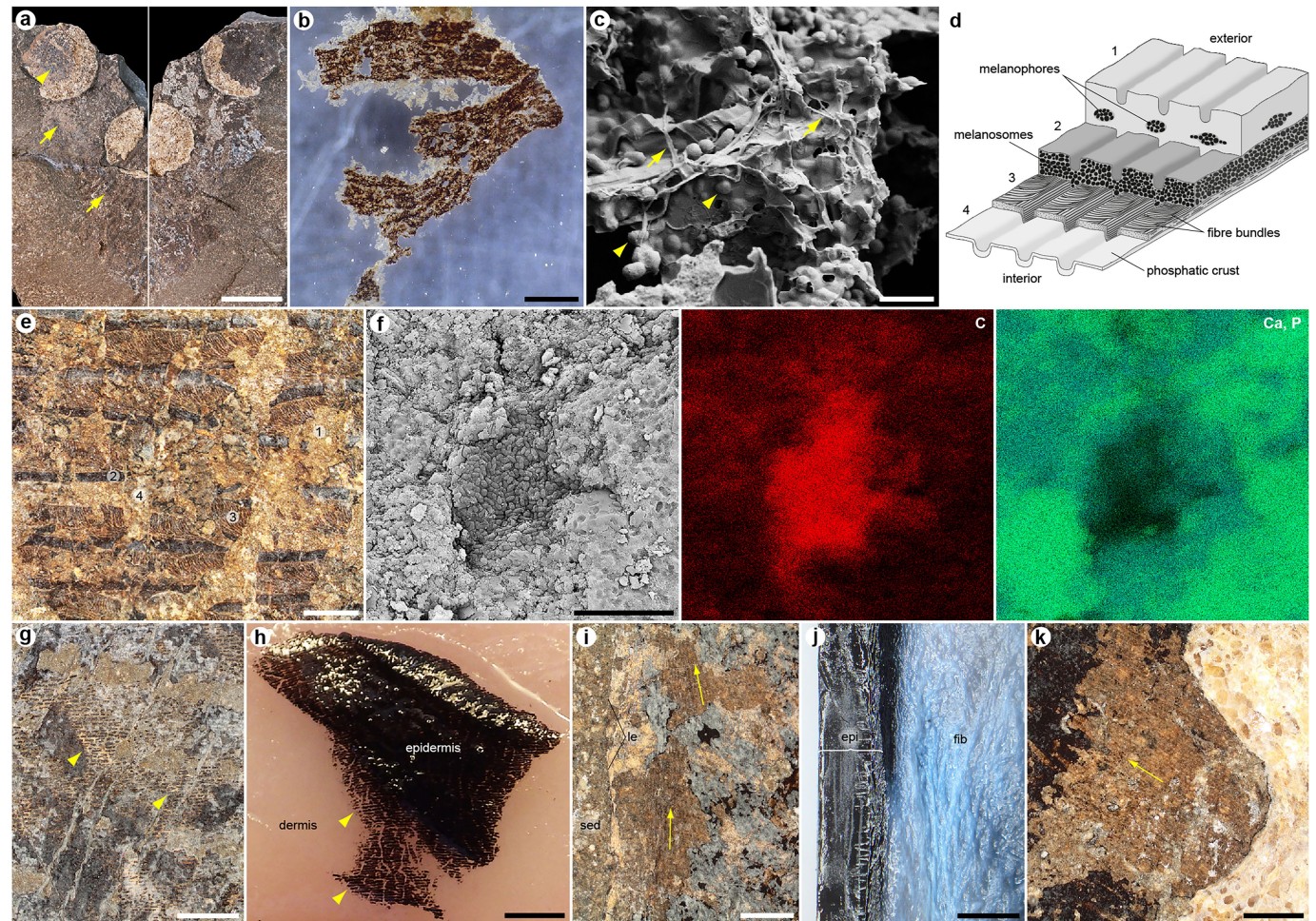

**Extended Data Fig. 4 | Soft tissues of *Temnodontosaurus* specimen SSN8DOR11. a**, Counterslab (left) and slab (right) sections showing soft tissues adhering to a phalanx (arrowhead), in addition to forming a bedding-parallel coating underneath the skeletal elements (arrows). **b**, Flexible organic residue after treatment with EDTA (*n* = 3 samples). **c**, FEG-SEM micrograph depicting melanosomes (arrowheads) and fibrous matter (arrows) liberated from demineralized soft tissue (*n* = 3 samples). **d**, Interpretative line drawing of the tissue layers preserved in SSN8DOR11 (not to scale): (1) epidermis with melanophores; (2) amassed melanosomes (representing the epidermal–dermal interface and superficial dermis); (3) fibre bundles; and (4) a phosphatic crust. **e**, Internal view of the stratified soft parts (LM micrograph; numbers as in **d**). **f**, Back-scattered electron micrograph and EDX maps of a melanophore (represented by aggregated melanosomes) in phosphatized matrix. Coloured images illustrate the relative abundance of each element, with higher intensities indicating greater abundance. Note localized enrichment of carbon (C; red) in the melanophore, whereas intensities from calcium (Ca; turquoise) and phosphorous (P; green) derive from the fossilized epidermis. Data are representative of five independent analyses. **g**, 'Dashed line' pattern (arrowheads) in the skin of SSN8DOR11. **h**, Comparable pattern (arrowheads) in *Phocoena phocoena* integument created by tangentially sectioned epidermal rete pegs. **i**, Leading edge (le) of SSN8DOR11. Arrows show the main fibre angles. sed, sediment. **j**, Leading edge of a sectioned *P. phocoena* flipper. epi, transversely sectioned epidermis; fib, fibrous tissues. **k**, Fibre bundles arising from the notched anterior margin of a phalanx of digit II. The primary fibre bundle direction is indicated by an arrow. Scale bars, 2 cm (**a**), 2 mm (**e**, **g**–**k**), 500 μm (**b**), 10 μm (**f**), 2 μm (**c**).

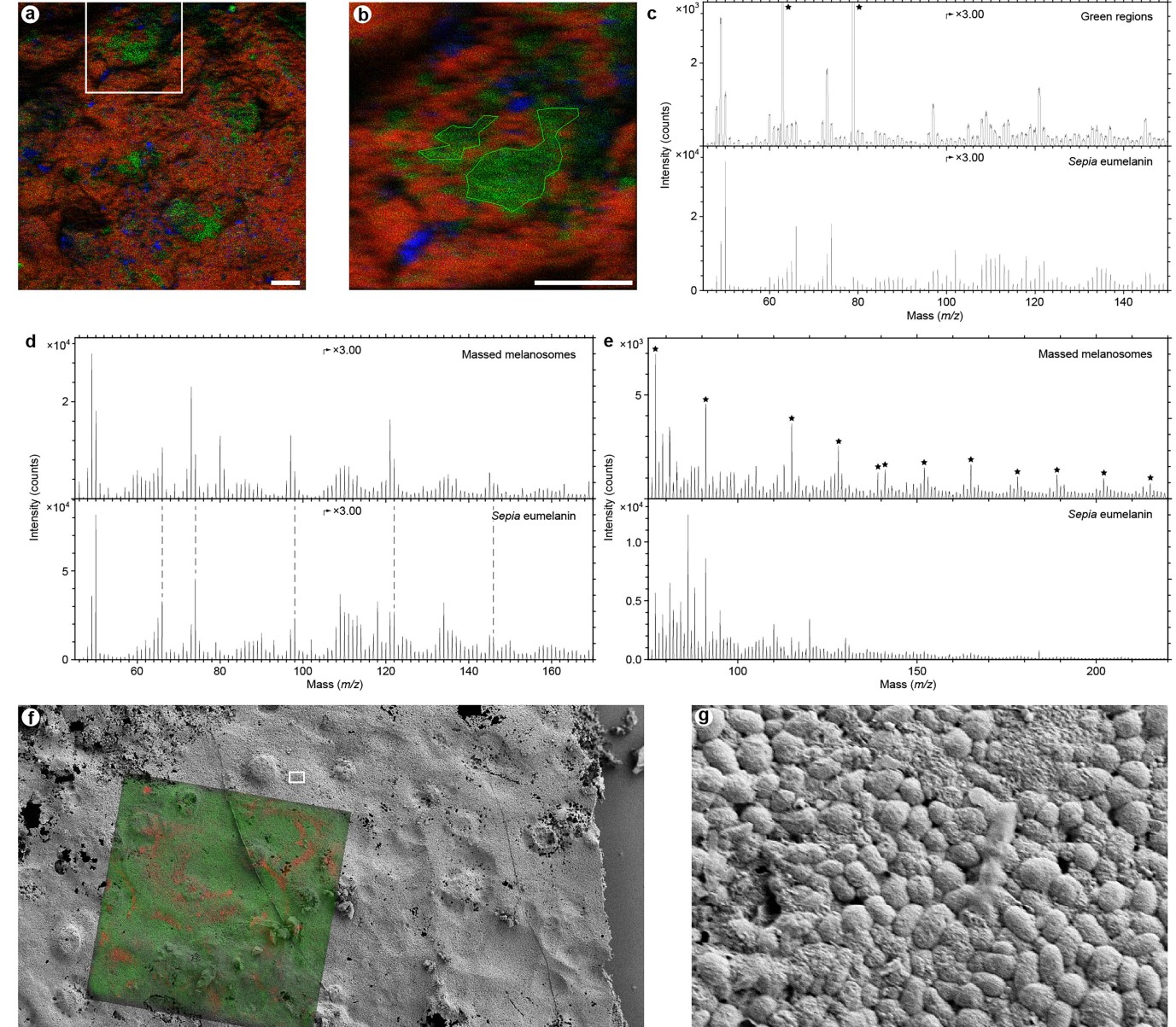

**Extended Data Fig. 5 | ToF-SIMS and FEG-SEM analysis of untreated and demineralised SSN8DOR11 integument. a**, Negative-ion ToF-SIMS image of untreated epidermis showing the spatial distribution of ions representing phosphate ($PO_2^-$ and $PO_3^-$) in red, eumelanin ($C_4H^-$, $C_6H^-$, $C_8H^-$, $C_3N^-$, $C_3NO^-$, $C_5N^-$, and $C_7N^-$) in green and silica ($SiO_2^-$, $SiO_3^-$ and $SiHO_3^-$) in blue. **b**, Enlargement of the area framed by a white box in **a**. **c**, Negative-ion ToF-SIMS spectra derived from the regions demarcated by green lines in **b** (a remnant melanophore) and a modern *Sepia officinalis* eumelanin reference sample. Phosphate ions ($PO_2^-$ and $PO_3^-$) are denoted by stars. Data are representative of eight independent measurements from four areas on one sample. **d**, Negative-ion ToF-SIMS spectrum obtained from the melanosome accumulation inferred to represent the condensed epidermal–dermal interface and superficial dermis, together

with a spectrum acquired from modern *S. officinalis* eumelanin. Note close resemblance between the two spectra, despite reduced intensities from N-containing ions (dashed vertical lines) in the fossil one. **e**, Positive-ion ToF-SIMS spectrum derived from the same melanosome mat as in **d**, together with a spectrum from *S. officinalis* eumelanin. Peaks characteristic of polyaromatic hydrocarbons are indicated by stars. Data are representative of three independent measurements from three areas on one sample. **f**, FEG-SEM micrograph of the melanosome mass from which the fossil spectra in **d** and **e** were acquired, partially overlain by a semi-transparent negative-ion ToF-SIMS image showing N-containing organics ($C_3N^-$, $C_3NO^-$ and $C_5N^-$) in green and silica ($SiO_2^-$, $SiO_3^-$ and $SiHO_3^-$) in red. **g**, Magnification of the white boxed region in **f**. Note densely packed melanosomes. Scale bars, 50 μm (**f**), 10 μm (**a**, **b**), 1 μm (**g**).

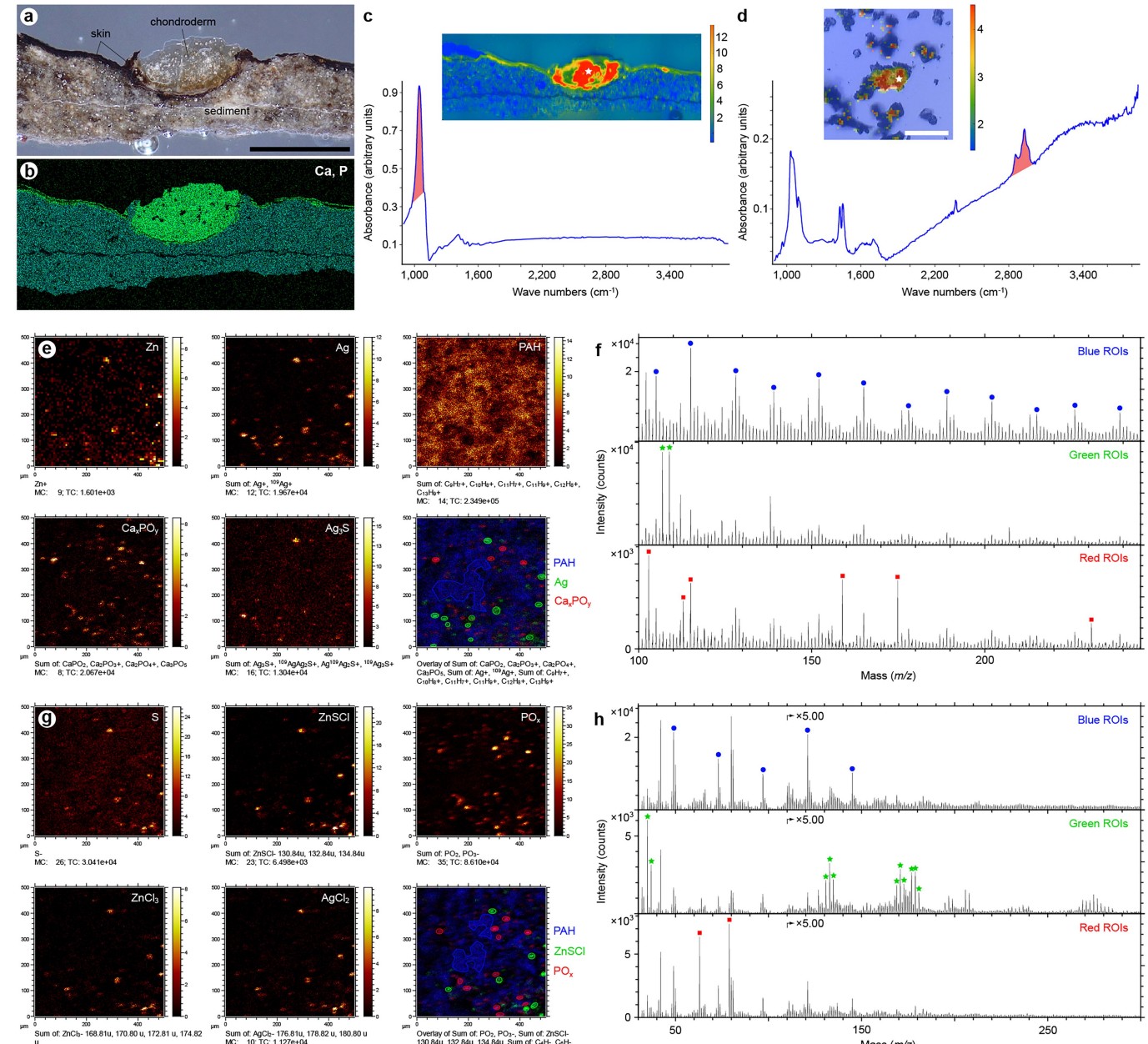

**Extended Data Fig. 6 | Chemistry of SSN8DOR11 chondroderms. a**, Ground section through a proximal chondroderm (*n* = 3 samples). **b**, EDX mapping of the two main chemical elements, calcium (Ca; turquoise) and phosphorous (P; green), present in the dermal mineralization (*n* = 3 analyses). **c**, Infrared spectrum from the sectioned chondroderm (star in the inset heat map; each pixel represents 512 coadded scans) showing a distinct peak at 1,044 cm$^{-1}$ that matches the asymmetric $v_3(P-O_4^{3-})$ mode of calcium phosphate. **d**, Infrared spectrum from organics (star in the inset heat map; each pixel represents 1,024 coadded scans) recovered from a demineralized chondroderm. The marked region, -2,800–3,000 cm$^{-1}$, corresponds to C–H stretching modes. **e**–**h**, ToF-SIMS data acquired from organic and inorganic matter liberated from a demineralized chondroderm. **e**, Images of positive ions showing sub-spherical ('cellular') bodies containing either calcium phosphate (Ca$_x$PO$_y$) or a mixture of zinc (Zn), silver (Ag), sulfur (S), and chlorine (Cl) embedded in an organic matrix comprising polyaromatic hydrocarbons (PAHs). **f**, Positive-ion spectra from the regions of interest (ROIs) indicated in the three-colour overlay image in **e**, representing the organic matrix (blue), Zn/Ag/S/Cl (green) and calcium phosphate (red), respectively. **g**, Images in negative mode (same area as in **e**) of ions associated with calcium phosphate (PO$_x$) and the Zn/Ag/S/Cl mixture, together with a three-colour overlay image of the two main types of chondrocyte-like bodies depicted in red and green, respectively, and 'typical' PAH ions in blue. **h**, Negative-ion spectra from the ROIs indicated in the overlay image in **g** representing the organic matrix (blue), Zn/Ag/S/Cl-containing particles (green) and calcium phosphate (red). Ions representing the three materials are listed in Table S1. Data in **e**–**h** are representative of eleven independent measurements from eight areas on two samples. Scale bars, 500 μm (**a**), 50 μm (**d**).

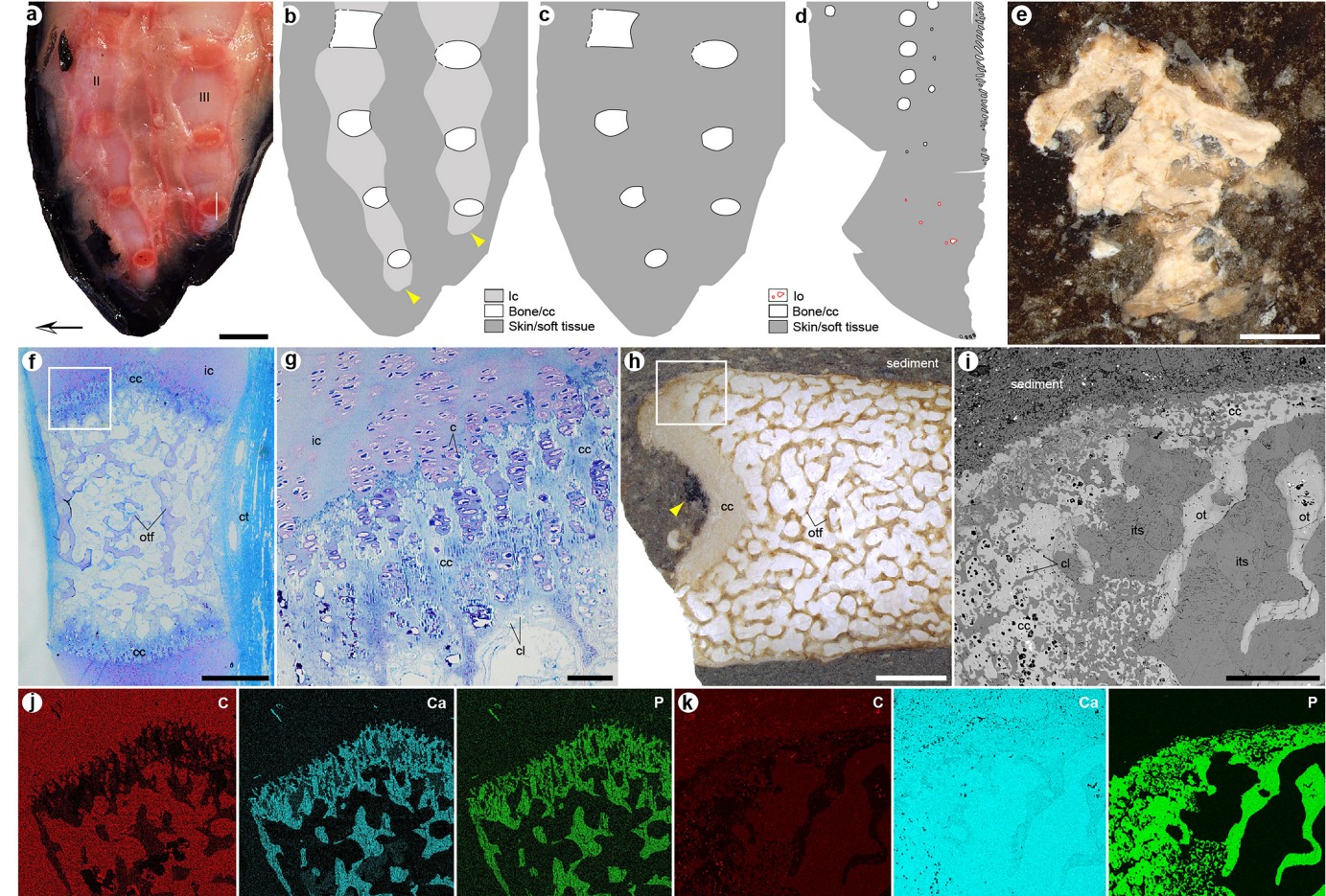

**Extended Data Fig. 7 | Taphonomy of SSN8DOR11. a**, Dissected *Phocoena phocoena* flipper to expose the skeletal system of digits II and III. Arrow denotes anterior. White line indicates the sectioned phalanx depicted in **f**, **g** and **j** (*n* = 3 specimens). **b**, Schematic representation of the forelimb where white indicates ossified elements (phalanges) and light grey un-mineralised interphalangeal cartilage. Note extension of the cartilaginous support beyond the distalmost phalanges (arrowheads). cc, calcified cartilage; ic, interphalangeal cartilage. **c**, Removal of all non-biomineralized structures (except the skin) mimics the condition in SSN8DOR11. **d**, Diagrammatic representation of SSN8DOR11 with incipient ossifications (io) in the distal fin blade. **e**, Photomicrograph of a distal ossification. **f**, Stained light microscopy section through the distalmost phalanx of digit III (*P. phocoena*). The region in the box is enlarged in **g** (*n* = 54 histological sections). ct, connective tissue; otf, osseous trabecular framework. **g**, Magnification of calcified and interphalangeal cartilage. c, chondrocytes;

cl, chondrocyte lacunae. **h**, Transversely sectioned SSN8DOR11 phalanx. The region demarcated by a box is shown in **i** (*n* = 2 ground sections). **i**, FEG-SEM micrograph of the calcified cartilage cap and neighbouring osseous trabeculae (ot). During diagenesis, the intertrabecular spaces (its) have been filled with calcium carbonate. **j**, Single-element EDX maps of the distalmost phalanx of digit III and associated tissues (*P. phocoena*). Note localized enrichment of calcium (Ca; turquoise) and phosphorous (P; green) in the calcified cartilage and osseous trabecular framework, whereas intensities from carbon (C; red) derive from surrounding tissues (*n* = 4 measurements). **k**, EDX maps of SSN8DOR11 differentiating increased levels of phosphorous in the calcified cartilage and osseous trabeculae versus calcium enrichment in the intertrabecular spaces (*n* = 4 measurements). Scale bars, 5 mm (**a**), 2 mm (**h**), 1 mm (**e**), 500 μm (**f**, **i**), 50 μm (**g**).

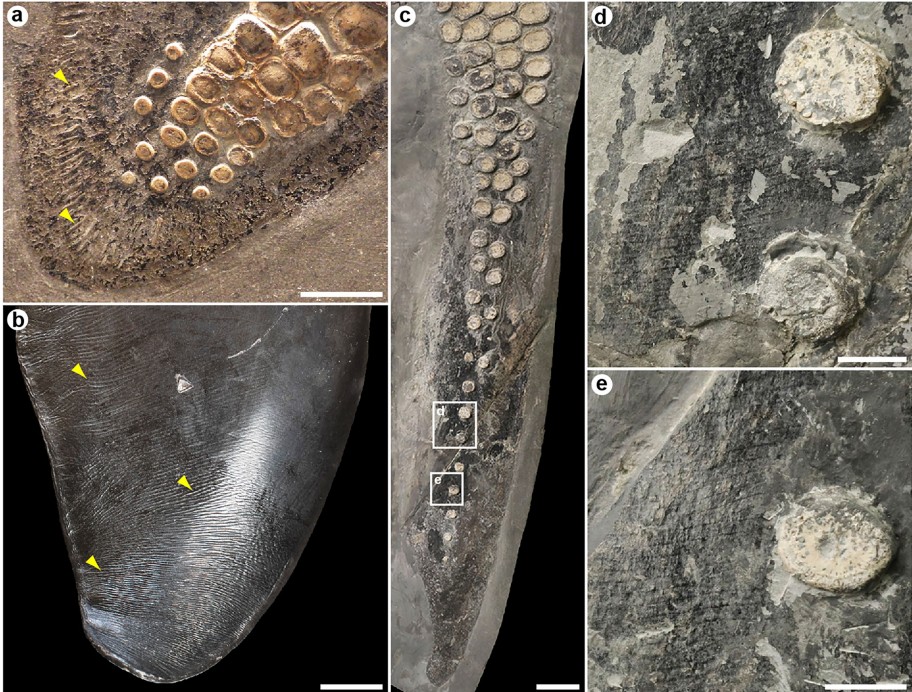

**Extended Data Fig. 8 | Marine tetrapod integument. a**, Hindlimb of GPIT-PV-30017, *Stenopterygius quadriscissus*. Arrowheads denote fin ray-like elements. **b**, Ventral view of the left flipper of a recently deceased *Phocoena phocoena*. Arrowheads indicate wrinkles on the skin. Some ridges may have been present already when the animal was alive; others emerged through post mortem tension loss to the dermal covering. **c**, Forelimb of SMNS 81842, *Eurhinosaurus quenstedti*. The regions demarcated by white boxes are enlarged in **d** and **e**. **d**, Magnification of two phalanges surrounded by remnant soft tissue displaying chordwise-parallel ornamentations. **e**, Close-up photograph of a second area with sub-parallel stripes. Scale bars, 5 cm (**c**), 1 cm (**a**, **b**, **d**, **e**).

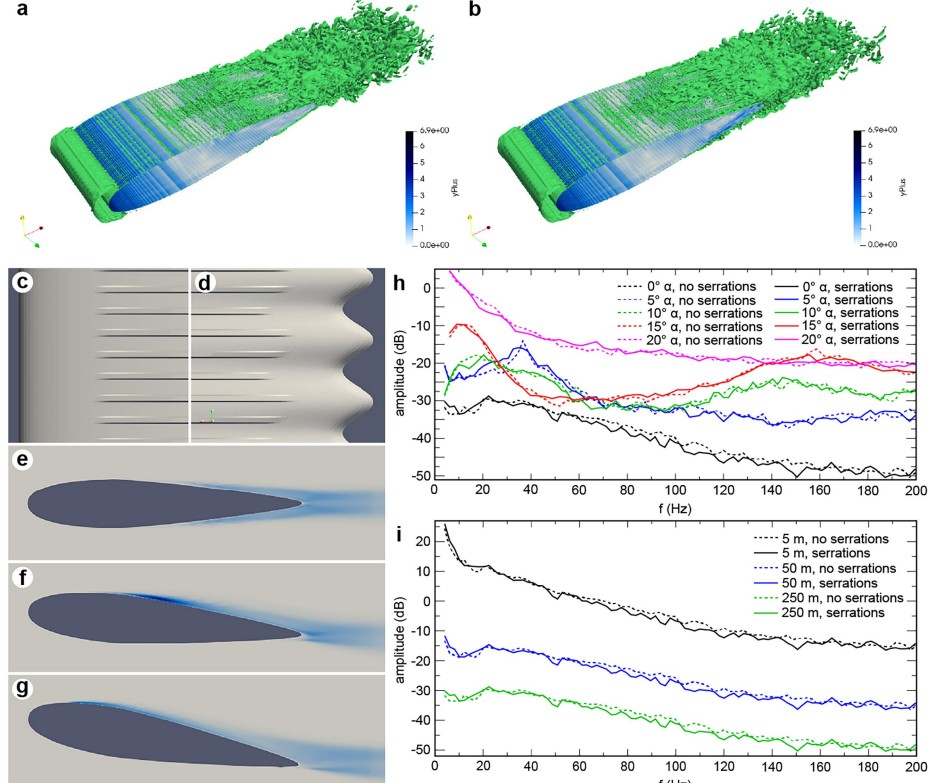

**Extended Data Fig. 9 | Noise attenuation by trailing edge serrations.**
**a**, **b**, Vortex structures visualized using the Lambda2 method for flipper
geometries (**a**) without and (**b**) with trailing edge serrations at 0° α. **c**, **d**, Details
of a virtual model featuring chordwise ridges and trailing edge serrations
(**c**, leading edge; **d**, trailing edge). **e**–**g**, Average velocity fluctuations observed
across a serration-equipped hydrofoil at (**e**) 0, (**f**) 5 and (**g**) 10° α, respectively.

**h**, Comparison of SPL 250 m upstream of flipper geometries with (solid lines)
and without (dashed lines) trailing edge serrations at five α (0, 5, 10, 15, and
20°). **i**, Comparison of SPL at three locations (5, 50 and 250 m) upstream of
flipper geometries with (solid lines) and without (dashed lines) trailing edge
serrations at 0° α. Data were collected for 67.5 convection times. Spectra in **h**
and **i** are based on 65,536 wide Hanning windows, 50% overlap and 33 samples.

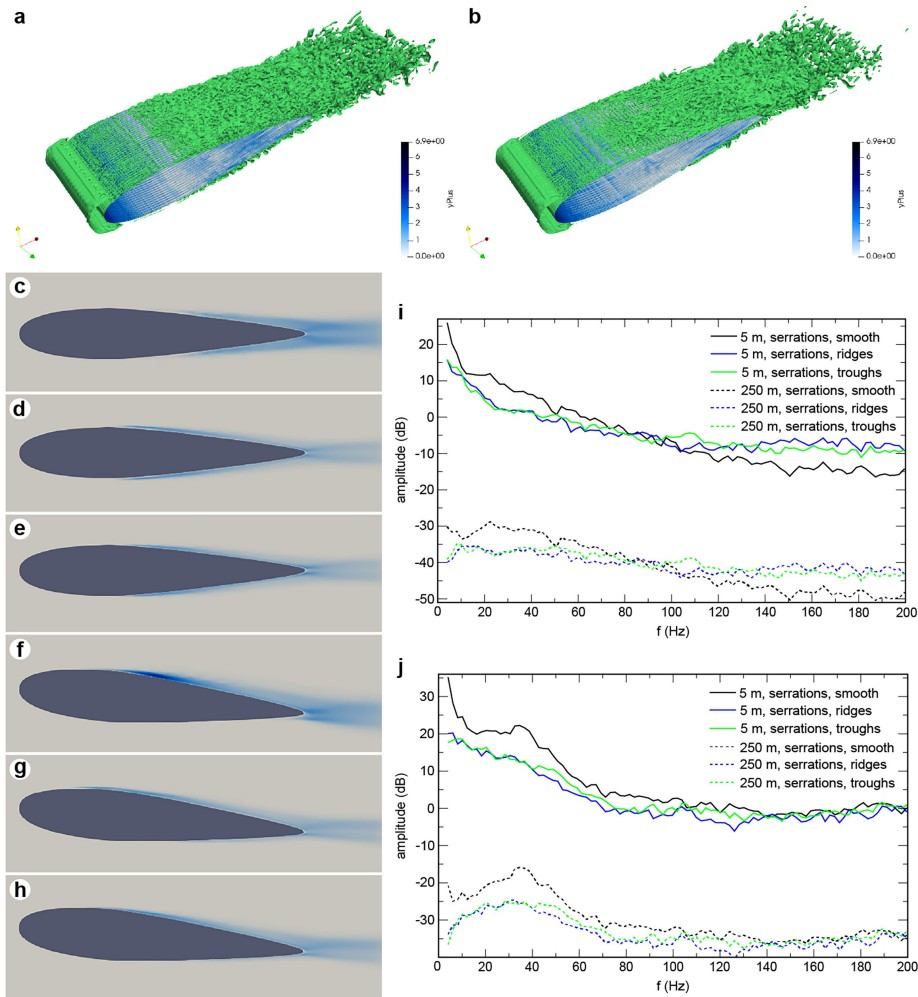

**Extended Data Fig. 10 | Noise abatement by surface treatments. a**, **b**, Vortex structures visualized using the Lambda2 method for serrated flipper geometries covered with either (**a**) ridges or (**b**) troughs at 0° α. **c**–**h**, Average velocity fluctuations for serrated hydrofoils exhibiting either (**d**, **g**) ridges or (**e**, **h**) troughs, compared against (**c**, **f**) a smooth (but serrated) model at (**c**–**e**) 0 and (**f**–**h**) 5° α, respectively. **i**, Comparison of SPL 5 (solid lines) and 250 m (dashed lines) upstream of a serrated flipper section covered with either chordwise ridges (blue lines) or troughs (green lines), compared against a smooth (but serrated) geometry (black lines) at 0° α. **j**, Comparison of SPL 5 (solid lines) and 250 m (dashed lines) upstream of a serrated flipper section covered with either chordwise ridges (blue lines) or troughs (green lines), compared against a smooth (but serrated) geometry (black lines) at 5° α. Data were collected for 67.5 convection times. Spectra in **i** and **j** are based on 65,536 wide Hanning windows, 50% overlap and 33 samples.

| | |
|---|---|

# Reporting Summary

## Statistics

For all statistical analyses, confirm that the following items are present in the figure legend, table legend, main text, or Methods section.

| n/a | Confirmed | |
|---|---|---|
| ☐ | ☒ | The exact sample size (*n*) for each experimental group/condition, given as a discrete number and unit of measurement |
| ☐ | ☒ | A statement on whether measurements were taken from distinct samples or whether the same sample was measured repeatedly |
| ☒ | ☐ | The statistical test(s) used AND whether they are one- or two-sided<br>*Only common tests should be described solely by name; describe more complex techniques in the Methods section.* |
| ☒ | ☐ | A description of all covariates tested |
| ☒ | ☐ | A description of any assumptions or corrections, such as tests of normality and adjustment for multiple comparisons |
| ☐ | ☒ | A full description of the statistical parameters including central tendency (e.g. means) or other basic estimates (e.g. regression coefficient) AND variation (e.g. standard deviation) or associated estimates of uncertainty (e.g. confidence intervals) |
| ☒ | ☐ | For null hypothesis testing, the test statistic (e.g. *F*, *t*, *r*) with confidence intervals, effect sizes, degrees of freedom and *P* value noted<br>*Give P values as exact values whenever suitable.* |
| ☒ | ☐ | For Bayesian analysis, information on the choice of priors and Markov chain Monte Carlo settings |
| ☒ | ☐ | For hierarchical and complex designs, identification of the appropriate level for tests and full reporting of outcomes |
| ☒ | ☐ | Estimates of effect sizes (e.g. Cohen's *d*, Pearson's *r*), indicating how they were calculated |

*Our web collection on statistics for biologists contains articles on many of the points above.*

## Software and code

Policy information about availability of computer code

| | |
|---|---|
| Data collection | Polarised and UV imaging: Adobe Photoshop (v.CC 22.3.0)<br>FEG-SEM/EDX: Aztec, version 3.3 and 6.1 (Oxford Instruments Nanotechnology Tools Ltd.) and Zeiss SmartSEM v6<br>TEM: TEM Centre for JEM1400 Plus software<br>X-ray computed microtomography: XradiaReconstructorApp V11.0<br>ToF-SIMS: SurfaceLab, versions 6.7, 7.1 and 7.3 (IONTOF GmbH, Münster, Germany)<br>IR microspectroscopy: Bruker OPUS 8.5 (Bruker, Ettlingen, Germany), Agilent Resolutions Pro 5.3.0 (Agilent Technologies Inc.)<br>CFD: OpenFOAM v2112, v2206 |
| Data analysis | FEG-SEM/EDX: Aztec, versions 3.3 and 6.1 (Oxford Instruments Nanotechnology Tools Ltd.)<br>X-ray computed microtomography: 3D Slicer 4.6.2 (3D Slicer image computing platform); Drishti 3.0 (VizLab, National Computational Infrastructure, Canberra, Australia)<br>SRXTM: Voxler 3, SAVU 4.2 (doi.org/10.5281/zenodo.6900630), gridrec (doi:10.1107/S0909049512032864), and 3D Slicer 5.2.2<br>ToF-SIMS: SurfaceLab, versions 6.7, 7.1 and 7.3 (IONTOF GmbH, Münster, Germany)<br>IR microspectroscopy: Bruker OPUS 8.5 (Bruker, Ettlingen, Germany), Agilent Resolutions Pro 5.3.0 (Agilent Technologies Inc.) and Quasar 1.7.0<br>CFD pre-processing:<br>- cfMesh, version 1.1.2<br>- Custom code used to generate the flipper geometry (59). (59) is a reference to the Figshare repository accompanying the submission.<br>CFD post-processing:<br>- ParaView 5.7.0<br>- Grace 5.1.25 |

- Spectra computed using custom code (59) based on the open-source fftw library. (59) is a reference to the Figshare repository accompanying the submission.

For manuscripts utilizing custom algorithms or software that are central to the research but not yet described in published literature, software must be made available to editors and reviewers. We strongly encourage code deposition in a community repository (e.g. GitHub). See the Nature Portfolio guidelines for submitting code & software for further information.

## Data

Policy information about availability of data

All manuscripts must include a data availability statement. This statement should provide the following information, where applicable:
- Accession codes, unique identifiers, or web links for publicly available datasets
- A description of any restrictions on data availability
- For clinical datasets or third party data, please ensure that the statement adheres to our policy

All data required for assessing the conclusions are contained in the Article, Extended Data Figures 1–10 and Supplementary Information. The raw X-ray computed microtomographic and SRXTM data (Fig. 3f, h and Supplementary Videos 1, 2) are provided in the accompanying MorphoSource (60) repository.
(60) is a reference to the MorphoSource repository accompanying the submission.

## Research involving human participants, their data, or biological material

Policy information about studies with human participants or human data. See also policy information about sex, gender (identity/presentation), and sexual orientation and race, ethnicity and racism.

| Reporting on sex and gender | N/A |
|---|---|
| Reporting on race, ethnicity, or other socially relevant groupings | N/A |
| Population characteristics | N/A |
| Recruitment | N/A |
| Ethics oversight | N/A |

Note that full information on the approval of the study protocol must also be provided in the manuscript.

# Field-specific reporting

Please select the one below that is the best fit for your research. If you are not sure, read the appropriate sections before making your selection.

☒ Life sciences ☐ Behavioural & social sciences ☐ Ecological, evolutionary & environmental sciences

For a reference copy of the document with all sections, see nature.com/documents/nr-reporting-summary-flat.pdf

# Life sciences study design

All studies must disclose on these points even when the disclosure is negative.

| Sample size | The sample size is limited to a single fossil specimen, SSN8DOR11 (Temnodontosaurus). |
|---|---|
| Data exclusions | No data were excluded from the analyses. |
| Replication | Five chondroderm, three skin, five bone/ossifications, and three cartilage samples were collected from SSN8DOR11 and its associated matrix; 12 of these were demineralised. All samples were photographically documented, together with comparative tissues from extant vertebrates. The following experiments were repeated independently with similar results: LM (histological sections): SSN8DOR11 five samples, Phocoena three samples; FEG-SEM: SSN8DOR11 15 samples; EDX: SSN8DOR11 10 samples; TEM: SSN8DOR11 four samples; X-ray computed microtomography: two sets of scans from two chondroderms; SRXTM: six sets of scans from six samples (four chondroderm and two skin samples); IRIS: each pixel in the hyperspectral image in Extended Data Figure 6c consists of 512 co-added individual scans, whereas each pixel in the corresponding image in Extended Data Figure 6d comprises 1,024 co-added individual scans; ToF-SIMS: eumelanin identification, eight measurements from four areas on one sample; eumelanin and PAH identification, three measurements from three areas on one sample; composition of chondrocyte-like bodies, 11 measurements from eight areas on two samples; CFD: the flow developed for 22.5 integral time scales (based on freestream velocity and chord length); statistics were collected for 67.5 integral time scales; the FFTs were computed using 65,536 wide Hanning windows with 50% overlap, resulting in 33 samples. The other experiments, including the X-ray computed tomography, polarised and ultraviolet light imaging and EDX analysis of Phocoena tissues, were not repeated as one set of data from each was deemed as sufficient. |
| Randomization | Not relevant to this study as no statistical analyses requiring randomization were performed. |

| Blinding | Not relevant to this study as no analyses requiring blinding were performed. |

# Reporting for specific materials, systems and methods

We require information from authors about some types of materials, experimental systems and methods used in many studies. Here, indicate whether each material, system or method listed is relevant to your study. If you are not sure if a list item applies to your research, read the appropriate section before selecting a response.

## Materials & experimental systems

| n/a | Involved in the study |
|---|---|
| ☒ | ☐ Antibodies |
| ☒ | ☐ Eukaryotic cell lines |
| ☐ | ☒ Palaeontology and archaeology |
| ☒ | ☐ Animals and other organisms |
| ☒ | ☐ Clinical data |
| ☒ | ☐ Dual use research of concern |
| ☒ | ☐ Plants |

## Methods

| n/a | Involved in the study |
|---|---|
| ☒ | ☐ ChIP-seq |
| ☒ | ☐ Flow cytometry |
| ☒ | ☐ MRI-based neuroimaging |

## Palaeontology and Archaeology

| Specimen provenance | SSN8DOR11 was collected by one of us (Georg Göltz) from a temporary exposure of dark, laminated limestone belonging to the Liassic εII5 ('Unterer Stein') part of the Early Toarcian Posidonia Shale in the municipality of Dotternhausen, south-western Germany. The fossil was discovered during construction blasting work, and consequently retrieved as a three-dimensional jigsaw puzzle of odd-sized rock slabs, collectively forming opposing part and counterpart sections of a flipper that has been cleaved along the sagittal plane. |
|---|---|
| Specimen deposition | The ichthyosaur fossil examined in this study (SSN8DOR11) is permanently accessioned into the collections of Paläontologisches Museum Nierstein, Nierstein, Germany. |
| Dating methods | Not applicable |

☒ Tick this box to confirm that the raw and calibrated dates are available in the paper or in Supplementary Information.

| Ethics oversight | This study deals with a single fossil collected from a temporary exposure of the Posidonia Shale in Germany; hence, no ethical approval or guidance was required. |
|---|---|

Note that full information on the approval of the study protocol must also be provided in the manuscript.

## Plants

| Seed stocks | N/A |
|---|---|
| Novel plant genotypes | N/A |
| Authentication | N/A |

