## [Peer Review File · Nature]

Adaptations for stealth in the wing-like flippers of a large ichthyosaur

Corresponding Author: Dr Johan Lindgren

Version 0:

Reviewer comments:

Referee #1

(Remarks to the Author)

With great interest I read about the amazing, fossilised ichthyosaur flipper from the Early Jurassic described by Johan Lindgren and colleagues.

The level of detail that the fossil preserves is outstanding, as are the numerous analytical approaches the authors conducted to study it. The performance of these all seems to be well conducted. Title and abstract are informative and cover the necessary details of the study.

In general, the manuscript is devoted to the soft tissue anatomy of the front flipper and its components, especially what the author's consider to be novel integumentary structure, so-called chondroderms. These descriptions are well embedded into the functional analysis of the flipper, discussing noise-cancelling properties which are proposed to enable a stealthier approach of these large Jurassic predators. For most parts the manuscript is well written and the images accompanying the text are more mostly adequate to support the interpretations.

Besides pointing out some spelling mistakes, I do have some more critical points that I would like to raise here for consideration by the authors.

Images of the fossil in Fig. 1 and in the supplement Figures 1 and 2: As the forelimb is encompassing slab and counterslab I tried to overlap images using a imaging program – unfortunately I had troubles doing so, because I encountered some distortion in the photographs. I tried to put these together in a tiff file, also to underscore my comments on the length of the flipper and the convergence typically seen in temnodontosaurus.

p. 2 line 33: “the giant Jurassic ichthyosaur” – Given the size range of ichthyosaurs in general with some taxa reaching >20 m body length, I think it would be better to speak of a large-sized ichthyosaur. Later in the text the authors themselves refer to it as e.g. “large-sized temnodontosaurid [...]”. Most Temnodontosaurus reached sizes of modern Orcas (or maybe a bit longer), also considered to be large but not giant.

p.2 line 50: “gradually attaining more fish-like bauplans” – even earliest ichthyosaurs had eel-like bodies, and as eels are also fish, this expression is not really correct and should be more specific

p.3 line 54: “both basal and derived” – is basal used in terms of “early branching forms”?

p.3 line 56 “small to moderate-sized species” – I understand what is meant here but I still like to advise caution here because as the authors know, size is not always a good predictor of trophic level

p.3 line 75: “The metre-long forelimb” – this is vague. Indeed exact measurements are missing from the text and the supplement so the reader is left wondering how large the flipper really is – given from the provided images, the flipper is ca. 80 cm as preserved. For comparison, SMNS 50000 shown in the supplement is ca. 120 cm, based only on the preserved bones and the soft tissue paddle would be even longer. As such it is difficult for the reader to assess the overall size and maybe also the ontogenetic stage of the fossil. I therefore suggest to add measurements and add more infos to the discussion.

p.3 line 76: if identification of the digits is based on Motani 1999 [currently ref 27] this paper should be referenced here already. In Motani 1999, the temnodontosaurus paddle usually encompasses digits II-IV + accessory digit, and not II-V as stated in line 75.

p.4 line 85: “can be confidently identified as a forefin of the large-sized temnodontosaurid *Temnodontosaurus trigonodon*” - Although I agree with the assessment that this forelimb likely belongs to *Temnodontosaurus* and not for example to *Eurhinosaurus*, another long-flipped form from the Early Jurassic, I found the assignment to be far from confident. All the temnodontosaur forelimbs I personally saw show distally converging digits II-IV and also the accessory digit. SSN8DOR11 on the other hand lacks this convergence instead presenting almost parallel extending digits. Furthermore, digits III and IV should be of similar length to digit II and the accessory one – in the new fossil, digits II and the accessory digit are clearly longer and digits III and IV appear truncated. I strongly suggest that the authors add to the discussion as it might well be that this specimen was affected by an early stage developmental aberration or even a pathology. This could in turn affect the non-convergence of digits and affect the shape and aspect ratio of the whole flipper.

p.4 line 88: please consider adding the locality of Dotternhausen which also offered many ichthyosaur fossils over the past decades

p.6, line 134: “occur at the bottom” – does this refer to the external-most layer then?

p.6, line 142: “epidermal–dermal interface and superficial dermis” – is the superficial dermis not part of the epidermal-dermal interface?

p.6, line 151 “integumental skeletal organs” – I would not consider them to be organs themselves but part of the organ “integument”

p.7, line 162: please explain a bit more what is meant by “territorial matrix”

p.7, line 167: “fossilised tissue” – does this refer to the globules?

p.7, line 171-172: “does not involve any cartilaginous precursor” – Scheyer 2007 (*J Anat Vol 211*) discussed the potential occurrence of cartilage in osteoderms (“postcranial fibro-cartilaginous bone” in osteoderms that are usually considered purely dermal in nature and lacking cartilage) of placodont reptiles. Please consider adding this to the discussion.

p.9, line 208: “allowing a stealthy approach” – orcas are considered to often use stealthy approach tactics when hunting marine mammals, but their forelimbs lack any serrations. Would it make sense to add a bit more to the discussion here?

p.9 line 209 “SSN8DOR11 has a substantially higher aspect ratio” – these data need to appear somewhere in the main text or the supplement – especially of long-finned forms such as *Eurhinosaurus* from Europe or *Guizhouichthyosaurus* from China.

p.9 lines 215-216: “permitted limited mobility in life” – was this ever tested? McGowan and Motani is a great volume but I am not sure whether this is the best reference for this sentence here.

p.9, line 219: “fin ray-like elements” – in my opinion this needs better documentation. Could you provide an image of these structures in the real fossil?

p.11, line 261: why was the living Minke whale chosen here?

p.13, line 311 “extant reptiles (ref. 43[...])” – is ref 43 not dealing exclusively with terrestrial taxa? Why was no aquatic or amphibious species added here (e.g. crocs with their dome pressure receptors)?

Ref. 24 and 25: please check names “Fish, F. E.” and “Fish, E. F.”

Figure legend Fig. 1: “proxo-posterior” – do you mean “proximo-posterior” here?

Figure legend Fig. 1: “sediment that has been digitally removed” – unclear to me what exactly is meant here – does this refer to the CT scan data?

Data availability: I would like to encourage to make the CT scan data openly available to support open data movement in Science.

Referee #2

(Remarks to the Author)

The paper examines new information on the morphology of the pectoral flipper from a fossil ichthyosaur as it related to hydroacoustic performance. The paper presents a description of an elongation of the flipper and the possession of periodic extensions from the trailing edge. The first half of the text is devoted to a description of the fossil, including a description of the morphology include soft tissue. Although I am not a paleontologist with an understanding of the nuances of taphonomy, these sections appear to be complete and well-written to indicate the strength of the description of the fossil to support the morphological findings. What is far more interesting is the interpretation of trailing edge structures (serrations) on the flipper. The authors make the case by analogy and use of CFD to state that the trailing edge serrations are to suppress hydrodynamically radiated noise. Such an effect would be useful for a nocturnal marine predator that is attempting to sneak up on potential prey without disclosing its presence. Although mentioned but not adequately interpreted are chordwise-

parallel surface ornamentations and the elongation and flexibility of flippers. Arguments and support for these morphological structures could be further strengthened and elaborated. There is a large literature base on trailing edge serrations from engineering indicating significant noise suppression and drag reduction. Although the authors cite works by Fish, Wolfe, and Fish & Lauder (22-24) examining noise reduction, there are some additional studies about trailing edge serrations, surface geometries, and finlets that would further lend support to the main hypothesis. These include:

Smith, T.A. and Klettner, C.A., 2022. Airfoil trailing-edge noise and drag reduction at a moderate Reynolds number using wavy geometries. *Physics of Fluids*, 34(11).

Hu, Y.S., Zhang, P.J.Y., Wan, Z.H., Liu, N.S., Sun, D.J. and Lu, X.Y., 2022. Effects of trailing-edge serration shape on airfoil noise reduction with zero incidence angle. *Physics of Fluids*, 34(10).

Ananthan, V.B. and Akkermans, R.A., 2023. Trailing edge noise reduction using bio-inspired finlets. *Journal of Sound and Vibration*, 549, 117553.

Clark, I.A., Alexander, W.N., Devenport, W., Glegg, S., Jaworski, J.W., Daly, C. and Peake, N., 2017. Bioinspired trailing-edge noise control. *AIAA Journal*, 55(3), 740-754.

The use of possible noise reduction from biology comes from the crenulations of the trailing edge of the flukes of the humpback whale and owl wings, although the original reports demonstrating noise suppression on owls was based on the leading edge rather than the trailing edge. These early works are cited in:

Blick, E. F., Watson, D., Belie, G. and Chu, H. 1975. Bird aerodynamic experiments. In: *Swimming and Flying in Nature*, Vol. 2 (T. Y-T. Wu, C. J. Brokaw and C. Brennen, eds.), pp. 939-952, Plenum Press, New York.

However for a cautionary note related to owl feathers and whale flukes, owls wings and whale flukes oscillate, whereas the flippers of the ichthyosaur are passive control surfaces that are used to stabilize the body when transiting and are used for maneuverability. It would be expected that oscillating structures generate noise while passive structures would generate substantially less noise unless there is boundary layer separation and stall promoting turbulence. The angle of attack used in the CFD analysis were well below the angle for stall. The chordwise-parallel surface ornamentation could potentially change the stall characteristic of the flipper as well as enhance drag reduction. There is only brief consideration beyond noise suppression of how the chordwise-parallel surface ornamentation and the elongation of flippers would play into drag reduction and increased maneuverability. Both would be important in foraging. Although the argument is made that the large ichthyosaur must use stealth to catch the smaller elusive prey, whereas maneuverability as well as speed and other behaviors can close the gap between predator and prey. Indeed, the enhanced lift properties of the flipper would be an advantage for increased maneuverability rather than facilitating "calm gliding motions". Enhanced lift for gliding would only be necessary if the ichthyosaur was negatively or even positively buoyant. Consideration of these drag reduction and maneuvering performance attributes should be more completely discussed. There is no evidence that *Temnodontosaurus* was a "vigorous ambush hunter". These characterizations, vigorous and ambush, seem to be opposites. Ambush predators are typically sit-and-wait predators and can only accelerate over very short distances. Large oceanic predators are generally cruisers and the more massive the animal the more restricted any acceleration. Characterization as being both vigorous and an ambush hunter seems inappropriate. Another area to strengthen is the assertions being made in this report regarding the specific acoustic range of the target species and see how the noise generated by the flipper is reduced but out of the target's range. Hearing in fishes are generally confined to the low frequency range of 800 to 1000 Hz. Although salmon have a range of 40-350, cod have a range of 20-38,000 and some clupeids have a range of 200-180,000 Hz. See the references below:

Popper, A.N. and Coombs, S., 1980. Acoustic detection by fishes. In *Environmental physiology of fishes* (pp. 403-430). Boston, MA: Springer US.

Ladich, F., 2013. Effects of noise on sound detection and acoustic communication in fishes. In *Animal communication and noise* (pp. 65-90). Berlin, Heidelberg: Springer Berlin Heidelberg.

Popper, A.N., Hawkins, A.D., Sand, O. and Sisneros, J.A., 2019. Examining the hearing abilities of fishes. *The Journal of the Acoustical Society of America*, 146(2), pp.948-955.

Putland, R.L., Montgomery, J.C. and Radford, C.A., 2019. Ecology of fish hearing. *Journal of Fish Biology*, 95(1), pp.39-52.

Fay, R.R., 2014. The sense of hearing in fishes. In *Perspectives on auditory research* (pp. 107-123). New York, NY: Springer New York.

Lastly, the very last sentence (331-334) seems to be an overreach. I might be nice to see a morphological design be integrated in modern technologies, but unless the geometry is truly novel and has nothing similar, then engineers will not be utilizing this design for biomimetic applications.

In general, the paper is very good with some exciting ideas, but needs to be strengthened with references relating to trailing edge hydrodynamics and acoustics for flipper elongation, trail edge geometry, fin flexibility, fish hearing, and consideration of maneuverability.

Frank E. Fish

Referee #3

(Remarks to the Author)

GENERAL

The discovery of a large, apparently well-preserved flipper of an ichthyosaur allows some inferences on the acoustic and hydrodynamic properties of the small structural properties thus revealed. There are 2 sets of geometric details – the presence of a serrated trailing edge, and the presence of small, regular streamwise structures that could be troughs or ridges. The possible influence was estimated in companion computations. The reconstruction and computations are the two key elements of the work.

It is the structure of many Nature articles to bury much detail in supplemental information, but the exercise of extracting it back out here was quite time-consuming. The computations are very important in this manuscript, and though certain details would be of interest to only a few, they are required for any evaluation of the computational models. There are some suggestions below for what details would be valuable.

Works based on fossil records are necessarily somewhat speculative, but as far as I can judge, those presented here are worth considering.

DETAILS

Fig. 5b: Here, straight lines are used to join 2 dots, from 0 to 5 degrees angle of attack. Though we might expect this relationship for C_l , we would not for C_d . In either case, this is too much interpolation.

Fig. 5c: Similar comment but here the line graph implies some continuous change on the abscissa, but the three points are actually categories. A bar graph would be better. There are no uncertainties on any of these data points, partly because they have appeared somewhat magically from a CFD package. But there should be some estimate of the sensitivity of C_l and C_d to assumptions and suppositions in the computation. Perhaps some variance could be obtained at least from the time history of the force coefficients.

P10, l241: Self-noise occurs mostly at low angles of attack and comes from shedding at the trailing edge. It is not a sure thing that such conditions are found for this postulated wing/fin. However, if we estimate the chord Reynolds number (it is never given) for a 20 cm chord (from Fig. 1) it is about 3×10^5 , and at this Re self-noise can occur at a wide range of α – it is actually the regime for the most likely tonal noise (see, for example, Probsting et al., J. Fluid Mech., 2014, Fig. 4). To estimate effects of geometry variation on the coherent structures formed over the wing, we would be interested in details close to the surface and on scales similar to the geometry variations (ridge height, spacing). These are not given.

P11, CFD: The CFD is LES, with some unspecified subgrid-scale model. The operating Re is not given, and neither is the Mach number, M . How are acoustic fluctuations resolved in this model? The span of the two-dimensional model is said to match the flipper length, but the model is a two dimensional slab and not a three-dimensional geometry, so the 'flipper length' is not relevant. It would be helpful to know the domain size in chords. It looks as though the span is about $0.2c$, when we are not sure that possible vortex modes are limited by the span. The run times are given in seconds but what is needed is how many convection times Ut/c . Similarly, frequencies should also be noted in Strouhal number, $St = fc/U$.

Fig. 5def: The acoustic spectra are compared for various conditions, but we have no sense of the variations/sensitivity/uncertainty of SPL at any given frequency. There is a difference of 2 dB. Is that a big number? Is it significant? There is some information and some interesting results here, but it is difficult to extract it from a combination of manuscript and extended material.

P10, ridges: Fig. 2a,b – the ridges are indeed remarkably regular. Their spacing is estimated in mm. Given likely Re , then how does the spacing compare with a likely boundary layer thickness?

Extended, somewhere: The virtual microphone is said to be 250 m upstream. How is this possible given the size of the computational domain, which extends forward only 2 m? Presumably a model is used for sound attenuation in sea water, but it would be helpful to have that given explicitly. Lengths should be rescaled by c . Some remarks are given on sound attenuation and some examples, scaled to the postulated animal size could be helpful. What are SPL $1c$ in front of the fin? How about $5c$ which looks to be about the location of the forward tip of the mouth? There are data for 5 m and for 250 m (!!) in front of the model, but it would be informative to have these lengths related to likely body sizes.

P11: "Although approximations and simplifying assumptions are necessary when modelling multiscale fluid flows and their interactions with complex biological structures, such analyses nonetheless provide valuable information about the core function(s) of specific features." Not sure this sentence says something useful.

P11, l262: More details of this hydrofoil shape would be useful. It is symmetric, and looks to be quite thick. At very least the thickness should be given and then chord location of maximum thickness. Is there a close NACA equivalent – I'm guessing the 0018.

P14: "Due to their ability to propagate fast and far underwater". How far, how fast?

RECOMMENDATION

There are three main concerns above. The first is that certain computational details need to be given, and preferably in the main text. The second is that some effort should be made to identify uncertainties in the force coefficients and acoustic SPL. The third is that the small-scale detail required to understand the hydrodynamic influence of the ridges and serration are not given; instead, we see from afar, the overall flow fields, or time-averaged fluctuations (extended fig. 9, 10).

It is easy for a reviewer to recommend further details in their speciality, and less easy to accommodate these suggestions in a compact form required by Nature. I am biased in my interest but it seems to me that some space in the reconstruction images and story could be saved and that the computations need to have more attention and precision so that some details buried in supplemental data, and others not yet presented could be given. In a short manuscript, providing complete details of either reconstruction or computation is probably impossible, but the focus could be on reconstruction details of direct relevance to the computations. Fig. 1, 2a,b, 4, 5 are all essential. Then some flow details on the scale of the ridges and serrations might be quite informative.

Noting this point, the speculations about the flexible tip and its influence are not supported by any of the computations here, and are probably unnecessary.

The manuscript has potential that would be better realized with a less tentative approach to numerical details.

Referee #4

(Remarks to the Author)

This article concerns a bio-inspired marine feature, albeit one that has already extinct more than a hundred millions years ago, for effective flow control and low noise radiation characteristics. After a thorough investigation of the sample (e.g. SSN8DOR11), a large aspect ratio flipper has been identified that contains multiscale elements and combination of eidonomic and anatomical features in several categories, such as the streamwise surface ornamentations in a form of finlets, and a serrated trailing edge (mostly at the outer-board region of the flipper) strengthened by chondroderms. The combination of the non-smooth surface texture and corrugated edge within an aerofoil mock-up were computationally studied using the freeware OpenForm, where a Large Eddy Simulation method was used. The results seem to indicate that the trailing edge serration can reduce self-noise from the mid-frequency range. If the trailing edge serration is combined with the surface texture in finlet, further noise reduction at low frequency can be realised. The results enable the authors to make an important statement in the article that the ancient ichthyosaur can catch their prey owing to the evolutionary low noise feature of their flippers.

I must declare that my field of study is neither the bioscience, nor the palaeontology, although I must also mention that I was fascinated and have enjoyed reading the first seven pages of the articles that focus on the description of the fossil features, and cartilaginous integumentary structures, including their figures and supplementary information. My peer review of this article probably starts from Page 8, when the narratives touch upon the subjects of functional implications, CFD and the analysis of the simulated results. I have also read the supplementary information especially Parts G, H and I.

My first comment is related to the trailing edge serration size relative to the aerofoil overall chord length. From the supplementary document, the serration has an amplitude of 4 mm. If I understood correctly, it scales the last 25% of the aerofoil chord. Does this mean that the overall aerofoil chord length is 16 mm, which seems to be too small to model the flipper properly? I might have missed something but I think this point needs better clarification. From the same line of enquiry, the amplitude-to-wavelength ratio of the serration you have chosen is 0.5, which is not necessary the most optimum configuration. Usually, a "sharper" serration like the one we observe from the trailing edge fringe of owl is more effective for self-noise reduction. In any case, I can accept that the nature of morphology might not only have low-noise characteristic in mind (e.g. a smaller aspect ratio for the serration might be better from the structural integrity point of view).

Second, the state of the simulated flow needs better clarification. What is the Reynolds number of the flow, especially at the trailing edge? If the Reynolds number is low, a natural transition to fully turbulent boundary layer at the trailing edge is unlikely. In this particular case, the self-noise generated at the trailing edge might have a distinct tonal characteristic that follows a particular frequency scaling law. On the other hand, for a relatively high Reynolds number, the radiated noise will be more broadband. The vortices depicted by the lambda2 criterion are for instantaneous snapshots, but they all seem quite intermittent to me, where a statistical fully-developed turbulent boundary layer at the trailing edge might not be a given. The presence of separation bubble at the suction side at 5 degree AoA represents a clue that the boundary layer at most of the surface is not yet turbulent. I must also comment that the authors only correlate the turbulent velocity fluctuations to the radiated noise, but the key noise source is actually the underlying surface pressure fluctuations. Some evidences of the reduction of the surface pressure fluctuations, especially for the streamwise surface ornamentations case, could be provided to reinforce the claims made in this article.

What is also missing in the article is a validation of the numerical methodology. I would compare some of the noise spectra obtained from the simulation against either the experimental data (from wind tunnel, not from the virtual aerofoil), or Direct Numerical Simulation (DNS) from the literatures so that the readers have the confidence of the predicted noise spectra, which are important evidences to substantiate the main story of the article.

Referee #5

(Remarks to the Author)
Lindgren et al for Nature

Dear editor at Nature,

Thank you for the opportunity to review an interesting manuscript, Adaptations for stealth in the wing-like flippers of a giant ichthyosaur, by Lindgren et al.

I accepted to review the paper because I have worked quite a lot on ichthyosaurs, including one paper on soft tissue preservation (using SEM-EDS and XRD), some using petrographic thin sections and some with use of CT data. I have not personally used all the methods that are used for obtaining the results in this paper, and I am not versed in fluid mechanics theory.

Summary

This manuscript by Lindgren et al reports on a well-preserved forefin from the Jurassic ichthyosaur *Temnodontosaurus*. In addition to most of the skeletal elements, the fossil preserves soft tissue of several different types, which shows the shape of the forefin and its structure. It preserves a series of smaller elements along its trailing edge that are suggested to be a new structure, chondroderms. The chondroderms, as well as other soft tissue components, are imaged and analyzed through several methods. This results in a description of components of the fin such as the dermis, melanosomes, its outer structure including ridges, as well as the chondroderms. The fin then possesses a set of characteristics that are hypothesized to be adaptations for reducing the underwater noise created by the animal itself during hunt for prey. These are analysed in a computational fluid dynamics framework in order to test whether this is actually possible. The study finds that the fin design might have reduced the underwater noise, and names this "stealth swimming".

Overall recommendation:

The forefin is a very nicely preserved fossil and I enjoy the idea behind the paper which I find novel and worth investigating, as well as the extensive methods usage in order to extract a maximum amount of data. I do like the structure built around the four-fold combination of features for stealth (summary from line 202) and think this works well. However, there are some points that I want to raise that should be considered and probably improved before this work is potentially published in Nature.

The full story of the paper (forefin structures including chondroderms were an adaptation to silent swimming) is at the moment interesting and backed by results, but some points weaken the argument, including a lack of discussion of other possibilities for the observed structures and a lack of modern day parallels. I have no reason to doubt the results of the methods that analyze the components of the preserved soft tissue. However, there is a lack of justification for methods used.

Main comments

1) Justification of methods

I miss a section justifying the methods chosen, that states what each of the methods contribute with. An impressive array of methods was used, producing interesting images and results on chemistry etc. However, as the article is presently written, it is hard to follow the steps of the experimental procedure. For instance, I would appreciate an explanation on which methods are used for assessing the taphonomy (alterations in chemistry after death) and the soft tissue structures (what is preserved from the original material), respectively, and for instance what the petrographic slides contribute versus what the other methods contribute. This also includes the usage of the three porpoise specimens. Comparing modern day cetaceans with ichthyosaurs is obvious and highly interesting, and it should be clearly explained what this part of the work contributed to and how the samples from porpoises compared to the ichthyosaur.

The justification for using the minke whale flipper shape for the fluid dynamics is good.

2) Description of the soft tissue composition and architecture (lines 89-137)

This part is the one I find the weakest of the manuscript in terms of clarity. As it stands now, it is hard to grasp which methods have been used for detecting what (see above), but also which parts of the preserved soft tissue contributes to the hypothesized noise reduction. I think it can strengthen the argument in the paper if it is made clear why the identification of the different types of soft tissue is important: e.g. is it seldom identified, does it contribute to stealth etc.

I also find it hard to picture from the text how the skin was built and want to suggest a figure showing the different layers and cells that are detected. This could potentially be interesting together with e.g. figure 2c.

I also suggest doing a full check on the terminology used, for instance what is meant by "stripes", "ornamentation", "surface treatments" and whether all of these terms are needed or might be overlapping. In the paragraph line 228-241, ridges (also used in caption for figure 5), riblets, finlets, grooves and fences are also mentioned. This also includes the usage of

“chordwise”, vs perpendicular, parallel etc. In the paragraph lines 228-241, I miss a concluding remark on how the ridges in the ichthyosaur specimen are interpreted – e.g. are they interpreted as “finlets adapted for noise reduction”.

The study detects pigment cells (eg. Pictured in 2d- g.), and it should be added what their role was. This includes whether they tell anything about coloration of the skin, and how they relate to the stripes and to the chondroderms?

3) Modern day parallels

As the text reads now, the case for noise reducing adaptations in living animals is not fully convincing with regard to the examples that are presented. The humpback whale is a documented example, however with a modified leading edge (as opposed to the trailing edge in the ichthyosaur) and caudal fluke. The Fish 2020 paper mentions the possibility in California sea lions, but this has not been tested. The owl wing example is interesting, but not in every way an ideal parallel for ichthyosaurs. Looking at the literature, I was not able to find additional examples of modern day parallels. This does not mean that noise reducing structures cannot be found, but it should be made clearer that this is not a widely found, well-documented phenomenon in modern animals at the moment. Some of the literature referred do not discuss adaptations in living animals, and are instead about physics, mechanics and design. That is very interesting, and relevant for marine, streamlined animals. It can be made clear where the literature refers to biology and in which cases it is about physical principles that also have to apply to animals.

In line 110, “living amniotes” are mentioned as a comparison. I suggest making clear what parts of these observations are made based on the experiments on the porpoise specimens done in this work, versus what is collected from the literature.

4) Other possibilities for the fin structures

In this article the authors test the forefin structures in the fossil specimen for its possible stealth-traits. That is interesting, and builds on theory that biological structures can inhibit or reduce noise (albeit not with very many modern day parallels, see point 3). However, the possibility that the structures are adaptation for other uses, is not sufficiently discussed. The paper states that osteoderms, which are relatively similar structures, in other animals usually are adaptations for protection (line 242), and that trailing edge serrations have been shown to reduce drag and alleviate stall (line 250).

In the fluid dynamics experiment (where I am no expert), drag and lift were calculated under different scenarios (fig 5b+c, and supplementary from line 278). These results could possibly be reported in the main text, in order to discuss the different possible effects that the fin design has. If this is not relevant, it should be stated that alternative hypotheses are not tested.

5) Noise generation by an animal under water

In the section on noise production by the animal’s own body, I miss a more thorough discussion on what body parts actually make the noise. I especially wonder about the caudal fluke, which was the main propulsive organ in parvipelvian ichthyosaurs. As mentioned in the paper, *Temnodontosaurus* was probably relatively flexible compared to some other taxa, but still a thunniform body (Buchholz 2001). The supplementary text (line 320) states that other parts of the body were not considered, and thus it is worth mentioning that these other parts probably made noise too. Do the results obtained here predict noise-reducing structures on the tail as well? Line 303 seems to suggest that it swam more with forefins than caudal fluke, which is surprising.

Other comments:

- First mentioned at line 247, the “~75% of the span” needs clarification. What does this mean? I am also not convinced (but this might be because I misunderstand the 75% reference) that this corresponds to a similar area or range in a caudal fluke, of which an animal only has one, and which is placed mid-line.
- Line 273; where the noise reduction is given, I miss a number that shows how much this is, i.e. 1-5 dB out of how many dB in total? As this is a core result to the paper, maybe some more text should be moved from supp mat?
- Line 322 coevolutionary arms race. This inference should be expanded and explained a bit more.

Figures

- Which methods are used for imaging should be consistently mentioned in all captions
- It might be an idea to combine figure 4 with figure 1, which would be space-efficient, and also good for comparisons.
- Figure 3J – I find this caption hard to follow and hope it can be elaborated.

In the methods section: In line 547-548 it says that experimental parameters were identical between ancient and modern samples. However, when looking at the Statistics and reproducibility paragraph, it does not seem to add up, because only a few methods were performed on the porpoise specimens. I hope this can be clarified, also in relation to point 1 above.

Data availability: There is no mention of where the porpoise specimens are held.

Thank you for a very interesting manuscript.

Regards, Lene Liebe Delsett

Version 1:

Reviewer comments:

Referee #1

(Remarks to the Author)

I thank the authors to have taken into account my previous comments and suggestions. I have no additional comments to make at this stage.

Referee #2

(Remarks to the Author)

Having now read the revision of the manuscript, it is improved and the authors have satisfactorily addressed my concerns. I find the article intriguing and it has taken our understanding of the biology of ichthyosaurs in a new direction. That the morphology of the flippers could have advantages in noise reduction for acquiring prey is a novel idea although based on hydro acoustics of living animals. However, there are a couple of points that could use additional references to help justify the assertions.

286-287- it is asserted that large flippers could be used for thrust production. A better reference for this statement comes from observations of humpback whales (Segre et al., 2017), which have high-aspect ratio, elongate flippers that use the motion of these appendages to swim at low speeds.

Segre, P. S., Seakamela, S. M., Meyer, M. A., Kindley, K. P. and Goldbogen, J. A. 2017. A hydrodynamically active flipper-stroke in humpback whales. *Current Biology* 27: R636-R637.

In addition, an article by Riess (1986) considered that a species of ichthyosaur with large pectoral fins could use them for swimming despite having a highly derived lunate tail fin.

Riess, J. (1986). Fortbewegungsweise, Schwimmbiophysik und Phylogenie der Ichthyosaurier. (Locomotion, biophysics of swimming and phylogeny of the ichthyosaurs) *Palaeontographica A*, 192, 93-155.

313-317- In this sentence, only acoustics are considered for a large-bodied ichthyosaur to prey on highly maneuverable prey. While reducing an acoustic signature may be important, as it is one of the main points of the paper, the large size and flexible body of the predator may come into play to capture agile prey. Maresh et al. showed that dolphins could use their body flexibility to out-manuever and cut off the escape for smaller prey.

Maresh, J. L., Fish, F. E., Nowacek, D. P., Nowacek, S. M. and Wells, R. S. 2004. High performance turning capabilities during foraging by bottlenose dolphins. *Marine Mammal Science* 20(3): 498-509.

Please consider these references in your arguments. Frank E. Fish

Referee #3

(Remarks to the Author)

The authors have responded to my and other reviewers' comments in a very thorough and logical fashion. Indeed, after reading 35 pages of comments and responses, one has a heightened appreciation of the task of publishing a paper that caters to and depends on multiple disciplines.

In my opinion, it is good to go.

Referee #4

(Remarks to the Author)

I appreciate the authors' efforts to address my previous comments and queries. I have carefully reviewed the rebuttal, revised manuscript (from the Functional Implication to the computational fluid dynamics), and the revised supplementary notes (specifically, Part H: Notes on Analyses and Experiments). Unfortunately, I find that some explanations in these documents raise more questions than they answer.

This paper must convincingly demonstrate that the surface texture (streamwise finlet ornamentations) and serrated trailing edge found in the wing-like flippers of a giant ichthyosaur can effectively suppress noise radiation. The authors' choice of a simple symmetrical aerofoil (NACA0018) to mimic the complex flippers of the ichthyosaur, whose exact shape, features, and taxonomy remain unproven, is an oversimplification, albeit an acceptable approach. Consequently, my examination focuses

on the feasibility of these finlets and serrated trailing edge, based on the evidence provided, to effectively reduce trailing edge self-noise from a symmetrical aerofoil under low turbulence inflow conditions. This implies that the turbulence-induced leading edge impingement noise is negligible and that no bypass transition mechanism is present on the aerofoil surface.

However, the clarification of a Reynolds number of only 0.3 million raises concerns that the authors may have generated an aerofoil noise mechanism not primarily susceptible to serration and finlets. The mechanism in question is aerofoil instability tonal noise, one of the five mechanisms proposed by Brooks et al. (1989, NASA technical report). This type of noise relies on boundary layer instabilities (such as Tollmien-Schlichting instability waves) that propagate to the trailing edge and are amplified by a local separation bubble before scattering into tonal noise in the far field (refer to the Bristol papers by Nash et al. and McAlpine et al., as well as Kingan and Pearse, JSV, 2009). Thus, the radiated tonal noise will be determined by the T-S wavenumber amplification growth rate, leading to similar dominant frequencies.

To suppress this type of boundary layer instability tonal noise using serration, Chong and Joseph (JSV, 2013) attempted to utilise serrations to disrupt the separation bubble, which serves as an effective amplifier for T-S waves. However, the short amplitude length employed by the authors may not effectively suppress the separation bubble, potentially explaining the low overall noise reduction observed in their noise spectra (Fig. 9h).

The authors' comparison of their predicted tonal noise with experimental results from Nakano et al. as a means of validating their computational algorithm—where significantly different tonal noise frequencies were observed—may be misguided.

It is also important to note that instability tonal noise is less prevalent under turbulent inflow conditions or in a three-dimensional flipper, as the separation bubble can be more easily suppressed. The real test for the serration and finlet lies in turbulent broadband noise, which I believe should be the primary focus of this paper. While I understand there is a broader palaeontological context in the manuscript, the low Reynolds number analysis, lacking external forcing terms (e.g., high inflow turbulence or boundary layer tripping), do not adequately address the critical research questions posed.

In addition, I have some comments and queries that I hope the authors can address:

1. In the main manuscript, lines 239–240, “streamwise finlet ‘fences’ with a spacing of 2mm...”: This sentence should be revised, as dimensional terms are not typically used to describe an optimal geometrical parameter. The optimal size should be presented non-dimensionally, scaled with a representative boundary layer length scale.
2. In the main manuscript, lines 248–250, the statement that trailing edge serration can reduce drag and alleviate stall is quite extraordinary. I suggest deleting this sentence along with the quoted references.
3. In the main manuscript, line 253, the word “experimentally” should not be used to describe the work, as it could be misleading. Since OpenFOAM is used as the research tool, I recommend replacing “experimentally” with “numerically.” This suggestion applies throughout the paper.
4. In the main manuscript, line 272, the Strouhal number is used but without clarification. Could the authors specify the length scale used in the Strouhal number?
5. In the supplementary note, under the “Noise attenuation by passive flow control devices” section, “... Passive flow control devices introduce small disturbances into the flow, which enhance the breakup of larger vortices, in turn causing noise attenuation...”: This statement is overly broad and not entirely accurate. For example, serrated trailing edges are not used to break up large-scale vortices.
6. In the supplementary note, under the “Constructing the flipper geometry” section, replace “experimentally” with “numerically.” The same change applies throughout the rest of the notes.
7. In the supplementary note, under the “Constructing the flipper geometry” section, “The height of both surface treatments increased linearly from 0 to 0.8 mm between 5 and 10% of the chord, and decreased again to 0 mm between 90 and 95% of the chord.” What is the context of these heights when scaled with the wall unit? It's important to understand whether the treatments are within the inner layer or protrude into the outer layer.
8. The title of the section “Experimental set-up” should be changed to “Numerical set-up.”
9. In the supplementary note, under the “Impact of trailing edge serrations” section, “The strongest fluctuations occur on the suction side, in the region where the transition to turbulence takes place, which suggests that the separation point is unsteady.” This sentence is difficult to understand. Did transition actually occur?
10. “For the $10^\circ \alpha$ case, the separation occurs near the leading edge, but the maximum fluctuation levels are lower than in the previous ($5^\circ \alpha$) case.” Does the suction side boundary layer undergo a complete stall at 10° or 15° ? It's unclear if the trailing edge serration would work in this regime.
11. “The largest differences in the radiated noise are observed at this α , which can be explained by the trailing edge serrations extending further into the streamlines than at $0^\circ \alpha$.” This sentence is unclear and difficult to interpret.
12. In the supplementary note, under the “Noise abatement by surface treatments” section, “It is further noteworthy that the

lighter blue colour in the wake (Extended Data Fig. 10d, e) indicates that the velocity fluctuations are smaller in this region when surface treatments are added to our model." The reduction in velocity fluctuations in the wake region due to surface treatments does not necessarily indicate noise reduction benefits.

13. "Similar conclusions can be drawn from the results obtained at $5^\circ \alpha$ (Extended Data Fig. 10f–h); that is, surface treatments lead to earlier development of vortices and a reduction of the maximum velocity fluctuation magnitude." What is the mechanism by which the finlet reduces the maximum velocity fluctuations?

14. "This decrease is attributed to the small vortices generated by the ornamentations that trigger flow separation." Does the ornamentation trigger or suppress flow separation?

15. "At $0^\circ \alpha$ (Extended Data Fig. 10i), there is a significant ($\sim 4\text{--}5$ dB) noise reduction in the low-frequency (< 50 Hz, $St = 6.7$) region for serrated flipper sections equipped with either ridges or troughs." Do the authors have noise results for surface treatments alone? It's possible that much of the noise reduction is due to the surface treatment, rather than the serration, as Figure 9 suggests the serration does not significantly reduce noise. Are the authors suggesting that the low-frequency noise reduction is caused by the breakdown of large-scale turbulence in the wake? The noise spectra appear quite tonal.

16. "This is presumably because small-scale vortices are present in the separation region surrounding the smooth flipper as well." The connection between higher frequency noise and the separation bubble requires more evidence.

Referee #5

(Remarks to the Author)

Thank you for the opportunity to review this paper a second round. I find that the authors have responded very thoroughly to my comments, and also to the comments of the other reviewers. All in all, the second round improved a paper that was already good and I believe it can be published without any further changes.

Version 2:

Reviewer comments:

Referee #2

(Remarks to the Author)

At this point it appears that the authors have made the appropriate changes for the manuscript. My only comment refers to lines 284–285, where it is stated that the large flippers could possibly produce thrust as referenced by Bucholtz (2001). The only place that any mention of the thrust production aside from the caudal fin occurs in Bucholtz is in regard to the "large hind limbs" for various ichthyosaurs (left column, p. 71). If the authors are making a case for the fore (pectoral) flippers to produce thrust and it is indistinct which flippers the authors are referring to, a better reference is Riess (1986), who speculated on this for ichthyosaurs and more particularly by Segre et al. (2017). My justification for including Segre et al. is that there is actual video evidence of a large aquatic predator (humpback whale) using its elongate pectoral flippers for propulsion. Bucholtz and Riess merely speculate without any visual evidence, other than morphological evidence, that the pectoral fins can be used for propulsion. Segre et al. provides real evidence that of the possibility and thus strengthens and supports the author's assertion.

Referee #4

(Remarks to the Author)

I would like to begin by thanking the authors for their efforts in addressing my comments during this second round of review. I appreciate that the primary focus of the manuscript relates to the biological science, and I acknowledge the interdisciplinary nature of the work.

However, I must express my concern regarding the authors' remark that "...we find the intense focus on our simulations to be blown a bit out of proportion." I would reject this characterization. The central premise of the paper is to explore the stealthy predatory capabilities of the ichthyosaur and to link this behaviour to the anatomical and morphological features revealed in fossilized flippers, specifically the serrated trailing edge, chordwise parallel skin ornamentations, and the protracted distal tip functioning as flexible winglets. The numerical simulations serve as the primary tool to validate the aerodynamic and acoustic implications of these features. Therefore, the fidelity of the computational study is not a peripheral detail but an essential component underpinning the novelty and impact of the manuscript, especially in a high impact, multidisciplinary journal such as *Nature*.

While the authors have stated their intent to target readers primarily from the biology and paleontology communities, I believe this is a narrow view. The potential of this study to inspire cross disciplinary interest, such as among bio-inspired engineers or fluid dynamicists, is significant. For instance, a wind turbine designer could conceivably draw inspiration from these features and seek to apply them to turbine blade designs. In this context, reporting design parameters such as "2 mm fence spacing" in purely dimensional terms is of limited use. Given the large disparity in scale between ichthyosaur flippers and engineered systems, such values must be made dimensionless, for example with respect to the local boundary layer thickness, to facilitate meaningful comparisons. This was a minor request, easily achievable using the existing LES database, and would improve the paper's rigor.

I do not believe that issuing a point-by-point rebuttal to every response would be productive or fair, especially given the extended duration of the review process has already taken. Instead, I would like to suggest a single decisive action from the authors. They should either:

(1) Provide a clear and consistent rationale regarding the type of aeroacoustics noise they aim to suppress. The authors indicate in multiple responses that they focus on turbulent broadband noise:

"...we instead focused on the broadband spectrum, and whether... have any impact on the emitted broadband noise."
"...as our focus is on noise in the broadband spectrum..."

However, the simulation conditions used in the study (that is, Reynolds number around 300000, low inflow turbulence, and low angle of attack) do not support this as a dominant noise mechanism. Under such conditions, the primary noise source is more likely to be laminar instability tonal noise, a well-documented phenomenon in airfoil acoustics. For instance, Arcondoulis et al. (2010, *Acoustics Australia*, 38, 129–133) provide clear evidence of this, including relevant spectral envelopes (such as in Figure 6), which, although based on a NACA0012 profile, remain broadly applicable to NACA0018.

(For context, the laminar instability tonal noise can be eliminated very easily, such as by applying simple surface roughness to trigger bypass transition and suppress the Tollmien-Schlichting wave. It certainly does not require elaborate ancient animal features to do so).

or

(2) Remove the aeroacoustics analysis section entirely and reshape the manuscript to focus more exclusively on the biological and morphological features. This would align the paper more clearly with the intended audience and ensure that the conclusions remain robust within the relevant disciplinary scope.

In summary, I appreciate the interdisciplinary ambition of the manuscript and commend the authors' efforts to explore a novel biological hypothesis. However, given the current ambiguity surrounding the noise mechanisms and the pivotal role of the simulations in supporting the central claims, further clarification or restructuring is necessary.

We greatly appreciate the constructive comments and useful suggestions made by Referees #1–#5. Our point-by-point responses to each individual referee are listed below. Overall, we have accommodated their recommendations wherever possible, and believe that these improve both the presentation and quality of our text and figures for the standard expected by *Nature*.

Please do not hesitate to contact me if you have any queries, or require further information.

Yours sincerely,

Johan Lindgren

Responses to Referee #1

With great interest I read about the amazing, fossilised ichthyosaur flipper from the Early Jurassic described by Johan Lindgren and colleagues.

The level of detail that the fossil preserves is outstanding, as are the numerous analytical approaches the authors conducted to study it. The performance of these all seems to be well conducted. Title and abstract are informative and cover the necessary details of the study. In general, the manuscript is devoted to the soft tissue anatomy of the front flipper and its components, especially what the author's consider to be novel integumentary structure, so-called chondroderms. These descriptions are well embedded into the functional analysis of the flipper, discussing noise-cancelling properties which are proposed to enable a stealthier approach of these large Jurassic predators. For most parts the manuscript is well written and the images accompanying the text are more mostly adequate to support the interpretations.

We thank Referee #1 for their very positive comments and validation of our rigorous analyses, original data, and significance of the results.

Besides pointing out some spelling mistakes, I do have some more critical points that I would like to raise here for consideration by the authors.

Images of the fossil in Fig. 1 and in the supplement Figures 1 and 2: As the forelimb is encompassing slab and counterslab I tried to overlap images using a imaging program – unfortunately I had troubles doing so, because I encountered some distortion in the photographs. I tried to put these together in a tiff file, also to underscore my comments on the length of the flipper and the convergence typically seen in temnodontosaurs.

The photographs taken under polarised and ultraviolet light (Fig. 1 and Extended Data Figs. 1, 2) were meant to provide the reader with high-quality images of the unique soft tissues, which appear less distinct under conventional light. As described in our initial methods section (lines 590–604), this was accomplished by digitally merging multiple high-resolution photographs into a single image using Adobe Photoshop. Unfortunately, this procedure introduced some optical distortion to the final images. We certainly did not intend to misrepresent our fossil, and therefore thank Referee #1 for pointing this out to us. We have now rectified the distortion using Adobe Photoshop, and provide revised figures (see also comparison below).

Comparison between images based on a, a single overview photograph, and b, c, multiple merged photographs taken under polarised light (a, b, slab; c, counterslab). The horizontal lines are intended to highlight the corrected distortion. Note that the composite images match the overview photograph. Also note the aligned details of the slab and counterslab.

p. 2 line 33: “the giant Jurassic ichthyosaur” – Given the size range of ichthyosaurs in general with some taxa reaching >20 m body length, I think it would be better to speak of a large-sized ichthyosaur. later in the text the authors themselves refer to it as e.g. “large-sized temnodontosaurid [...]”. Most Temnodontosaurus reached sizes of modern Orcas (or maybe a bit longer), also considered to be large but not giant.

Even though *Temnodontosaurus* has been repeatedly referred to as a giant ichthyosaur in the published literature (e.g., Godefroit, 1995; McGowan, 1996; Nilsson *et al.*, 2012), we agree with Referee #1 that ‘large-sized/large-bodied’ would be a better description, and hence have replaced ‘giant’ with ‘large’ and ‘large-bodied’ in our revised manuscript (e.g., lines 1 and 33).

References

- Godefroit, P. Biodiversite des reptiles marins du Jurassique inferieur belgo-luxembourgeois. *Bull. Soc. belge Géol.* **104**, 67–76 (1995).
- McGowan, C. Giant ichthyosaurs of the Early Jurassic. *Can. J. Earth Sci.* **33**, 1011–1021 (1996).
- Nilsson, D.-E., Warrant, E. J., Johnsen, S., Hanlon, R. & Shashar, N. A unique advantage for giant eyes in giant squid. *Curr. Biol.* **22**, 683–688 (2012).

p.2 line 50: “gradually attaining more fish-like bauplans” – even earliest ichthyosaurs had eel-like bodies, and as eels are also fish, this expression is not really correct and should be more specific

We have replaced ‘gradually attaining more fish-like bauplans’ with ‘gradually attaining more streamlined profiles’ to comply (line 50).

p.3 line 54: “both basal and derived” – is basal used in terms of “early branching forms”?

Yes; this has been changed in our revised manuscript (line 53).

p.3 line 56 “small to moderate-sized species” – I understand what is meant here but I still like to advise caution here because as the authors know, size is not always a good predictor of trophic level

We thank Referee #1 for their input, and have replaced ‘small to moderate-sized’ with ‘smaller-bodied, piscivorous and teuthophagous’ in our revised manuscript (lines 55, 56).

p.3 line 75: “The metre-long forelimb” – this is vague. Indeed exact measurements are missing from the text and the supplement so the reader is left wondering how large the flipper really is – given from the provided images, the flipper is ca. 80 cm as preserved. For comparison, SMNS 50000 shown in the supplement is ca. 120 cm, based only on the preserved bones and the soft tissue paddle would be even longer. As such it is difficult for the reader to assess the overall size and maybe also the ontogenetic stage of the fossil. I therefore suggest to add measurements and add more infos to the discussion.

We would like to note that the initial version of our Supplementary Information includes some measurements of the specimen’s length, both skeletal ones and with soft tissues (lines 163–171). However, we appreciate that additional measurements would be helpful, and consequently have elaborated further on these in the revised version of our Supplementary Information (lines 77–79 and 183–185). We hope this clarifies any confusion regarding the size of our fossil flipper.

p.3 line 76: if identification of the digits is based on Motani 1999 [currently ref 27] this paper should be referenced here already. In Motani 1999, the temnodontosaur paddle usually encompasses digits II-IV + accessory digit, and not II-V as stated in line 75.

We thank the referee for their insightful comment, and have added Motani (1999) to our revised manuscript (line 75). As we discuss in the initial version of our Supplementary Information (lines 137–141), Motani (1999) actually identified the digits in *Temnodontosaurus* as digit II–V. He specifically states (Motani, 1999, p. 37): “Digit V... of *Temnodontosaurus* (Fig. 5O...) shows extremely truncated development. This digit may be better developed in other specimens of *Temnodontosaurus* (Fig. 4), but it seems to ossify from the middle elements toward the more proximal and distal ones...”. We have expanded upon this in the revised version of our Supplementary Information (lines 138–148) to explain that there is some uncertainty with regards to the identification of a digit V versus a prominent postaxial accessory in *Temnodontosaurus* (see, e.g., Maisch & Matzke, 2000; Swaby & Lomax, 2020).

References

- Maisch, M. W. & Matzke, A. T. The Ichthyosauria. *Stuttgarter Beiträge zur Naturkunde, Serie B (Geologie und Paläontologie)* **298**, 1–159 (2000).
- Motani, R. On the evolution and homologies of ichthyopterygian forefins. *J. Vert. Paleont.* **19**, 28–41 (1999).
- Swaby, E. J. & Lomax, D. R. A revision of *Temnodontosaurus crassimanus* (Reptilia: Ichthyosauria) from the Lower Jurassic (Toarcian) of Whitby, Yorkshire, UK. *Hist. Biol.* **33**, 2715–2731 (2020).

p.4 line 85: “can be confidently identified as a forefin of the large-sized temnodontosaurid

Temnodontosaurus trigonodon” - Although I agree with the assessment that this forelimb likely belongs to *Temnodontosaurus* and not for example to *Eurhinosaurus*, another long-flipped form from the Early Jurassic, I found the assignment to be far from confident. All the temnodontosaur forelimbs I personally saw show distally converging digits II-IV and also the accessory digit. SSN8DOR11 on the other hand lacks this convergence instead presenting almost parallel extending digits.

We appreciate the concern regarding the taxonomic identification; however, based on our detailed comparisons with multiple *Temnodontosaurus* skeletons, we are confident in our referral of SSN8DOR11 to *T. trigonodon*. As discussed in our Supplementary Information (lines 114–119 in the initial submission and lines 114–119 in the revised version), there are at least seven distinct characters that our forelimb shares with recognised front flippers of this species.

Regarding the distally converging digits: although most specimens of *T. trigonodon* show some degree of adduction, considerable variation on this theme exists; some individuals have closely spaced fingers, whereas others exhibit digits that are set more widely apart (see panel a–d in the figure below). This discrepancy could be due to ontogenetic differences; however, given that similarly merged digits occasionally can be seen also in other ichthyosaur taxa—and especially in extremities devoid of soft tissues (compare panel e, f below)—we are more inclined to consider this phenomenon to be the result of taphonomic processes acting on the decomposing limbs. Adducting digits is further at odds with the finger configuration of extant aquatic tetrapods, which all show parallel to splayed out poses (panel g–j below).

While resting on the seafloor, decaying organic matter can adhere to the underlying substrate, thereby maintaining skeletal fidelity by limiting current-induced mobilisation of the bones on the downward-directed side of vertebrate carcasses (Orr *et al.*, 2016). Moreover, a taphonomic scenario that is broadly comparable to that we envisage for parvipelvic flippers has been previously proposed for extinct aquatic turtles (see Joyce *et al.*, 2021; Augustin *et al.*, 2023). In these fossils, the digits often are adducted and additionally lack adhering soft-tissue traces (panel k, l), a preservational mode thought to reflect an absence of stiffening scaly skin when the animals were alive (Joyce *et al.*, 2021; Augustin *et al.*, 2023). Notably, all hitherto documented *Temnodontosaurus* flippers are represented solely by skeletal remains; thus, it is plausible that SSN8DOR11 is the first fossil in which the digits remain in life position. A discussion on this matter has been added to our revised Supplementary Information (lines 149–168).

[REDACTION]

Forelimbs of select aquatic amniotes. a–d, Temnodontosaurus, specimen (a) SMNS 52340, (b) UMH-0020, (c) SMNS 50004, and (d) UMH-0021. e, f, Stenopterygius, specimen (e) SMNS 14846 and (f) GPIT-PV-30017. g–j, Radiographs of the flippers of (g) Phocarcos, (h) Delphinus, (i) Chelonia (all from DeBlois & Motani, 2019, fig. 1a–c), and (j) Carettochelys (from Delfino et al., 2010, fig. 3a). k, Forelimb of the Jurassic pan-cryptodiran Solnhofia (detail from Augustin et al., 2023, fig. 5b). l, Forelimb of the Eocene carettochelyid Allaeochelys (detail from Joyce et al., 2021, fig. 7a). Note that the photographs in panel a, c and d are reversed to facilitate comparisons. Scale bars, 10 cm (a–d), 3 cm (e), 1 cm (f, k, l).

References

- Augustin, F. J. et al. A new specimen of *Solnhofia parsonsi* from the Upper Jurassic (Kimmeridgian) Plattenkalk deposits of Painten (Bavaria, Germany) and comments on the relationship between limb taphonomy and habitat ecology in fossil turtles. *PLoS ONE* **18**, e0287936 (2023).
- DeBlois, M. C. & Motani, R. Flipper bone distribution reveals flexible trailing edge in underwater flying marine tetrapods. *J. Morph.* **280**, 908–924 (2019).
- Delfino, M., Fritz, U. & Sánchez-Villagra, M. R. Evolutionary and developmental aspects of phalangeal formula variation in pig-nose and soft-shelled turtles (Carettochelyidae and Trionychidae). *Org. Divers. Evol.* **10**, 69–79 (2010).
- Joyce, W. G., Mäuser, M. & Evers, S. W. Two turtles with soft tissue preservation from the platy limestones of Germany provide evidence for marine flipper adaptations in Late Jurassic thalassochelydians. *PLoS ONE* **16**, e0252355 (2021).
- Orr, P. J. et al. “Stick ‘n’ peel”: explaining unusual patterns of disarticulation and loss of completeness in fossil vertebrates. *Palaeogeog., Palaeoclim., Palaeoecol.* **457**, 380–388 (2016).

Furthermore, digits III and IV should be of similar length to digit II and the accessory one – in the new fossil, digits II and the accessory digit are clearly longer and digits III and IV appear truncated. I strongly suggest that the authors add to the discussion as it might well be that this specimen was affected by an early stage developmental aberration or even a pathology. This

could in turn affect the non-convergence of digits and affect the shape and aspect ratio of the whole flipper.

Based on the published literature and our personal examination of multiple *T. trigonodon* specimens, this interpretation is not entirely correct. Rather, in complete flippers of *T. trigonodon* (see panel b, d in the figure above), digit II is always longer than digit IV and as long or longer than digit III. When complete, digit V is the longest of all digits or at least of the same length as digit II. Thus, we find it highly unlikely that the digits in SSN8DOR11 represent either an early stage developmental aberration or a pathology. To aid with interpretation, we have elaborated further on this in our revised Supplementary Information (lines 120–137) and included a new figure in our revised manuscript (Extended Data Fig. 3b).

p.4 line 88: please consider adding the locality of Dotternhausen which also offered many ichthyosaur fossils over the past decades

While the locality of Dotternhausen certainly has produced ichthyosaur fossils (including partial remains of *Temnodontosaurus*), there are, to the best of our knowledge, no complete and articulated skeletons of *T. trigonodon* known from that area. Hence, we have retained the sentence as originally written.

p.6, line 134: “occur at the bottom” – does this refer to the external-most layer then?

No; we have rewritten this sentence to clarify that we are referring to the topographically innermost (in terms of the animal) portion of the preserved soft-tissue sequence (line 134).

p.6, line 142: “epidermal–dermal interface and superficial dermis” – is the superficial dermis not part of the epidermal–dermal interface?

We have adopted the terminology of Lindgren *et al.* (2018), which in turn builds on that of Landmann (1986); that is, the epidermal–dermal interface is considered to be broadly consistent with the basement membrane/basal lamina (and its immediate surroundings) of extant tetrapods, while the superficial dermis (stratum spongiosum) represents the melanophore-rich outer portion of the dermis (corresponding to the lower half of the image below).

Stained light microscopy section through the skin of a juvenile leatherback turtle, Dermochelys coriacea. The epidermal–dermal interface corresponds to the basement membrane (Bm) and its immediate surroundings, whereas the superficial dermis (De) covers the entire lower half of the micrograph. Note presence of melanophores (Mel) in both the epidermis (Ep) and dermis (image modified from Lindgren et al., 2018, extended data fig. 9a).

References

Landmann, L. in *Biology of the Integument* (eds Bereitner-Hahn, J. et al.) 150–187 (Springer,

Berlin, 1986).

Lindgren, J. et al. Soft-tissue evidence for homeothermy and crypsis in a Jurassic ichthyosaur. *Nature* **564**, 359–365 (2018).

p.6, line 151 “integumental skeletal organs” – I would not consider them to be organs themselves but part of the organ “integument”

To the best of our knowledge, an organ is a collection of tissues that structurally forms a functional unit specialised to perform a particular task (see, e.g., ‘or-ga-num’, 2020). This can be a heart, lung or an eye, but the border to an organ system or an organ part is blurry. The brain can be considered an organ, but can also be divided into parts that are organs themselves; e.g., a tiny part of the brain—the pineal gland—is typically referred to as the pineal organ (e.g., Falcón *et al.*, 2006).

The integument has many functions and is often considered to be an organ system (e.g., Pawlina, 2024). Parts of the integument that are specialised to perform a particular task would thus fit well into the definition of an organ. However, all this being said, we agree with the referee that it would be even better to refer to osteoderms as ‘integumental skeletal structures’, and accordingly have done so in the revised version of our manuscript (line 150).

References

Falcón, J., Besseau, L. & Boeuf, G. Molecular and cellular regulation of pineal organ responses. *Fish Phys.* **25**, 243–306 (2006).

‘or-ga-num’ in *Dorland’s Illustrated Medical Dictionary*, 33rd edition (Elsevier, Philadelphia, 2020).

Pawlina, W. *Histology: a Text and Atlas: with Corrected Cell and Molecular Biology* (Wolters Kluwer Health, Philadelphia, 2024).

p.7, line 162: please explain a bit more what is meant by “territorial matrix”

In cartilage, isogenous groups (clusters) of chondrocytes are generally surrounded by a territorial matrix, which in turn is embedded in a wider interterritorial matrix (e.g., Clarke, 1971, 1974; Poole *et al.*, 1982, fig. 2); see also image below from Pawlina (2024, fig. 7.7). To clarify the terminology, we have cited Clarke (1974) in our revised manuscript (line 161).

[REDACTION]

References

Clarke, I. C. Articular cartilage: a review and scanning electron microscope study. I. The interterritorial fibrillar architecture. *J. Bone Joint Surg.* **53B**, 732–750 (1971).

Clarke, I. C. Articular cartilage: a review and scanning electron microscope study. II. The territorial fibrillar architecture. *J. Anat.* **118**, 261–280 (1974).

Pawlina, W. *Histology: a Text and Atlas: with Corrected Cell and Molecular Biology*

(Wolters Kluwer Health, Philadelphia, 2024).

Poole, A. R., Pidoux, I., Reiner, A. & Rosenberg, L. An immunoelectron microscope study of the organization of proteoglycan monomer, link protein, and collagen in the matrix of articular cartilage. *J. Cell Biol.* **93**, 921–937 (1982).

p.7, line 167: “fossilised tissue” – does this refer to the globules?

Yes; it refers to the globules and associated extracellular matrix.

p.7, line 171-172: “does not involve any cartilaginous precursor” – Scheyer 2007 (*J Anat Vol 211*) discussed the potential occurrence of cartilage in osteoderms (“postcranial fibro-cartilaginous bone” in osteoderms that are usually considered purely dermal in nature and lacking cartilage) of placodont reptiles. Please consider adding this to the discussion.

We already discussed this issue in the initial version of our paper (see our original Supplementary Information, lines 196–200), but thank Referee #1 for indicating a point of clarification. Although Scheyer (2007) reported the presence of ‘fibro-cartilaginous bone’ in the dermal armour of some placodonts, the chondrocyte lacunae-like microstructures have subsequently been re-interpreted as voids left behind by decayed large-diameter collagen fibre bundles (see Vickaryous & Sire, 2009). In the revised Supplementary Information (lines 225–230), we have further emphasised that the placodont plate tissue is both structurally and histologically different from the globular calcified cartilage we identify in *Temnodontosaurus* (e.g., it lacks well-defined isogenous cell groups, Liesegang banding patterns, and morphologically distinct territorial and interterritorial matrices).

References

Scheyer, T. M. Skeletal histology of the dermal armor of Placodontia: the occurrence of ‘postcranial fibro-cartilaginous bone’ and its developmental implications. *J. Anat.* **211**, 737–753 (2007).

Vickaryous, M. K. & Sire, J.-Y. The integumentary skeleton of tetrapods: origin, evolution, and development. *J. Anat.* **214**, 441–464 (2009).

p.9, line 208: “allowing a stealthy approach” – orcas are considered to often use stealthy approach tactics when hunting marine mammals, but their forelimbs lack any serrations. Would it make sense to add a bit more to the discussion here?

We thank the referee for providing an interesting and thought-provoking reflection. Predatory animals that either search for or pursue prey use various means and strategies to avoid being detected (see lines 189–194 in the initial version of our paper), and orcas are no exception (see our revised Supplementary Information, lines 610–622). These whales live in social groups, and like most other extant odontocetes, communicate through whistling and pulsing calls (e.g., Thomsen *et al.*, 2002). However, when hunting for marine mammals with a good sense of underwater hearing, they are known to go silent (e.g., Riesch & Deecke, 2011), a condition some researchers (e.g., Reeves *et al.*, 2006) consider to be a ‘stealth’ mode. Nonetheless, although highly efficient predators in their own right, killer whales apparently are not as specialised for life in scotopic (light-poor) conditions as was *Temnodontosaurus*. Not only does this large-bodied parvipelvic have the largest eyeballs of any vertebrate (living or extinct), but it also possesses flow control devices to attenuate trailing edge self-noise, to suggest that it was uniquely suited for life in dim-lit pelagic environments (and thus without any directly comparable modern analogue, save perhaps for owls on land). Because

massive eyes are costly to both build and maintain, we hypothesise that an evolutionary arms race between predator and prey resulted in the exceptional visual and stealth adaptations seen in *Temnodontosaurus*. Moreover, we also propose that silent swimming emerged as a means to avoid being detected by prey equipped with coevolving hearing and/or tactile senses (see lines 674–683 in our revised Supplementary Information).

References

- Reeves, R. R., Berger, J. & Clapham, P. J. in *Whales, Whaling, and Ocean Ecosystems* (eds Estes, J. A. et al.) 174–187 (Univ. California Press, Berkeley, 2006).
- Riesch, R. & Deecke, V. B. Whistle communication in mammal-eating killer whales (*Orcinus orca*): further evidence for acoustic divergence between ecotypes. *Behav. Ecol. Sociobiol.* **65**, 1377–1387 (2011).
- Thomsen, F., Franck, D. & Ford, J. K. B. On the communicative significance of whistles in wild killer whales (*Orcinus orca*). *Naturwissenschaften* **89**, 404–407 (2002).

p.9 line 209 “SSN8DOR11 has a substantially higher aspect ratio” – these data need to appear somewhere in the main text or the supplement – especially of long-finned forms such as *Eurhinosaurus* from Europe or *Guizhouichthyosaurus* from China.

As requested, we have added a section with aspect ratios to our revised Supplementary Information (lines 251–266). However, because neither *Eurhinosaurus* nor *Guizhouichthyosaurus* are known from any soft-tissue specimen (save for SMNS 81842, in which some taphonomic distortion of the middle–distal phalanges and surrounding skin envelope cannot be excluded; compare Extended Data Fig. 8c and the image compilation on p. 5 of this document), it has not been possible to calculate reliable ratios for the flippers of these taxa. So, while we agree in principle with the referee, these numbers unfortunately cannot be given pending the discovery of articulated limbs with undisturbed soft-tissue outlines.

p.9 lines 215-216: “permitted limited mobility in life” – was this ever tested? McGowan and Motani is a great volume but I am not sure whether this is the best reference for this sentence here.

McGowan & Motani (2003, p. 35) provide a general description of the forefin of Early Jurassic ichthyosaurs, noting that the closely packed skeletal elements in ‘a tightly interlocking mosaic’ make ‘for a rigid structure’. Although they do not specifically discuss *Temnodontosaurus*, the same reasoning should be applicable also to this taxon given the similarly dense arrangement of its sub-rectangular to polygonal carpals, metacarpals and proximal phalanges (which additionally often are broader than long). However, to better reflect the uncertainty of our inference, this part of the sentence now reads ‘...presumably only permitted limited mobility in life’ (lines 214, 215).

Reference

- McGowan, C. & Motani, R. in *Handbook of Paleoherpetology* (ed. Sues, H.-D.) Vol. 8, 175 pp. (Verlag Dr. Friedrich Pfeil, Munich, 2003).

p.9, line 219: “fin ray-like elements” – in my opinion this needs better documentation. Could you provide an image of these structures in the real fossil?

Fin ray-like elements in ichthyosaurs have been previously documented by, e.g., Owen (1841) and Riess (1986), but also illustrated by, e.g., Motani (1999, fig. 6a, b, 2005, fig. 5—bottom left panel) and DeBlois & Motani (2019, fig. 7c, d). However, as requested, we have added a photograph (panel ‘a’) of the left hindlimb of *Stenopterygius quadriscissus* specimen GPIT-PV-30017 to our revised Extended Data Figure 8.

References

- DeBlois, M. C. & Motani, R. Flipper bone distribution reveals flexible trailing edge in underwater flying marine tetrapods. *J. Morph.* **280**, 908–924 (2019).
- Motani, R. On the evolution and homologies of ichthyopterygians forefins. *J. Vert. Paleontol.* **19**, 28–41 (1999).
- Motani, R. Evolution of fish-shaped reptiles (Reptilia: Ichthyopterygia) in their physical environments and constraints. *Annu. Rev. Earth Planet. Sci.* **33**, 395–420 (2005).
- Owen, R. XIX—A description of some of the soft parts, with the integument, of the hind-fin of the *Ichthyosaurus*, indicating the shape of the fin when recent. *Trans. Geol. Soc. Lond.* **6**, 199–201 (1841).
- Riess, J. Fortbewegungsweise, schwimmbiophysik und phylogenie der ichthyosaurier. *Palaeontographica Abt. A Paläozool. Stratigr.* **192**, 93–155 (1986).

p.11, line 261: why was the living Minke whale chosen here?

As explained in our initial Supplementary Information (lines 306–311), we consider the Minke whale, *Balaenoptera acutorostrata*, to be a reasonable modern analogue because its flippers are elongate and somewhat ‘wing-like’, yet dorsoventrally rather narrow, and thus broadly comparable to those of *Temnodontosaurus* (assuming the existence of a relationship between phalanx thickness and overall fin depth in *Temnodontosaurus* that was not completely overprinted by the encasing connective tissues).

p.13, line 311 “extant reptiles (ref. 43[...])” – is ref 43 not dealing exclusively with terrestrial taxa? Why was no aquatic or amphibious species added here (e.g. crocs with their dome pressure receptors)?

Because we have a limited number of references at our disposal (50), we opted to use a paper that includes micrographs that are directly comparable to those published by Lindgren *et al.* (2018, fig. 2i, j); that is, they are obtained using the same technique (SEM), show a similar view, and additionally depict sensory organs that morphologically resemble the integumentary structures preserved in MH 432 (*Stenopterygius* sp.).

Reference

- Lindgren, J. et al. Soft-tissue evidence for homeothermy and crypsis in a Jurassic ichthyosaur. *Nature* **564**, 359–365 (2018).

Ref. 24 and 25: please check names “Fish, F. E.” and “Fish, E. F.”

The name has been corrected in our revised manuscript (line 388).

Figure legend Fig. 1: “proxo-posterior” – do you mean “proximo-posterior” here?

We realise that ‘proxo-posterior’ can be confusing in this caption, and accordingly have omitted it in our revised manuscript (see also our response below).

Figure legend Fig. 1: “sediment that has been digitally removed” – unclear to me what exactly is meant here – does this refer to the CT scan data?

No; this refers to photographs of the fossil taken under polarised and ultraviolet light. SSN8DOR11 was retrieved as a three-dimensional jigsaw puzzle of odd-sized rock slabs, collectively forming opposing part and counterpart sections of a flipper that has been cleaved along the sagittal plane. When reassembled into their original, multi-layered configuration, one of the larger blocks (‘rock slab A’ in the figure below) partially obscures bones exposed in an underlying, smaller piece of rock (‘rock slab B’ below). Hence, in order to depict the full complement of skeletal elements, we opted to digitally remove (using Adobe Photoshop) part of the overlying limestone in the images taken under polarised and ultraviolet light (unfortunately, the surrounding sedimentary matrix cannot be mechanically prepped away without compromising the structural integrity of the fossil).

Digital removal of sedimentary matrix in photographs of SSN8DOR11. a, Reassembled proximal end of SSN8DOR11 (slab and counterslab) prior to preparation. b, Section through the unprepared slab and counterslab. The fossil surface is marked by yellow arrowheads. Note overlapping ‘rock slab A’ and ‘rock slab B’ (arrow). c, Proximal end of SSN8DOR11 during the preparation of the fossil. The red box frames grey limestone (of ‘rock slab A’) that obscures the underlying ‘rock slab B’ containing part of the fossil. d, Proximal end of the flipper after preparation (slab section). Bones and adhering soft tissues revealed by digital removal of the above-lying sediment are marked with a red box (compare with c).

Data availability: I would like to encourage to make the CT scan data openly available to support open data movement in Science.

We appreciate the encouragement by Referee #1, and will make the CT-data openly available upon publication of our paper.

Responses to Referee #2

The paper examines new information on the morphology of the pectoral flipper from a fossil ichthyosaur as it related to hydroacoustic performance. The paper presents a description of an elongation of the flipper and the possession of periodic extensions from the trailing edge. The first half of the text is devoted to a description of the fossil, including a description of the morphology include soft tissue. Although I am not a paleontologist with an understanding of the nuances of taphonomy, these sections appear to be complete and well-written to indicate the strength of the description of the fossil to support the morphological findings.

We thank Referee #2 for their positive comments, and can confirm that our integrated experimental approach with multiple cross-referencing molecular and imaging methodologies was constructed with the utmost rigour.

What is far more interesting is the interpretation of trailing edge structures (serrations) on the flipper. The authors make the case by analogy and use of CFD to state that the trailing edge serrations are to suppress hydrodynamically radiated noise. Such an effect would be useful for a nocturnal marine predator that is attempting to sneak up on potential prey without disclosing its presence. Although mentioned but not adequately interpreted are chordwise-parallel surface ornamentations and the elongation and flexibility of flippers. Arguments and support for these morphological structures could be further strengthened and elaborated.

We thank Referee #2 for pointing out portions of the text that could be further elaborated. In the revised version of our Supplementary Information, we have included a new section entitled ‘Function of the flippers in *Temnodontosaurus*’ (lines 608–654). This segment deals, among other things, with various functional aspects of surface ornamentations and elongate flippers.

There is a large literature base on trailing edge serrations from engineering indicating significant noise suppression and drag reduction. Although the authors cite works by Fish, Wolfe, and Fish & Lauder (22-24) examining noise reduction, there are some additional studies about trailing edge serrations, surface geometries, and finlets that would further lend support to the main hypothesis. These include:

Smith, T.A. and Klettner, C.A., 2022. Airfoil trailing-edge noise and drag reduction at a moderate Reynolds number using wavy geometries. *Physics of Fluids*, 34(11).

Hu, Y.S., Zhang, P.J.Y., Wan, Z.H., Liu, N.S., Sun, D.J. and Lu, X.Y., 2022. Effects of trailing-edge serration shape on airfoil noise reduction with zero incidence angle. *Physics of Fluids*, 34(10).

Ananthan, V.B. and Akkermans, R.A., 2023. Trailing edge noise reduction using bio-inspired finlets. *Journal of Sound and Vibration*, 549, 117553.

Clark, I.A., Alexander, W.N., Devenport, W., Glegg, S., Jaworski, J.W., Daly, C. and Peake, N., 2017. Bioinspired trailing-edge noise control. *AIAA Journal*, 55(3), 740-754.

We thank Referee #2 for recommending these additional publications [please note that Clark *et al.*, 2017 is already cited (ref. 29) in our initial submission]. Because of space limitations, the above listed papers are cited in our revised Supplementary Information.

The use of possible noise reduction from biology comes from the crenulations of the trailing edge of the flukes of the humpback whale and owl wings, although the original reports demonstrating noise suppression on owls was based on the leading edge rather than the trailing edge. These early works are cited in:

Blick, E. F., Watson, D., Belie, G. and Chu, H. 1975. Bird aerodynamic experiments. In: *Swimming and Flying in Nature*, Vol. 2 (T. Y-T. Wu, C. J. Brokaw and C. Brennen, eds.), pp. 939-952, Plenum Press, New York.

We thank the referee for this information while noting that subsequent studies (e.g., Lilley, 1998; Bachmann *et al.*, 2007; Sagar *et al.*, 2017) also have considered contributions from the trailing edge fringe.

References

- Bachmann, T. *et al.* Morphometric characterisation of wing feathers of the barn owl *Tyto alba pratincola* and the pigeon *Columba livia*. *Front. Zool.* **4**, 1–15 (2007).
- Lilley, G. M. A study of the silent flight of the owl. Paper presented at the 4th AIAA/CEAS Aeroacoustics Conference, Toulouse, France, *AIAA Pap.* 1998-2340 (1998).
- Sagar, P., Teotia, P., Sahlot, A. D. & Thakur, H. C. An analysis of silent flight of owl. *Mat. Today Proc.* **4**, 8571–8575 (2017).

However for a cautionary note related to owl feathers and whale flukes, owls wings and whale flukes oscillate, whereas the flippers of the ichthyosaur are passive control surfaces that are used to stabilize the body when transiting and are used for maneuverability. It would be expected that oscillating structures generate noise while passive structures would generate substantially less noise unless there is boundary layer separation and stall promoting turbulence. The angle of attack used in the CFD analysis were well below the angle for stall. The chordwise-parallel surface ornamentation could potentially change the stall characteristic of the flipper as well as enhance drag reduction. There is only brief consideration beyond noise suppression of how the chordwise-parallel surface ornamentation and the elongation of flippers would play into drag reduction and increased maneuverability. Both would be important in foraging. Although the argument is made that the large ichthyosaur must use stealth to catch the smaller elusive prey, whereas maneuverability as well as speed and other behaviors can close the gap between predator and prey. Indeed, the enhanced lift properties of the flipper would be an advantage for increased maneuverability rather than facilitating “calm gliding motions”. Enhanced lift for gliding would only be necessary if the ichthyosaur was negatively or even positively buoyant. Consideration of these drag reduction and maneuvering performance attributes should be more completely discussed.

Because we primarily focussed on the noise production during sustained swimming (cruising), the investigated angles of attack (α) were intentionally chosen to avoid stall conditions. However, the referee is correct that at least some of the passive flow control devices we identify in SSN8DOR11 potentially could have served functions other than noise suppression, including drag reduction and/or delay of stall. Consequently, we performed complementary simulations to evaluate the impact of trailing edge serrations and surface ornamentations at $15^\circ \alpha$ (that is, near-stall conditions based on the almost identical NACA 0018 profile, see panel ‘a’ and ‘b’ below). Our findings are exemplified by a comparison between a serrated geometry with a smooth surface and one that is covered by chordwise ridges (see also our revised Fig. 5b, c and Extended Data Fig. 9h). Notably, the ridged geometry has a higher drag coefficient and lower lift coefficient than the smooth flipper section at $15^\circ \alpha$ (see our revised Fig. 5c), resulting in a hydrodynamic performance penalty (that is, increased drag). Moreover, as shown in the ‘c’ panel below, the velocity fluctuation fields are much the same when comparing a smooth fin geometry without serrations (top), a geometry with a smooth surface and trailing edge serrations (centre), and a ridged geometry with trailing edge serrations (bottom). This similarity likely stems from the diminutive proportions of the surface ornamentations and their location behind the leading edge (see panel ‘d’ below), an arrangement that is markedly different from the distinct, protruding leading edge tubercles seen on the flippers of the agile humpback whale, *Megaptera novaeangliae* (van Nierop *et al.*, 2008; Fish *et al.*, 2011). Our data thus indicate that the

ridges/troughs in SSN8DOR11 likely were too small to have any impact on the stall characteristics of the forefins in *Temnodontosaurus* (see our revised Supplementary Information, lines 646–654).

Similarly, albeit being elongate, the flippers in *Temnodontosaurus* were not as enlarged as in the humpback whale (14–17% versus 25–33% of body length), but may instead have been more comparable to the pectoral fins of certain extant pelagic sharks that occasionally are known to drift with ocean currents (see Moss, 1984; Compagno, 2001).

a, Comparison between a NACA 0018 profile and the geometry used in our work (which is based on a Minke whale flipper cross-section). **b**, Lift coefficient as a function of α for a NACA 0018 profile at various Reynolds numbers (from <http://airfoiltools.com/airfoil/details?airfoil=naca0018-il>). **c**, Velocity fluctuation magnitudes for a flipper geometry with a smooth surface and lacking trailing edge serrations (top), one with serrations but without surface ornamentations (centre), and one equipped with both chordwise ridges and serrations (bottom). **d**, Smooth leading edge of SSN8DOR11 (UV-image) together with a short section of our digital model (drawn to scale). Scale bar, 10 mm.

Finally, concerning ichthyosaur buoyancy: it remains debated whether these ancient marine reptiles were positively or negatively buoyant (see, e.g., Reisdorf *et al.*, 2012, 2014; van Loon, 2013), although the bulk of the evidence seems to favour a density higher than that of seawater (see Reisdorf *et al.*, 2012 and references therein). Because an enlarged lifting area allows for swimming at a slower speed in a negatively buoyant animal (Magnuson, 1973; Moss, 1984), our hypothesis where the elongate flippers of *Temnodontosaurus* primarily functioned to provide lift at slow cruising speeds is thus entirely feasible (and consistent with the behaviour of equally long-finned pelagic sharks; e.g., Moss, 1984; Compagno, 2001).

References

- Compagno, L. J. V. *Sharks of the World. An Annotated and Illustrated Catalogue of Shark Species Known to Date. Volume 2. Bullhead, Mackerel and Carpet Sharks (Heterodontiformes, Lamniformes and Orectolobiformes)*. FAO Special Catalogue for Fishery Purposes. No. 1, Vol. 2 (FAO, Rome, 2001).
- Fish, F. E., Weber, P. W., Murray, M. M. & Howle, L. E. The tubercles on humpback whales' flippers: application of bio-inspired technology. *Int. Comp. Biol.* **51**, 203–213 (2011).

- Magnuson, J. J. Comparative study of adaptations for continuous swimming and hydrostatic equilibrium of scombroid and xiphooid fishes. *Fish Bull.* **71**, 337–356 (1973).
- Moss, S. A. *Sharks—an Introduction for the Amateur Naturalist* (Prentice-Hall, Inc., Englewood Cliffs, 1984).
- Reisdorf, A. G. et al. Float, explode or sink: postmortem fate of lung-breathing marine vertebrates. *Palaeobio. Palaeoenv.* **92**, 67–81 (2012).
- Reisdorf, A. G. et al. Reply to “Ichthyosaur embryos outside the mother body: not due to carcass explosion but to carcass implosion” by van Loon (2013). *Palaeobio. Palaeoenv.* **94**, 487–494 (2014).
- van Loon, A. J. Ichthyosaur embryos outside the mother body: not due to carcass explosion but to carcass implosion. *Palaeobio. Palaeoenv.* **93**, 103–109 (2013).
- van Nierop, E. A., Alben, S. & Brenner, M. P. How bumps on whale flippers delay stall: an aerodynamic model. *Phys. Rev. Lett.* **100**, 054502 (2008).

There is no evidence that Temnodontosaurus was a “vigorous ambush hunter”. These characterizations, vigorous and ambush, seem to be opposites. Ambush predators are typically sit-and-wait predators and can only accelerate over very short distances. Large oceanic predators are generally cruisers and the more massive the animal the more restricted any acceleration. Characterization as being both vigorous and an ambush hunter seems inappropriate.

We agree with the referee that this is an unfortunate wording, and hence have deleted the sentence in the revised version of our manuscript.

Another area to strengthen is the assertions being made in this report regarding the specific acoustic range of the target species and see how the noise generated by the flipper is reduced but out of the target’s range. Hearing in fishes are generally confined to the low frequency range of 800 to 1000 Hz. Although salmon have a range of 40-350, cod have a range of 20-38,000 and some clupeids have a range of 200-180,000 Hz. See the references below:

- Popper, A.N. and Coombs, S., 1980. Acoustic detection by fishes. In *Environmental physiology of fishes* (pp. 403-430). Boston, MA: Springer US.
- Ladich, F., 2013. Effects of noise on sound detection and acoustic communication in fishes. In *Animal communication and noise* (pp. 65-90). Berlin, Heidelberg: Springer Berlin Heidelberg.
- Popper, A.N., Hawkins, A.D., Sand, O. and Sisneros, J.A., 2019. Examining the hearing abilities of fishes. *The Journal of the Acoustical Society of America*, 146(2), pp.948-955.
- Putland, R.L., Montgomery, J.C. and Radford, C.A., 2019. Ecology of fish hearing. *Journal of Fish Biology*, 95(1), pp.39-52.
- Fay, R.R., 2014. The sense of hearing in fishes. In *Perspectives on auditory research* (pp. 107-123). New York, NY: Springer New York.

While perhaps not restricted to 800–1000 Hz, it is nonetheless true that, depending on species, modern fishes have different hearing ranges. In general, however, they are all sensitive to low-frequency sounds (e.g., Kasumyan, 2005, table 3; Yan *et al.*, 2010; McQueen *et al.*, 2019; Sertlek *et al.*, 2019). According to, e.g., McQueen *et al.* (2019, p. 11, 12): “Fish can detect

frequencies ranging between 30 to 1,000 Hz (Erbe 2011), and some fish can even detect infrasound (<20 Hz; e.g., Clupeid spp.) and ultrasound (>20,000 Hz; e.g., Atlantic herring; Normandeau Associates 2012). More commonly the 100 to 400 Hz frequencies are detected by a majority of fish studied (e.g., see Offutt 1974; Yan 2001; Codarin et al., 2009; Parmentier et al. 2011)... Fish appear to be particularly well adapted to detecting lower frequency sounds (<1,000 Hz)...”.

Although the time-step used in our computations (7.5e-6 sec) theoretically allows us to explore frequencies that are several orders of magnitude higher than those discussed in our paper, we opted to focus on the low-frequency range because, in water, the damping of acoustic waves increases with frequency (see, e.g., Tyack & Janik, 2013, table 9.1). As a consequence, lower frequency sounds propagate over longer distances, and are thus more important in pelagic environments (e.g., Ladich, 2013; Tyack & Janik, 2013). In Figure 5d, we depict an expanded frequency interval (up to 1 kHz), which shows the same noise attenuation trend by trailing edge serrations as seen below 200 Hz (albeit at lower amplitudes). This demonstrates that we do not lose any significant information by our choice of frequency range. Moreover, from a numerical point of view, higher frequencies tend to be contaminated by errors; hence, focusing on lower frequencies provides more reliable data.

It is finally worth pointing out that both preserved gastric contents and regurgitates associated with or attributed to *Temnodontosaurus* indicate a diet consisting primarily of other ichthyosaurs and coleoid cephalopods (Böttcher, 1989; Serafini & Maxwell, 2023); that is, groups that are considered to be (or have been) sensitive to low-frequency sounds (e.g., Packard *et al.*, 1990; Hetherington, 2008; Mooney *et al.*, 2010).

References

- Böttcher, R. Über die Nahrung eines *Leptopterygius* (Ichthyosauria, Reptilia) aus dem süddeutschen Posidonienschiefer (Unterer Jura) mit Bemerkungen über den Magen der Ichthyosaurier. *Stuttgarter Beitr. Naturk. B* **155**, 1–19 (1989).
- Hetherington, T. in *Sensory Evolution on the Threshold: Adaptations in Secondarily Aquatic Vertebrates* (eds Thewissen, J. G. M. & Nummela, S.) 183–209 (Univ. California Press, Berkeley, 2008).
- Kasumyan, A. O. Structure and function of the auditory system in fishes. *J. Ichthyol.* **45**, S223–S270 (2005).
- Ladich, F. in *Animal Communication and Noise* (ed. Brumm, H.) 65–90 (Springer-Verlag, Berlin Heidelberg, 2013).
- McQueen, A. D., Suedel, B. C. & Wilkens, J. L. Review of the adverse biological effects of dredging-induced underwater sounds. *WEDA J. Dredging* **17**, 1–22 (2019).
- Mooney, T. A. et al. Sound detection by the longfin squid (*Loligo pealeii*) studied with auditory evoked potentials: sensitivity to low-frequency particle motion and not pressure. *J. Exp. Biol.* **213**, 3748–3759 (2010).
- Packard, A., Karlsen, H. E. & Sand, O. Low frequency hearing in cephalopods. *J. Comp. Physiol. A* **166**, 501–505 (1990).
- Serafini, G. & Maxwell, E. E. in *8th International Meeting on Mesozoic Fishes and Aquatic Tetrapods: Bio-crisis and Recoveries* (eds Maxwell, E. E. et al.) 54 (Abstract Volume, Stuttgart, 2023).
- Sertlek, H. Ö., Slabbekoorn, H., ten Cate, C. & Ainslie, M. A. Source specific sound mapping: spatial, temporal and spectral distribution of sound in the Dutch North Sea. *Environ. Poll.* **247**, 1143–1157 (2019).
- Tyack, P. L. & Janik, V. M. in *Animal Communication and Noise* (ed. Brumm, H.) 251–271 (Springer-Verlag, Berlin Heidelberg, 2013).
- Yan, H. Y., Anraku, K. & Babaran, R. P. in *Behavior of Marine Fishes: Capture Processes*

and Conservation Challenges (ed. He, P.) 45–64 (Wiley-Blackwell, Ames, 2010).

Lastly, the very last sentence (331-334) seems to be an overreach. I might be nice to see a morphological design be integrated in modern technologies, but unless the geometry is truly novel and has nothing similar, then engineers will not be utilizing this design for biomimetic applications.

We have modified the last sentence to comply (see lines 326, 327 in our revised manuscript).

In general, the paper is very good with some exciting ideas, but needs to be strengthened with references relating to trailing edge hydrodynamics and acoustics for flipper elongation, trail edge geometry, fin flexibility, fish hearing, and consideration of maneuverability.

Frank E. Fish

We thank Referee #2 for their expert opinion, and have added the requested information to our revised manuscript.

Responses to Referee #3

GENERAL

The discovery of a large, apparently well-preserved flipper of an ichthyosaur allows some inferences on the acoustic and hydrodynamic properties of the small structural properties thus revealed. There are 2 sets of geometric details – the presence of a serrated trailing edge, and the presence of small, regular streamwise structures that could be troughs or ridges. The possible influence was estimated in companion computations. The reconstruction and computations are the two key elements of the work.

It is the structure of many Nature articles to bury much detail in supplemental information, but the exercise of extracting it back out here was quite time-consuming. The computations are very important in this manuscript, and though certain details would be of interest to only a few, they are required for any evaluation of the computational models. There are some suggestions below for what details would be valuable.

Works based on fossil records are necessarily somewhat speculative, but as far as I can judge, those presented here are worth considering.

We thank Referee #3 for their supportive comments, and have accommodated their suggestions wherever possible (see also our responses below).

DETAILS

Fig. 5b: Here, straight lines are used to join 2 dots, from 0 to 5 degrees angle of attack. Though we might expect this relationship for Cl, we would not for Cd. In either case, this is too much interpolation.

We apologise for the unfortunate choice of line graphs; our purpose was not to interpolate the data in between the dots. The line graphs have been replaced by bar plots in our revised Figure 5b.

Fig. 5c: Similar comment but here the line graph implies some continuous change on the abscissa, but the three points are actually categories. A bar graph would be better. There are no uncertainties on any of these data points, partly because they have appeared somewhat magically from a CFD package. But there should be some estimate of the sensitivity of C_l and C_d to assumptions and suppositions in the computation. Perhaps some variance could be obtained at least from the time history of the force coefficients.

As requested, we have replaced the line graphs with bar plots, and also included the standard deviation of the force coefficients (see our revised Fig. 5c).

P10, 1241: Self-noise occurs mostly at low angles of attack and comes from shedding at the trailing edge. It is not a sure thing that such conditions are found for this postulated wing/fin. However, if we estimate the chord Reynolds number (it is never given) for a 20 cm chord (from Fig. 1) it is about 3×10^5 , and at this Re self-noise can occur at a wide range of α – it is actually the regime for the most likely tonal noise (see, for example, Probsting *et al.*, *J. Fluid Mech.*, 2014, Fig. 4). To estimate effects of geometry variation on the coherent structures formed over the wing, we would be interested in details close to the surface and on scales similar to the geometry variations (ridge height, spacing). These are not given.

The Reynolds number (300,000) is now included in our revised Supplementary Information (lines 418, 419). The range of Reynolds numbers where tonal noise (self-noise) is emitted by a hydrofoil does indeed depend on the angle of attack (α); however, aero/hydrofoil thickness (Geyer & Moreau, 2021) and shape (Yakhina *et al.*, 2020) also contribute. Notably, the geometry investigated by Probsting *et al.* (2014) is a NACA 0012, whereas the profile we use is based on the flipper cross-section of a Minke whale (which is very similar to NACA 0018; see panel ‘a’ below). The thicker hydrofoil (compared to NACA 0012) and deviation from an ideal NACA 0018 profile (starting at about $x = -25$ mm in the ‘a’ panel below) facilitate the development of turbulent vortices along the surface or our geometry. These speed up the breakdown of the von Karman vortices generated at the trailing edge, thereby removing the primary source of tonal noise. Moreover, the results in Geyer & Moreau (2021) show that there is a distinct difference in Reynolds number range between a NACA 0012 and NACA 0018 when it comes to the generation of tonal noise. Therefore, it is not possible to accurately predict the tonal noise range for a thicker aero/hydrofoil based on the results from a NACA 0012 profile (the most commonly used standardised aerofoil shape).

Nevertheless, to test if our numerical approach is capable of capturing tonal noise, we set up a simulation that corresponds to the $0^\circ \alpha$ case by Nakano *et al.* (2007) using a NACA 0018 aerofoil; that is, the closest profile to our model with documented tonal noise. Our simulation did indeed predict the appearance of tonal noise, and even if the dominant frequencies were slightly shifted relative to those of Nakano *et al.* (2007)*, the order of magnitude of the noise levels matched Nakano *et al.*’s (2007) experimental observations (see panel ‘b’ below; note that the experimental spectrum, EXP, was manually digitised from figure 4 in Nakano *et al.*, 2007).

*The frequency of tonal noise depends on the size of the shed vortices, which in turn depends on the location of the shedding. Moreover, the separation along an aero/hydrofoil is sensitive to small perturbations, and therefore the precise frequency of the resulting tonal noise is difficult to predict.

a, Comparison between a NACA 0018 profile and the geometry used in our work (which is based on a Minke whale flipper cross-section). *b*, blue: experimentally measured sound pressure levels for a NACA 0018 aerofoil at $0^\circ \alpha$ (manually digitised from fig. 4 in Nakano et al., 2007); green: sound pressure levels predicted for the same case using the approach employed in our manuscript.

Regarding surface details: in the figure below, we depict the velocity fluctuation magnitudes for our $0^\circ \alpha$ cases at 50% of the chord (the images are zoomed in at the vicinity of the surface). Note the increased velocity fluctuation levels caused by surface treatments (ridges and troughs), which speed up the breakdown of large-scale vortices.

Average velocity fluctuation magnitudes visualised in the vicinity of the flipper surface for a case with a smooth surface (top), one with ridges ('ribs'; centre), and one with troughs (bottom).

References

- Geyer, T. F. & Moreau, D. J. A study of the effect of airfoil thickness on the tonal noise generation on finite, wall-mounted airfoils. *Aerosp. Sci. Technol.* **115**, 106768 (2021).
- Nakano, T., Fujisawa, N., Oguma, Y., Takagi, Y. & Lee, S. Experimental study on flow and noise characteristics of NACA0018 airfoil. *J. Wind Eng. Ind. Aerodyn.* **95**, 511–531 (2007).
- Pröbsting, S., Serpieri, J. & Scarano, F. Experimental investigation of airfoil tonal noise generation. *J. Fluid Mech.* **747**, 656–687 (2014).
- Yakhina, G., Roger, M., Moreau, S., Nguyen, L. & Golubev, V. Experimental and analytical investigation of the tonal trailing-edge tonal noise radiated by low Reynolds number aerofoils. *Acoustics* **2**, 293–239 (2020).

P11, CFD: The CFD is LES, with some unspecified subgrid-scale model. The operating Re is not given, and neither is the Mach number, M. How are acoustic fluctuations resolved in this model? The span of the two-dimensional model is said to match the flipper length, but the model is a two dimensional slab and not a three-dimensional geometry, so the ‘flipper length’ is not relevant. It would be helpful to know the domain size in chords. It looks as though the span is about $0.2c$, when we are not sure that possible vortex modes are limited by the span. The run times are given in seconds but what is needed is how many convection times Ut/c . Similarly, frequencies should also be noted in Strouhal number, $St = fc/U$.

Since no alterations were made to the SGS and acoustic models (WALE and Curle, respectively), they were only mentioned in the ‘Methods’ section of our initial submission (lines 701–703). Both the Reynolds (Re) and Mach (M) number are now provided as part of our revised Supplementary Information (lines 418, 419). The domain size in terms of the chord ($30 \times 14 \times 0.48 c$) has further been added to our revised Supplementary Information (lines 409, 410). Note that longitudinal vortices on a hydrofoil originate from spanwise instabilities. At low Re (below 5,000), the wavelengths of the dominant instabilities can indeed be quite large (see, e.g., Gupta *et al.*, 2023); however, at the Re considered in our study, the longitudinal vortices should be smaller than $0.1 c$ (cf. Zhenyao *et al.*, 2019). Furthermore, Zhang & Samtaney (2016) demonstrated that there are only minor differences in aerodynamic performance and magnitude of the velocity fluctuations between aerofoils with 0.4 and $0.8 c$ span. Therefore, we are confident that our selected span ($0.48 c$) is sufficient to capture all relevant spanwise instabilities in both the wake and boundary layer.

We would finally like to add that the computed physical times are now also given in terms of convection times (Supplementary Information, lines 440, 441). Likewise, frequencies are provided in terms of the Strouhal number (St) throughout the revised text (e.g., main text, lines 272 and 274).

References

- Gupta, S. et al. Two- and three-dimensional wake transitions of a NACA0012 airfoil. *J. Fluid Mech.* **954**, A26 (2023).
- Zhenyao, L., Lihao, F., Karbasian, H. R., Jinjun, W. & Kim, K. C. Experimental and numerical investigation of three-dimensional vortex structures of a pitching airfoil at a transitional Reynolds number. *Chin. J. Aeronaut.* **32**, 2254–2266 (2019).
- Zhang, W. & Samtaney, R. Assessment of spanwise domain size effect on the transitional flow past an airfoil. *Comput. Fluids* **124**, 39–53 (2016).

Fig. 5def: The acoustic spectra are compared for various conditions, but we have no sense of

the variations/sensitivity/uncertainty of SPL at any given frequency. There is a difference of 2 dB. Is that a big number? Is it significant? There is some information and some interesting results here, but it is difficult to extract it from a combination of manuscript and extended material.

According to evolutionary theory (e.g., Futuyma, 1998), any adaptation, big or small, that provides a selective advantage—and thereby positively affects an organism’s biological fitness—is of relevance. For instance, a wing with a backward swept leading edge and straight trailing edge provides a 4.5% increase in induced efficiency compared to a wing with an elliptical planform (van Dam, 1987); a seemingly low number, but still enough to be evolutionary beneficial.

We would also like to emphasise that the experimentally obtained numbers should *not* be taken at face value; instead, they are meant to serve as support for the general principle of noise suppression by trailing edge serrations and surface treatments, and that this concept is applicable also to a geometry based on SSN8DOR11. Nonetheless, all included cases have been computed and post-processed in exactly the same manner; therefore, we are confident that the observed trends are correctly predicted. Furthermore, the time to develop the flow (22.5 convection times) and to collect the statistics and record the noise data (67.5 convection times) is long enough to minimise any statistical error.

Post-processing of the recorded data can have an impact on the resulting spectra. Wider windows affect the lowest captured frequencies, allow narrower peaks, but also leave more numerical noise in the data since fewer individual spectra are available for statistics for the same amount of recorded sound. Conversely, narrower windows result in smoother spectra but the lowest resolved frequencies increase. In our study, we use Hanning windows with a width of 65536 points and 50% overlap to improve the clarity of the spectra. Below we show a comparison using 65536 and 131072 points as window width for the noise recorded 5 m upstream of the flipper geometry for the three different surfaces considered in our work (smooth, with ridges and with troughs, respectively). Although the absolute level of the sound pressure spectra is slightly shifted when the post-processing window is doubled, the distance between the spectra obtained for the different surfaces remains roughly the same. This demonstrates that the post-processing does not significantly affect predictions pertaining to the influence of surface ornamentations.

Sound pressure levels predicted for cases using a smooth flipper (black and grey), one with ridges (light and dark blue), and one with troughs (light and dark green), respectively. The post-processing is based on 65536 (black, dark blue and dark green) and 131072 point wide (grey, light blue and light green) Hanning windows.

Reference

Futuyma, D. J. *Evolutionary Biology* (Sinauer Associates, Inc., Sunderland, 1998).
van Dam, C. P. Efficiency characteristics of crescent-shaped wings and caudal fins.

Nature **325**, 435–437 (1987).

P10, ridges: Fig. 2a,b – the ridges are indeed remarkably regular. Their spacing is estimated in mm. Given likely Re , then how does the spacing compare with a likely boundary layer thickness?

We used one of the mean vorticity-based methods described by Griffin *et al.* (2021), which defines the boundary layer thickness as the location where the generalised velocity reaches 99% of its freestream value. Taking the $0^\circ \alpha$ case with troughs as an example, the boundary layer thickness at the maximum hydrofoil thickness is $\delta = 3.2$ mm. Thus, the trough depth (0.8 mm) is approximately 25% (0.25 δ) of the thickness of the boundary layer, whereas the trough spacing (2 mm) is 0.625 δ .

Lin (2002) and Lengani *et al.* (2011) used low-profile vortex generators with heights down to 0.2 δ and found these to be highly effective in controlling boundary layer separation. Even though their vortex generators were of another type than those examined by us, they are nonetheless in the same order of magnitude as our surface ornamentations. This demonstrates that the ridges and troughs have a height/depth and spacing that are large enough to have an impact on the breakdown of large-scale vortices.

References

- Griffin, K. P., Fu, L. & Moin, P. General method for determining the boundary layer thickness in nonequilibrium flows. *Phys. Rev. Fluids* **6**, 024608 (2021).
- Lengani, D., Simoni, D., Ubaldi, M., Zunino, P. & Bertini, F. Turbulent boundary layer separation control and loss evaluation of low profile vortex generators. *Exp. Therm. Fluid Sci.* **35**, 1505–1513 (2011).
- Lin, J. C. Review of research on low-profile vortex generators to control boundary-layer separation. *Progr. Aerosp. Sci.* **38**, 389–420 (2002).

Extended, somewhere: The virtual microphone is said to be 250 m upstream. How is this possible given the size of the computational domain, which extends forward only 2 m? Presumably a model is used for sound attenuation in sea water, but it would be helpful to have that given explicitly. Lengths should be rescaled by c . Some remarks are given on sound attenuation and some examples, scaled to the postulated animal size could be helpful. What are SPL $1c$ in front of the fin? How about $5c$ which looks to be about the location of the forward tip of the mouth? There are data for 5 m and for 250 m (!!) in front of the model, but it would be informative to have these lengths related to likely body sizes.

Since acoustic fluctuations scale with the square of the Mach number (M), and M in our case is low (0.001), direct computation of the acoustic waves by solving a compressible set of Navier–Stokes equations is practically impossible. Therefore, we opted to use a hybrid approach (the Curle analogy) to predict the acoustic waves. This method involves two stages: first, the time-dependent flow and acoustic source(s) are determined by solving the incompressible Navier–Stokes equations around (in our case) the flipper. Then, the acoustic pressure fluctuations are computed at desired virtual microphone positions based on the pressure fluctuations on the fin surface (for details; see Curle, 1955). Since the acoustic pressure wave propagation is not explicitly computed but instead analytically integrated, the position of the virtual microphones can be outside of the domain used for the flow computations. In this manner, acoustic predictions can be done with reasonable computing resources even for low M , and when the acoustic pressure needs to be investigated over long distances from the source.

A potential shortcoming of the Curle analogy is that the acoustic waves are assumed to propagate freely towards the microphone; hence, the impact of any obstacle is unaccounted for (importantly, though, this is not an issue for us). Another possible limitation is that the damping by the surrounding sea water is not accounted for, which means that the attenuation of acoustic waves by increasing distance is due only to volumetric effects. Since the damping by water increases with frequency (Tyack & Janik, 2013), any error related to this restraint inevitably will increase as well. Nevertheless, we focused on the lowest part of the frequency spectrum, and therefore we are confident that the trends observed are correctly captured.

Another implication of the analytical integration is that with extended distance, the acoustic source (in our case, the flipper surface) behaves more and more like a point source, and therefore the spectra recorded at different distances are qualitatively similar (as noted above, the wave amplitudes diminish only due to volumetric effects). The acoustic integration part is computationally inexpensive compared to the solution of the incompressible Navier–Stokes equations; therefore, we placed a relatively large number of virtual microphones in the domain. The largest distance (250 m) was included as an order of magnitude distance where we presume it would be important for a potential prey to be able to detect a hunting predator.

In the revised version of our Supplementary Information, we have included the microphone distance (250 m) normalised against body length (line 466). Moreover, when describing our experimental set-up, we have also added the domain size in terms of chord length (see our revised Supplementary Information, lines 409, 410).

Reference

Curle, N. The influence of solid boundaries upon aerodynamic sound. *Proc. R. Soc. Lond. A* **231**, 505–514 (1955).

Tyack, P. L. & Janik, V. M. in *Animal Communication and Noise* (ed. Brumm, H.) 251–271 (Springer-Verlag, Berlin Heidelberg, 2013).

P11: “Although approximations and simplifying assumptions are necessary when modelling multiscale fluid flows and their interactions with complex biological structures, such analyses nonetheless provide valuable information about the core function(s) of specific features.” Not sure this sentence says something useful.

This sentence has been omitted in the revised version of our manuscript.

P11, l262: More details of this hydrofoil shape would be useful. It is symmetric, and looks to be quite thick. At very least the thickness should be given and then chord location of maximum thickness. Is there a close NACA equivalent – I’m guessing the 0018.

The reviewer is correct about the NACA similarity; in fact, the two profiles are almost identical (see image on p. 19). The maximum thickness of our digital model is 0.182 c at 0.305 c. The main difference between the two geometries is that the Minke whale-based one is slightly thinner than the NACA profile at about the centre of the chord. This narrower shape likely leads to flow separation further upstream relative to the NACA 0018 profile.

P14: “Due to their ability to propagate fast and far underwater”. How far, how fast?

We have simplified the sentence by omitting this subordinate clause (line 320).

RECOMMENDATION

There are three main concerns above. The first is that certain computational details need to be

given, and preferably in the main text. The second is that some effort should be made to identify uncertainties in the force coefficients and acoustic SPL. The third is that the small-scale detail required to understand the hydrodynamic influence of the ridges and serration are not given; instead, we see from afar, the overall flow fields, or time-averaged fluctuations (extended fig. 9, 10).

We are aware that the numerical details were rather concise in our initial submission. However, we have included additional information in our revised Supplementary Information to comply (lines 361–441), and addressed all of the above listed uncertainties (see our previous responses). Please note though that error bars are difficult to add to results obtained from computational fluid dynamics simulations. There are many sources that potentially could introduce errors (such as mesh resolution, discretisation schemes, solution methods, and post-processing), and quantification of the error magnitude is therefore often not possible. To minimise numerical errors, we carried out a set of precursor simulations to check (among other things) grid sensitivity, sensitivity to domain size, and sensitivity to adopted turbulence model. We have added error bars to the force coefficient plots in our revised Figure 5b, c, which indicate the standard deviation in time. However, comparable error bars cannot be added to the SPL plots because these are based on the spectra of the acoustic pressure fluctuations. Nonetheless, we did check the sensitivity of the predicted noise spectra against the adopted post-processing window width (see discussion on p. 21), and we have now also compared our results against experimental data obtained for a similar case (see our response above, p. 18, 19).

It is easy for a reviewer to recommend further details in their speciality, and less easy to accommodate these suggestions in a compact form required by Nature. I am biased in my interest but it seems to me that some space in the reconstruction images and story could be saved and that the computations need to have more attention and precision so that some details buried in supplemental data, and others not yet presented could be given. In a short manuscript, providing complete details of either reconstruction or computation is probably impossible, but the focus could be on reconstruction details of direct relevance to the computations. Fig. 1, 2a,b, 4, 5 are all essential. Then some flow details on the scale of the ridges and serrations might be quite informative.

We appreciate that the referee acknowledges our dilemma when it comes to balancing information of interest to researchers from different fields and disciplines. Notably, our manuscript is intended as a biological sciences paper, and while we have tried to accommodate Referee #3's recommendations wherever possible (mainly in the revised Supplementary Information), the targeted audience primarily comprises biologists and palaeontologists (although we anticipate that the far-reaching implications of our results will be of interest also to a considerably wider group of scholars, including hydrodynamics and acoustics engineers).

Noting this point, the speculations about the flexible tip and its influence are not supported by any of the computations here, and are probably unnecessary.

We respectfully disagree, and instead view the described flipper adaptations as an interconnected biological system, where each part works together as a functional whole. Hence, we have retained our inference of a flexible tip in the revised manuscript.

The manuscript has potential that would be better realized with a less tentative approach to numerical details.

We thank the reviewer for their positive assessment, and have tried to accommodate their recommendations wherever possible (please also see our responses above).

Responses to Referee #4

This article concerns a bio-inspired marine feature, albeit one that has already extinct more than a hundred millions years ago, for effective flow control and low noise radiation characteristics. After a thorough investigation of the sample (e.g. SSN8DOR11), a large aspect ratio flipper has been identified that contains multiscale elements and combination of eidonomic and anatomical features in several categories, such as the streamwise surface ornamentations in a form of finlets, and a serrated trailing edge (mostly at the outer-board region of the flipper) strengthened by chondroderms. The combination of the non-smooth surface texture and corrugated edge within an aerofoil mock-up were computationally studied using the freeware OpenForm, where a Large Eddy Simulation method was used. The results seem to indicate that the trailing edge serration can reduce self-noise from the mid-frequency range. If the trailing edge serration is combined with the surface texture in finlet, further noise reduction at low frequency can be realised. The results enable the authors to make an important statement in the article that the ancient ichthyosaur can catch their prey owing to the evolutionary low noise feature of their flippers.

I must declare that my field of study is neither the bioscience, nor the palaeontology, although I must also mention that I was fascinated and have enjoyed reading the first seven pages of the articles that focus on the description of the fossil features, and cartilaginous integumentary structures, including their figures and supplementary information.

We are appreciative of Referee #4's constructive assessment, and thank them for endorsing both the significance and quality of our results.

My peer review of this article probably starts from Page 8, when the narratives touch upon the subjects of functional implications, CFD and the analysis of the simulated results. I have also read the supplementary information especially Parts G, H and I.

My first comment is related to the trailing edge serration size relative to the aerofoil overall chord length. From the supplementary document, the serration has an amplitude of 4 mm. If I understood correctly, it scales the last 25% of the aerofoil chord. Does this mean that the overall aerofoil chord length is 16 mm, which seems to be too small to model the flipper properly? I might have missed something but I think this point needs better clarification. From the same line of enquiry, the amplitude-to-wavelength ratio of the serration you have chosen is 0.5, which is not necessary the most optimum configuration. Usually, a “sharper” serration like the one we observe from the trailing edge fringe of owl is more effective for self-noise reduction. In any case, I can accept that the nature of morphology might not only have low-noise characteristic in mind (e.g. a smaller aspect ratio for the serration might be better from the structural integrity point of view).

We thank the referee for pointing out that our initial explanation was not self-contained; this has been corrected in our revised Supplementary Information (lines 397–404). Although the

amplitude is only 4 mm, the serrations nonetheless affect the hydrofoil shape over the last 25% of the chord (which corresponds to 5 cm), resulting in a sinusoidal thickness variation in the spanwise direction (a necessity to achieve an even transition from the gently curved anterior surface to serrated trailing edge).

Regarding the amplitude-to-wavelength ratio: as correctly indicated by Referee #4, our intention is to mimic the fossil structures rather than to optimise their effect.

Second, the state of the simulated flow needs better clarification. What is the Reynolds number of the flow, especially at the trailing edge? If the Reynolds number is low, a natural transition to fully turbulent boundary layer at the trailing edge is unlikely. In this particular case, the self-noise generated at the trailing edge might have a distinct tonal characteristic that follows a particular frequency scaling law. On the other hand, for a relatively high Reynolds number, the radiated noise will be more broadband. The vortices depicted by the λ^2 criterion are for instantaneous snapshots, but they all seem quite intermittent to me, where a statistical fully-developed turbulent boundary layer at the trailing edge might not be a given. The presence of separation bubble at the suction side at 5 degree AoA represents a clue that the boundary layer at most of the surface is not yet turbulent.

The Reynolds number (300,000) is now included in our revised Supplementary Information (lines 418, 419). Regarding tonal noise: to test if our numerical approach is capable of capturing tonal noise, we set up a simulation that corresponds to the $0^\circ \alpha$ case by Nakano *et al.* (2007) using a NACA 0018 aerofoil; that is, the closest profile to our model with documented tonal noise. Our simulation did indeed predict the appearance of tonal noise, and even if the dominant frequencies were slightly shifted relative to those of Nakano *et al.* (2007)*, the order of magnitude of the noise levels matched Nakano *et al.*'s (2007) experimental observations (see image below; note that the experimental spectrum, EXP, was manually digitised from figure 4 in Nakano *et al.*, 2007).

*The frequency of tonal noise depends on the size of the shed vortices, which in turn depends on the location of the shedding. Moreover, the separation along an aero/hydrofoil is sensitive to small perturbations, and therefore the precise frequency of the resulting tonal noise is difficult to predict.

[REDACTION]

Blue: experimentally measured sound pressure levels for a NACA 0018 aerofoil at $0^\circ \alpha$ (manually digitised from fig. 4 in Nakano et al., 2007); green: sound pressure levels predicted for the same case using the approach employed in our manuscript.

Reference

Nakano, T., Fujisawa, N., Oguma, Y., Takagi, Y. & Lee, S. Experimental study on flow and noise characteristics of NACA0018 airfoil. *J. Wind Eng. Ind. Aerodyn.* **95**, 511–531 (2007).

I must also comment that the authors only correlate the turbulent velocity fluctuations to the radiated noise, but the key noise source is actually the underlying surface pressure fluctuations. Some evidences of the reduction of the surface pressure fluctuations, especially for the streamwise surface ornamentations case, could be provided to reinforce the claims made in this article.

Our rationale for showing velocity fluctuations is that streamwise surface ornamentations generate vortical structures, thereby introducing velocity fluctuations. Hence, it is more straightforward to visualise noise generation using velocity rather than pressure fluctuations. Also, our flow computations solve an incompressible set of Navier–Stokes equations; therefore, the velocity and pressure fluctuations are strongly connected.

Regarding the impact of surface ornamentations on velocity fluctuations: in the figure below, we depict the velocity fluctuation magnitude for our $0^\circ \alpha$ cases at 50% of the chord (the images are zoomed in at the vicinity of the surface). Note the increased velocity fluctuation levels caused by surface treatments (ridges and troughs), which speed up the breakdown of large-scale vortices.

Average velocity fluctuation magnitude visualised in the vicinity of the flipper surface for a case with a smooth surface (top), one with ridges ('ribs'; centre), and one with troughs (bottom).

What is also missing in the article is a validation of the numerical methodology. I would compare some of the noise spectra obtained from the simulation against either the experimental data (from wind tunnel, not from the virtual aerofoil), or Direct Numerical Simulation (DNS) from the literatures so that the readers have the confidence of the predicted noise spectra, which are important evidences to substantiate the main story of the article.

Please note that we have now performed a comparison with experimental data for a similar case (see our response above, p. 26), which importantly demonstrates that we do capture the

noise levels, albeit with a minor frequency shift.

Responses to Referee #5

Lindgren et al for Nature

Dear editor at Nature,

Thank you for the opportunity to review an interesting manuscript, Adaptations for stealth in the wing-like flippers of a giant ichthyosaur, by Lindgren et al.

I accepted to review the paper because I have worked quite a lot on ichthyosaurs, including one paper on soft tissue preservation (using SEM-EDS and XRD), some using petrographic thin sections and some with use of CT data. I have not personally used all the methods that are used for obtaining the results in this paper, and I am not versed in fluid mechanics theory.

Summary

This manuscript by Lindgren et al reports on a well-preserved forefin from the Jurassic ichthyosaur *Temnodontosaurus*. In addition to most of the skeletal elements, the fossil preserves soft tissue of several different types, which shows the shape of the forefin and its structure. It preserves a series of smaller elements along its trailing edge that are suggested to be a new structure, chondroderms. The chondroderms, as well as other soft tissue components, are imaged and analyzed through several methods. This results in a description of components of the fin such as the dermis, melanosomes, its outer structure including ridges, as well as the chondroderms. The fin then possesses a set of characteristics that are hypothesized to be adaptations for reducing the underwater noise created by the animal itself during hunt for prey. These are analysed in a computational fluid dynamics framework in order to test whether this is actually possible. The study finds that the fin design might have reduced the underwater noise, and names this “stealth swimming”.

Overall recommendation:

The forefin is a very nicely preserved fossil and I enjoy the idea behind the paper which I find novel and worth investigating, as well as the extensive methods usage in order to extract a maximum amount of data. I do like the structure built around the four-fold combination of features for stealth (summary from line 202) and think this works well.

We are appreciative of Referee #5’s constructive assessment, and can confirm that our integrated experimental approach was constructed with the utmost rigour.

However, there are some points that I want to raise that should be considered and probably improved before this work is potentially published in Nature.

The full story of the paper (forefin structures including chondroderms were an adaptation to silent swimming) is at the moment interesting and backed by results, but some points weaken the argument, including a lack of discussion of other possibilities for the observed structures and a lack of modern day parallels. I have no reason to doubt the results of the methods that analyze the components of the preserved soft tissue. However, there is a lack of justification for methods used.

We thank Referee #5 for pointing out portions of the text that could be further elaborated, and have added a section dealing with alternative flipper functions and (the general lack of) modern-day parallels—in addition to a rationale for our methodological approach—to comply (see also our responses below). Because of space limitations, this new information has been placed in our revised Supplementary Information.

Main comments

1) Justification of methods

I miss a section justifying the methods chosen, that states what each of the methods contribute with. An impressive array of methods was used, producing interesting images and results on chemistry etc. However, as the article is presently written, it is hard to follow the steps of the experimental procedure. For instance, I would appreciate an explanation on which methods are used for assessing the taphonomy (alterations in chemistry after death) and the soft tissue structures (what is preserved from the original material), respectively, and for instance what the petrographic slides contribute versus what the other methods contribute. This also includes the usage of the three porpoise specimens. Comparing modern day cetaceans with ichthyosaurs is obvious and highly interesting, and it should be clearly explained what this part of the work contributed to and how the samples from porpoises compared to the ichthyosaur.

In accordance with the recommendations given by Lindgren *et al.* (2015a), we apply a methodology that has been developed over more than 10 years, and which combines a suite of sensitive imaging (e.g., SRXTM, FEG-SEM and TEM), elemental (EDX) and molecular (ToF-SIMS and IRIS) techniques. Notably, our integrated experimental approach has previously been employed to determine the structural and chemical composition of an array of multimillion-year-old animal soft-tissue remains, including eyes (Lindgren *et al.*, 2012), skin (Lindgren *et al.*, 2014, 2018; De La Garza *et al.*, 2022, 2023), feathers (Lindgren *et al.*, 2015b), and internal organs (Lindgren *et al.*, 2018). Furthermore, we would like to point out that we do provide some information on how the taphonomy was assessed in the Supplementary Information (see lines 524 and 525 in our initial submission). However, all this being said, we agree with the referee that a justification of the scientific approach would be beneficial, and have therefore added a methodological rationale to our revised Supplementary Information (lines 270–290).

References

- De La Garza, R. G. et al. An ancestral hard-shelled sea turtle with a mosaic of soft skin and scutes. *Sci. Rep.* **12**, 22655 (2022).
- De La Garza, R. G., Sjövall, P., Hauff, R. & Lindgren, J. Preservational modes of some ichthyosaur soft tissues (Reptilia, Ichthyopterygia) from the Jurassic Posidonia Shale of Germany. *Palaeontology* **66**, e12668 (2023).
- Lindgren, J. et al. Molecular preservation of the pigment melanin in fossil melanosomes. *Nat. Commun.* **3**, 824 (2012).
- Lindgren, J. et al. Skin pigmentation provides evidence of convergent melanism in extinct marine reptiles. *Nature* **506**, 484–488 (2014).
- Lindgren, J. et al. Interpreting melanin-based coloration through deep time: a critical review. *Proc. R. Soc. B* **282**, 20150614 (2015a).
- Lindgren, J. et al. Molecular composition and ultrastructure of Jurassic paravian feathers. *Sci. Rep.* **5**, 13520 (2015b).

Lindgren, J. et al. Soft-tissue evidence for homeothermy and crypsis in a Jurassic ichthyosaur. *Nature* **564**, 359–365 (2018).

The justification for using the minke whale flipper shape for the fluid dynamics is good.

We can confirm that the Minke whale was chosen after careful consideration of other alternatives.

2) Description of the soft tissue composition and architecture (lines 89-137)

This part is the one I find the weakest of the manuscript in terms of clarity. As it stands now, it is hard to grasp which methods have been used for detecting what (see above), but also which parts of the preserved soft tissue contributes to the hypothesized noise reduction. I think it can strengthen the argument in the paper if it is made clear why the identification of the different types of soft tissue is important: e.g. is it seldom identified, does it contribute to stealth etc.

We are unclear about which precise aspects of our description are being referred to by this comment. Regardless, we have constructed our manuscript in a logical manner, and our discoveries are dealt with in separate sections with concise subheadings ('Description' and 'Cartilaginous integumentary structures', respectively). Based on the described material, we identify a four-fold combination of eidonomic and anatomical features (mostly soft tissues) not previously seen in any aquatic vertebrate (see 'Functional implications'), and then use computational fluid dynamics simulations to experimentally demonstrate that at least some of these structures likely provided hydroacoustic benefits (see 'Computational fluid dynamics'). Finally, we incorporate a discussion on the implications of our findings for the inferred hunting strategy of *Temnodontosaurus* (see 'Mode of life').

While we present our primary findings in the main text (in accordance with the recommendations for *Nature* contributions), we now also provide a methodological rationale as part of our revised Supplementary Information (lines 270–290). Moreover, as stated in the introduction section (lines 54–62 in our initial submission; lines 54–62 in the revised version), SSN8DOR11 includes the first ever evidence of soft tissues in *Temnodontosaurus* (thereby entitling a detailed description), and devices potentially contributing to stealth are listed under 'Functional implications' (lines 202–208 in our initial submission; lines 201–207 in the revised version).

I also find it hard to picture from the text how the skin was built and want to suggest a figure showing the different layers and cells that are detected. This could potentially be interesting together with e.g. figure 2c.

As suggested, we have added an interpretative line drawing of the preserved tissue layers to our revised manuscript (new Extended Data Fig. 4d).

I also suggest doing a full check on the terminology used, for instance what is meant by "stripes", "ornamentation", "surface treatments" and whether all of these terms are needed or might be overlapping. In the paragraph line 228-241, ridges (also used in caption for figure 5), riblets, finlets, grooves and fences are also mentioned. This also includes the usage of "chordwise", vs perpendicular, parallel etc.

We are not entirely sure how to address this point; some of these words (e.g., ‘stripes’ and ‘ornamentation’) are purely descriptive, others (e.g., ‘chordwise’) are part of the general terminology of, e.g., aero- and hydronautics, but also commonly used in biology and palaeontology (e.g., when dealing with the wings of birds and pterosaurs or flippers of whales and ancient marine reptiles; e.g., Fish & Battle, 1995; Palmer & Dyke, 2012; Wagner *et al.*, 2017; Fish & Lauder, 2017; Gutarra *et al.*, 2022). Furthermore, although some of these expressions potentially could be a bit puzzling for readers unfamiliar in, e.g., hydrodynamics, they are not readily interchangeable (see, e.g., Clark *et al.*, 2017). Nonetheless, we have done our best to simplify the terminology in the revised version of our manuscript (e.g., line 237).

References

- Clark, I. A. et al. Bio-inspired trailing edge noise control. *AIAA J.* **55**, 740–754 (2017).
- Fish, F. E. & Battle, J. M. Hydrodynamic design of the humpback whale flipper. *J. Morph.* **225**, 51–60 (1995).
- Fish, F. E. & Lauder, G. V. Control surfaces of aquatic vertebrates: active and passive design and function. *J. Exp. Biol.* **220**, 4351–4363 (2017).
- Gutarra, S., Stubbs, T. L., Moon, B. C., Palmer, C. & Benton, M. J. Large size in aquatic tetrapods compensates for high drag caused by extreme body proportions. *Commun. Biol.* **5**, 380 (2022).
- Palmer, C. & Dyke, G. Constraints on the wing morphology of pterosaurs. *Proc. R. Soc. B* **279**, 1218–1224 (2012).
- Wagner, H., Weger, M., Klaas, M. & Schröder, W. Features of owl wings that promote silent flight. *Interface Focus* **7**, 20160078 (2017).

In the paragraph lines 228-241, I miss a concluding remark on how the ridges in the ichthyosaur specimen are interpreted – e.g. are they interpreted as “finlets adapted for noise reduction”.

Our functional interpretation is provided already in a preceding paragraph (see lines 202–208 in our initial submission). Moreover, we intentionally end the paragraph with “Certain surface treatments, such as riblets, finlets and grooves [the latter now replaced with ‘troughs’ to reduce the number of terms used in the text], can improve the aero/hydrodynamic and acoustic performance of aero- and hydrofoils^{29,30}. Notably, streamwise finlet ‘fences’ with a spacing of 2 mm (that is, the same distance as the stripes in SSN8DOR11) have been experimentally shown to attenuate trailing edge self-noise³¹.” to indicate the possibility of hydroacoustic adaptations also in our flipper (something that we then demonstrate using computational fluid dynamics simulations).

The study detects pigment cells (eg. Pictured in 2d- g.), and it should be added what their role was. This includes whether they tell anything about coloration of the skin, and how they relate to the stripes and to the chondroderms?

In similarity with virtually all other reptiles, the melanin (which is synthesised and stored in the melanophores) originally must have provided both colouration (for camouflage or display) and UV-protection; it additionally may have functioned in thermoregulation, for metal scavenging etc. (see, e.g., Lindgren *et al.*, 2015, 2018 and references therein). However, we intentionally refrain from speculating about the colouration (apart from mentioning in passing that the skin presumably had a dark hue in life; see line 583 in our revised Supplementary Information) as we find this information to be ancillary, and thus not the main focus of our paper. Furthermore, as stated in the manuscript (lines 110–112 in our initial submission), the

melanophore-rich substrate forms the topographically outermost layer of the preserved tissues (see also Fig. 2c and our new Extended Data Fig. 4d, e). It is directly underlain (in terms of the animal) by a coat of densely packed melanosomes (interpreted as the juxtaposed epidermal–dermal interface and superficial dermis) that contributes to the overall striped appearance of the flipper blade (see lines 122–126 in our initial submission); these accumulated pigment organelles also envelope the chondroderms (see lines 141, 142 in our initial submission), as is evident in Figure 2g.

References

- Lindgren, J. et al. Interpreting melanin-based coloration through deep time: a critical review. *Proc. R. Soc. B* **282**, 20150614 (2015).
- Lindgren, J. et al. Soft-tissue evidence for homeothermy and crypsis in a Jurassic ichthyosaur. *Nature* **564**, 359–365 (2018).

3) Modern day parallels

As the text reads now, the case for noise reducing adaptations in living animals is not fully convincing with regard to the examples that are presented. The humpback whale is a documented example, however with a modified leading edge (as opposed to the trailing edge in the ichthyosaur) and caudal fluke. The Fish 2020 paper mentions the possibility in California sea lions, but this has not been tested. The owl wing example is interesting, but not in every way an ideal parallel for ichthyosaurs. Looking at the literature, I was not able to find additional examples of modern day parallels. This does not mean that noise reducing structures cannot be found, but it should be made clearer that this is not a widely found, well-documented phenomenon in modern animals at the moment. Some of the literature referred do not discuss adaptations in living animals, and are instead about physics, mechanics and design. That is very interesting, and relevant for marine, streamlined animals. It can be made clear where the literature refers to biology and in which cases it is about physical principles that also have to apply to animals.

We thank the referee for their recommendations. It is correct that structures thought to reduce the acoustic signature of living animals are rare, something that could reflect the unusual hunting strategy we envisage for *Temnodontosaurus*. As discussed in our Supplementary Information (lines 617–622), this parvipelvician has the largest eyeballs of any vertebrate (in addition to possessing extraordinary noise-suppression devices), to suggest that it was uniquely suited for life in dim-lit pelagic environments (and thus without any directly comparable modern analogue). Furthermore, even if examples from nature are scarce, there is no shortage of reports dealing with trailing edge serrations and surface treatments on aero- and hydrofoils from physics and engineering points of views. This has all been clarified in our revised Supplementary Information (lines 623–632).

In line 110, “living amniotes” are mentioned as a comparison. I suggest making clear what parts of these observations are made based on the experiments on the porpoise specimens done in this work, versus what is collected from the literature.

We have added the requested information to our new methodological rationale (see our revised Supplementary Information, lines 284–288).

4) Other possibilities for the fin structures

In this article the authors test the forefin structures in the fossil specimen for its possible stealth-traits. That is interesting, and builds on theory that biological structures can inhibit or reduce noise (albeit not with very many modern day parallels, see point 3). However, the possibility that the structures are adaptation for other uses, is not sufficiently discussed. The paper states that osteoderms, which are relatively similar structures, in other animals usually are adaptations for protection (line 242), and that trailing edge serrations have been shown to reduce drag and alleviate stall (line 250).

As requested, we have added a section dealing with possible alternative functions of the forefins to our revised Supplementary Information (lines 608–654).

In the fluid dynamics experiment (where I am no expert), drag and lift were calculated under different scenarios (fig 5b+c, and supplementary from line 278). These results could possibly be reported in the main text, in order to discuss the different possible effects that the fin design has. If this is not relevant, it should be stated that alternative hypotheses are not tested.

We are not entirely sure what the referee refers to; however, in an attempt to comply, we have included a brief discussion on the (minimal) impact on the hydrodynamic performance in our revised Supplementary Information (lines 634–636).

5) Noise generation by an animal under water

In the section on noise production by the animal's own body, I miss a more thorough discussion on what body parts actually make the noise. I especially wonder about the caudal fluke, which was the main propulsive organ in parvipelvic ichthyosaurs. As mentioned in the paper, *Temnodontosaurus* was probably relatively flexible compared to some other taxa, but still a thunniform body (Buchholtz 2001). The supplementary text (line 320) states that other parts of the body were not considered, and thus it is worth mentioning that these other parts probably made noise too. Do the results obtained here predict noise-reducing structures on the tail as well? Line 303 seems to suggest that it swam more with forefins than caudal fluke, which is surprising.

While we do provide some information about noise production by swimming animals in our main text (lines 178–200), we intentionally refrain from speculate about body parts that so far are known solely from skeletal remains. However, to comply, we have added a conservative prediction regarding potential noise suppression devices in the thrust-producing tail fin of *Temnodontosaurus* to our revised Supplementary Information (lines 625–628).

We are not entirely sure what the referee refers to on line 303 of our initial submission. We hypothesise that the elongate front flippers were extended to provide hydrodynamic lift during cruising/steady swimming (as do the pectoral fins of many extant pelagic vertebrates; see, e.g., Magnuson, 1973), while the fluke propelled the animal through the water. Moreover, as stated on lines 288 and 289 in our initial submission, Buchholtz (2001) proposed that the limbs of *Temnodontosaurus* additionally could have generated thrust at slow swimming speeds, and thus may have been functionally similar to the flippers of the extant humpback whale in this regard (see, e.g., Segre *et al.*, 2017).

References

Buchholtz, E. A. Swimming styles in Jurassic ichthyosaurs. *J. Vert. Paleontol.* **21**, 61–73 (2001).

Magnuson, J. J. Comparative study of adaptations for continuous swimming and hydrostatic

equilibrium of scombroid and xiphoid fishes. *Fishery Bull.* **71**, 337–356 (1973).
Segre, P. S., Seakamela, S. M., Meÿer, M. A., Findlay, K. P. & Goldbogen, J. A. A hydrodynamically active flipper-stroke in humpback whales. *Curr. Biol.* **27**, R636–R637 (2017).

Other comments:

- First mentioned at line 247, the “~75% of the span” needs clarification. What does this mean? I am also not convinced (but this might be because I misunderstand the 75% reference) that this corresponds to a similar area or range in a caudal fluke, of which an animal only has one, and which is placed mid-line.

‘Fluke span’ is the distance between the two posterolateral tips of a fluke measured perpendicular to the longitudinal axis (mid-line) of the body (see Woodward *et al.*, 2006, fig. 2). Wolfe (2017, p. 5) states that “...the crenulation height appears to have a peak between 70-75% of the span of a fluke...”. As illustrated in the ‘a’ panel below, 70–75% of the span (centred around the mid-line of the animal) indeed corresponds to the area with prominent crenulations. Even if the fluke is sub-divided into two equal halves (to simulate a flipper), this ratio remains the same; that is, 70–75% of the new span (one half of the fluke) represents the same area as before (see the ‘b’ panel below). Also note that despite the anterior end of a whale being considerably wider than the fluke mid-line, ‘flipper span’ is likewise measured relative to the longitudinal axis of the animal; that is, from the tip to insertion of the flipper with the body.

[REDACTION]

Humpback whale fluke in planform view. a, Wolfe (2017) states that the crenulation height peaks at between 70 (indicated in red) and 75% of the fluke span. b, Even when sub-divided into two equal halves along the mid-line (to simulate a flipper), this ratio remains the same; that is, 70–75% of the new span corresponds to the same area as before (images modified from Wolfe, 2017, fig. 3).

References

- Wolfe, T. M. Review of fluid dynamic and acoustic performance of biologically inspired passive flow control trailing edge devices for design applications. *55th AIAA Aerospace Sciences Meeting, AIAA SciTech Forum*, AIAA 2017-0542 (2017).
- Woodward, B. L., Winn, J. P. & Fish, F. E. Morphological specializations of baleen whales associated with hydrodynamic performance and ecological niche. *J. Morph.* **267**, 1284 – 1294 (2006).

- Line 273; where the noise reduction is given, I miss a number that shows how much this is, i.e. 1-5 dB out of how many dB in total? As this is a core result to the paper, maybe some more text should be moved from supp mat?

The requested information is already provided in Figure 5d–f. However, we would like to emphasise that the experimentally obtained numbers should *not* be taken at face value;

instead, they serve as support for the general principle of noise suppression by trailing edge serrations and surface treatments, and that this concept is applicable also to a geometry based on SSN8DOR11.

- Line 322 coevolutionary arms race. This inference should be expanded and explained a bit more.

The requested information is already provided in the Supplementary Information (lines 532–559 in the initial submission and lines 656–683 in the revised version).

Figures

- Which methods are used for imaging should be consistently mentioned in all captions

Not necessarily; please see the captions in, e.g., Lindgren *et al.* (2018). We have a limited number of words at our disposal, and therefore need to focus on the primary message that each image is meant to convey (also, the requested information is provided in the ‘Statistics and reproducibility’ section). However, we have added methods wherever possible to comply (e.g., lines 471 and 500).

Reference

Lindgren, J. et al. Soft-tissue evidence for homeothermy and crypsis in a Jurassic ichthyosaur. *Nature* **564**, 359–365 (2018).

- It might be an idea to combine figure 4 with figure 1, which would be space-efficient, and also good for comparisons.

We thank the referee for their suggestion but have opted to retain Figure 4 to maintain the logical structure of our text.

- Figure 3J – I find this caption hard to follow and hope it can be elaborated.

Figure 3j depicts a FEG-SEM micrograph of a ground and polished section through a chondroderm. As described in the main body of text (see lines 153–167 in our initial submission), chondroderm calcified cartilage has a globular organisation, and concentric growth lines (interpreted by us as Liesegang banding patterns) are present in all sections investigated. We have now slightly rephrased the figure caption (see below), and added brackets and arrowheads to the image to make it clearer for the reader.

“j, Ground section of chondroderm calcified cartilage. Note globular organisation (brackets) and contour lines (arrowheads) interpreted as Liesegang banding patterns.”

In the methods section: In line 547-548 it says that experimental parameters were identical between ancient and modern samples. However, when looking at the Statistics and reproducibility paragraph, it does not seem to add up, because only a few methods were performed on the porpoise specimens. I hope this can be clarified, also in relation to point 1 above.

The experimental parameters *were* identical, what differs is the range of analyses performed on the two sets of samples. Regardless, we realise that this statement can be confusing, and hence decided to remove it from the revised version of our manuscript.

Data availability: There is no mention of where the porpoise specimens are held.

As requested, we have included the repository for our porpoise samples in the revised manuscript (lines 734–736).

Thank you for a very interesting manuscript.

Regards, Lene Liebe Delsett

We are grateful to Referee #5 for their supportive and helpful comments.

We are extremely pleased that four of our expert referees (#1–#3 and #5; #2 had minimal comments) have recommended publication of our manuscript largely as is. We acknowledge these comments below, and also provide a detailed point-by-point response to the statements of Referee #4.

Please do not hesitate to contact me if you have any queries or require further clarification of our results.

Yours sincerely,

Johan Lindgren

Response to Referee #1

I thank the authors to have taken into account my previous comments and suggestions. I have no additional comments to make at this stage.

We thank Referee #1 for acknowledging our compliance with their comments, and for recommending that our paper be accepted for publication.

Responses to Referee #2

Having now read the revision of the manuscript, it is improved and the authors have satisfactorily addressed my concerns. I find the article intriguing and it has taken our understanding of the biology of ichthyosaurs in a new direction. That the morphology of the flippers could have advantages in noise reduction for acquiring prey is a novel idea although based on hydro acoustics of living animals. However, there are a couple of points that could use additional references to help justify the assertions.

We thank the referee for their encouragement and constructive comments.

286-287- it is asserted that large flippers could be used for thrust production. A better reference for this statement comes from observations of humpback whales (Segre *et al.*, 2017), which have high-aspect ratio, elongate flippers that use the motion of these appendages to swim at low speeds.

Segre, P. S., Seakamela, S. M., Meyer, M. A., Kindley, K. P. and Goldbogen, J. A. 2017. A hydrodynamically active flipper-stroke in humpback whales. *Current Biology* 27: R636-R637.

Even though Segre and colleagues provide a detailed account on hydrodynamically active flipper-strokes, their study (Segre *et al.*, 2017) deals exclusively with the extant humpback whale (*Megaptera novaeangliae*), not *Temnodontosaurus*. Conversely, Buchholtz's (2001) paper (our ref. 40) specifically discusses the flipper function in *Temnodontosaurus*. Therefore, we have opted to cite Buchholtz (2001) also in the (second) revised version of our main text. However, we have added Segre *et al.* (2017) to the list of references cited in our revised Supplementary Information, as this paper is well suited for our discussion on “acrobatic” lunging manoeuvres in *M. novaeangliae* (line 640).

Reference

Buchholtz, E. A. Swimming styles in Jurassic ichthyosaurs. *J. Vert. Paleontol.* **21**, 61–73

(2001).

In addition, an article by Riess (1986) considered that a species of ichthyosaur with large pectoral fins could use them for swimming despite having a highly derived lunate tail fin.

Riess, J. (1986). Fortbewegungsweise, Schwimmbiophysik und Phylogenie der Ichthyosaurier. (Locomotion, biophysics of swimming and phylogeny of the ichthyosaurs) *Palaeontographica A*, 192, 93-155.

We thank Referee #2 for recommending this publication [please note that it is already cited (ref. 28) in the previous version of our main text (in a paragraph discussing the presence of fin ray-like elements in ichthyosaurs)]. While it is correct that Riess (1986) considered some ichthyosaurs to have been capable of flipper-based propulsion, it is not clear from either his text or images whether these also include *Temnodontosaurus* [for instance, the depicted “*Leptopterygius*” skeleton (see Riess, 1986, fig. 43) is in fact the neotype (NHMUK OR2003, traditionally referred to as BMNH 2003) of *Temnodontosaurus platyodon*; see, e.g., McGowan, 1974; McGowan & Motani, 2003]. Given this uncertainty, we have decided not to cite Riess (1986) when discussing inferred means of thrust production in *Temnodontosaurus*.

References

- McGowan, C. A revision of the longipinnate ichthyosaurs of the Lower Jurassic of England, with descriptions of two new species (Reptilia: Ichthyosauria). *Life Sci. Contr., R. Ont. Mus.* **97**, 1–37 (1974).
- McGowan, C. & Motani, R. Ichthyopterygia. In *Handbook of Paleoherpetology* (ed. Sues, H.-D.) Vol. 8, 175 pp. (Verlag Dr. Friedrich Pfeil, Munich, 2003).

313-317- In this sentence, only acoustics are considered for a large-bodied ichthyosaur to prey on highly maneuverable prey. While reducing an acoustic signature may be important, as it is one of the main points of the paper, the large size and flexible body of the predator may come into play to capture agile prey. Maresh et al. showed that dolphins could use their body flexibility to out-manuever and cut off the escape for smaller prey.

Maresh, J. L., Fish, F. E., Nowacek, D. P., Nowacek, S. M. and Wells, R. S. 2004. High performance turning capabilities during foraging by bottlenose dolphins. *Marine Mammal Science* 20(3): 498-509.

We thank the referee for pointing out a portion of the text that could be further elaborated, and accordingly have added a paragraph dealing with possible strategies to intercept and capture small-sized, elusive prey. Because of space limitations, this new information has been placed in our revised Supplementary Information (lines 652–657).

Please consider these references in your arguments. Frank E. Fish

We again thank Referee #2 for their input and useful advice.

Response to Referee #3

The authors have responded to my and other reviewers' comments in a very thorough and logical fashion. Indeed, after reading 35 pages of comments and responses, one has a heightened appreciation of the task of publishing a paper that caters to and depends on

multiple disciplines.

In my opinion, it is good to go.

We thank Referee #3 for endorsing the quality of our comprehensive responses, and for recommending that our manuscript is acceptable for publication.

Responses to Referee #4

I appreciate the authors' efforts to address my previous comments and queries. I have carefully reviewed the rebuttal, revised manuscript (from the Functional Implication to the computational fluid dynamics), and the revised supplementary notes (specifically, Part H: Notes on Analyses and Experiments). Unfortunately, I find that some explanations in these documents raise more questions than they answer.

We appreciate that the referee acknowledges our efforts to address their previous comments. Below, we provide detailed responses and clarifications also to their new feedback.

This paper must convincingly demonstrate that the surface texture (streamwise finlet ornamentations) and serrated trailing edge found in the wing-like flippers of a giant ichthyosaur can effectively suppress noise radiation. The authors' choice of a simple symmetrical aerofoil (NACA0018) to mimic the complex flippers of the ichthyosaur, whose exact shape, features, and taxonomy remain unproven, is an oversimplification, albeit an acceptable approach. Consequently, my examination focuses on the feasibility of these finlets and serrated trailing edge, based on the evidence provided, to effectively reduce trailing edge self-noise from a symmetrical aerofoil under low turbulence inflow conditions.

As stated in the initial submission of our manuscript (but subsequently removed following a recommendation by Referee #3): “approximations and simplifying assumptions are necessary when modelling multiscale fluid flows and their interactions with complex biological structures”. This applies to *all* artificially made geometries, numerical simulations as well as physical models. Moreover, the purported choice of a NACA 0018 aerofoil is based on a slight misunderstanding. As described in our Supplementary Information (lines 365–375), we used the profile of a Minke whale flipper to complement measurements obtained directly from SSN8DOR11 (the resulting similarity between our digital replica and a NACA 0018 aerofoil is thus a mere coincidence). Notably, cetacean flippers are both symmetrical and uncambered (e.g., Fish & Battle, 1995; Fish, 2004, 2020; Cooper *et al.*, 2008; Weber *et al.*, 2014; Fish & Lauder, 2017), and so were likely also ichthyosaur forefins (McGowan, 1991). Moreover, following our detailed responses to Referees #1 and #5 (two palaeontologists), there is no longer any uncertainty regarding the taxonomic affinity of SSN8DOR11, as both reviewers were satisfied with our explained rationale for assigning the fossil to *Temnodontosaurus trigonodon*.

However, all this being said, our computations admittedly contain a number of inherent approximations and limitations. For instance, because the animal to which SSN8DOR11 belonged is long extinct, both the exact cross-sectional shape and degree of flexibility of the flipper are unknown, and so is the cruising speed of the ichthyosaur. In addition, the height/depth of the surface ornamentations cannot be ascertained from the flattened fossil, and it is likewise not known if the trailing edge was entirely passive during swimming or could somehow be manipulated. And to further complicate matters, any evolutionary adaptation, big or small, that provides a selective advantage (and thereby positively affects an organism's

biological fitness) is of relevance for an animal (e.g., Futuyma, 1998)*. Given these considerations, we strongly oppose the opinion advocated by Referee #4 that we “must convincingly demonstrate” that the ornamentations and serrated trailing edge identified in SSN8DOR11 “can effectively suppress noise radiation”. Rather, we would argue that *any* noise dampening effect shown by our (by necessity) rather crude virtual fin section provides compelling circumstantial evidence for the presence of such adaptations in *Temnodontosaurus*. Also, and as emphasised in our previous response, the resulting calculated numbers should *not* be taken at face value, but instead are meant to serve as support for the general principle of noise suppression by trailing edge serrations and surface treatments (e.g., Arce León *et al.*, 2016; Clark *et al.*, 2017; Wolfe, 2017; Muhammad & Chong, 2022; Fiscaletti *et al.*, 2022; Smith & Klettner, 2022; Hu *et al.*, 2022; Ananthan & Akkermans, 2023), and that this concept can be applied also to a geometry that is partially based on SSN8DOR11.

*To reiterate our previous example: A wing with a backward swept leading edge and straight trailing edge provides a mere 4.5% increase in induced efficiency compared to a wing with an elliptical planform (van Dam, 1987). This is a low number but still enough to be beneficial, and consequently semi-crescent-shaped flippers and flukes have evolved independently in, e.g., whales and secondarily aquatic reptiles (e.g., van Dam, 1987; Lindgren *et al.*, 2013; Pavlov *et al.*, 2021).

References

- Ananthan, V. B. & Akkermans, R. A. D. Trailing edge noise reduction using bio-inspired finlets. *J. Sound Vib.* **549**, 117553 (2023).
- Arce León, C., Ragni, D., Pröbsting, S., Scarano, F. & Madsen, J. Flow topology and acoustic emissions of trailing edge serrations at incidence. *Exp. Fluids* **57**, 91 (2016).
- Clark, I. A. et al. Bioinspired trailing-edge noise control. *AIAA J.* **55**, 740–754 (2017).
- Cooper, L. N. et al. Hydrodynamic performance of the minke whale (*Balaenoptera acutorostrata*) flipper. *J. Exp. Biol.* **211**, 1859–1867 (2008).
- Fiscaletti, D., Luesutthiviboon, S., Avallone, F. & Casalino, D. Streamwise fences for the reduction of trailing-edge noise in a NACA633018 airfoil. *AIAA SCITECH 2022 Forum*, 1925 (2022).
- Fish, F. E. Structure and mechanics of nonpiscine control surfaces. *IEEE J. Oceanic Engin.* **29**, 605–621 (2004).
- Fish, F. E. Advantages of aquatic animals as models for bio-inspired drones over present AUV technology. *Bioinspir. Biomim.* **15**, 025001 (2020).
- Fish, F. E. & Battle, J. M. Hydrodynamic design of the humpback whale flipper. *J. Morph.* **225**, 51–60 (1995).
- Fish, F. E. & Lauder, G. V. Control surfaces of aquatic vertebrates: active and passive design and function. *J. Exp. Biol.* **220**, 4351–4362 (2017).
- Futuyma, D. J. *Evolutionary Biology* (Sinauer Associates, Inc., Sunderland, 1998).
- Hu, Y.-S. et al. Effects of trailing-edge serration shape on airfoil noise reduction with zero incidence angle. *Phys. Fluids* **34**, 105108 (2022).
- Lindgren, J., Kaddumi, H. F. & Polcyn, M. J. Soft tissue preservation in a fossil marine lizard with a bilobed tail fin. *Nat. Commun.* **4**, 2423 (2013).
- McGowan, C. *Dinosaurs, Spitfires, and Sea Dragons* (Harvard Univ. Press, Cambridge, 1991).
- Muhammad, C. & Chong, T. P. Mitigation of turbulent noise sources by riblets. *J. Sound Vib.* **541**, 117302 (2022).
- Pavlov, V. et al. Form, function, and divergence of a generic fin shape in small cetaceans. *PLoS ONE* **16**, e0255464.
- Smith, T. A. & Klettner, C. A. Airfoil trailing-edge noise and drag reduction at a moderate

- Reynolds number using wavy geometries. *Phys. Fluids* **34**, 117107 (2022).
- van Dam, C. P. Efficiency characteristics of crescent-shaped wings and caudal fins. *Nature* **325**, 435–437 (1987).
- Weber, P. W., Howle, L. E., Murray, M. M., Reidenberg, J. S. & Fish, F. E. Hydrodynamic performance of the flippers of large-bodied cetaceans in relation to locomotor ecology. *Mar. Mamm. Sci.* **30**, 413–432 (2014).
- Wolfe, T. M. Review of fluid dynamic and acoustic performance of biologically inspired passive flow control trailing edge devices for design applications. *55th AIAA Aerospace Sciences Meeting, AIAA SciTech Forum*, AIAA 2017-0542 (2017).

This implies that the turbulence-induced leading edge impingement noise is negligible and that no bypass transition mechanism is present on the aerofoil surface.

Our computations are meant to mimic an ichthyosaur swimming under relatively calm conditions; that is, out at sea and at a certain depth of water, where surface ornamentations and trailing edge serrations can be expected to have an impact on the hydroacoustic signature. Conversely, amplified background noise in a turbulent environment likely renders these passive flow control devices less relevant (at least from a noise abatement point of view). Therefore, we opted not to introduce any turbulent fluctuations at the inlet of our set-up. Moreover, by including turbulence, additional assumptions would have been necessary (pertaining, e.g., to turbulence scales), thereby increasing the number of unknown parameters.

However, the clarification of a Reynolds number of only 0.3 million raises concerns that the authors may have generated an aerofoil noise mechanism not primarily susceptible to serration and finlets. The mechanism in question is aerofoil instability tonal noise, one of the five mechanisms proposed by Brooks et al. (1989, NASA technical report). This type of noise relies on boundary layer instabilities (such as Tollmien-Schlichting instability waves) that propagate to the trailing edge and are amplified by a local separation bubble before scattering into tonal noise in the far field (refer to the Bristol papers by Nash et al. and McAlpine et al., as well as Kingan and Pearse, JSV, 2009). Thus, the radiated tonal noise will be determined by the T-S wavenumber amplification growth rate, leading to similar dominant frequencies.

The Reynolds number used in our computations is primarily a consequence of the inferred cruising speed of 1.5 m/s (by necessity, we had to limit the number of parameters in our simulations). The recommended literature focuses on identifying mechanisms that cause dominant noise frequencies by feedback mechanisms and/or amplification of flow instabilities, in order to avoid such unwanted disturbance in practical applications. However, to have tonal noise radiating from a swimming ichthyosaur is hardly likely, as this would reveal its presence to prey and adversaries alike. Also, if such noise emerges, one would expect the animal to either slow down or alter the angle of attack (α) of its flippers. Therefore, we instead focused on the broadband spectrum, and whether the observed surface features and trailing edge serrations have any impact on the emitted broadband noise.

To suppress this type of boundary layer instability tonal noise using serration, Chong and Joseph (JSV, 2013) attempted to utilise serrations to disrupt the separation bubble, which serves as an effective amplifier for T-S waves. However, the short amplitude length employed by the authors may not effectively suppress the separation bubble, potentially explaining the low overall noise reduction observed in their noise spectra (Fig. 9h).

Our aim was not to suppress certain noise generating mechanisms. Even though there are similarities between our set-up and the one employed by Chong & Joseph (2013) (e.g., same order of magnitude of the chord, comparable Reynolds numbers), there are also significant differences (use of different Mach numbers, adoption of a NACA 0012 aerofoil, larger and sharper serrations etc.) that make direct comparison difficult.

We are aware that the simulated effect of the inferred passive flow control devices is small; however, as explained above, the calculated numbers should not be taken at face value. Also, from an evolutionary perspective, even a minor reduction in self-induced noise could be beneficial for a hunting pelagic animal.

The authors' comparison of their predicted tonal noise with experimental results from Nakano et al. as a means of validating their computational algorithm—where significantly different tonal noise frequencies were observed—may be misguided.

As stated in our previous response, the comparison with Nakano *et al.* (2007) was done to test if our numerical approach can capture tonal noise in a set-up that is similar (but not identical) to the one we use in our computations. Admittedly, the tonal frequency produced by our simulation does not exactly match that of Nakano *et al.* (2007); however, it is important to point out that predicting the precise frequency is not a straightforward task. Rather, it can be influenced by a multitude of factors, including the size of the shed vortices, location of the separation (which in turn is sensitive even to small perturbations), blockage effects and so on. Notably, our intention was not to replicate the results of Nakano *et al.* (2007) in detail, but to determine whether or not our method is capable of predicting tonal noise in a case where such noise has been obtained experimentally. With further testing and fine-tuning of the numerical set-up, we are confident that a better agreement with Nakano *et al.* (2007) can be achieved; however, this is beyond the scope of our rebuttal.

Reference

Nakano, T., Fujisawa, N., Oguma, Y., Takagi, Y. & Lee, S. Experimental study on flow and noise characteristics of NACA0018 airfoil. *J. Wind Eng. Ind. Aerodyn.* **95**, 511–531 (2007).

It is also important to note that instability tonal noise is less prevalent under turbulent inflow conditions or in a three-dimensional flipper, as the separation bubble can be more easily suppressed. The real test for the serration and finlet lies in turbulent broadband noise, which I believe should be the primary focus of this paper. While I understand there is a broader palaeontological context in the manuscript, the low Reynolds number analysis, lacking external forcing terms (e.g., high inflow turbulence or boundary layer tripping), do not adequately address the critical research questions posed.

We respectfully disagree as our focus is on noise in the broadband spectrum (see discussion above). Moreover, by adopting a quasi-two-dimensional aero/hydrofoil, we can isolate the parameters of interest (conversely, additional details would lead to further assumptions, in turn decreasing the reliability of our results), and even if a complete flipper geometry had been employed, there would still be major limitations related, e.g., to the incoming velocity profile, presence/absence of turbulent fluctuations, and potential flow disturbances caused by the adjacent body. Moreover, large eddy simulations (LES) of a complete forefin would be exceedingly costly without contributing much to the study (for instance, the required computer processing hours would be several orders of magnitude larger using the same mesh resolution).

Also, while we appreciate that the referee shares their view on what they think should be the focus of our paper, we reiterate (and build on) our previous response to Referee #3 that our manuscript is intended as a biological sciences paper, and, as such, has three principal objectives:

1. To document the first-ever soft tissues of *Temnodontosaurus*;
2. To describe novel integumentary structures (chondroderms); and
3. To make functional inferences based on the observed eidonomic and anatomical features.

Notably, our computational fluid dynamics simulations are merely meant to serve as a complement to the extensive published literature on the function of surface treatments and trailing edge serrations on aero/hydrofoils (see references listed in our response above). Accordingly, although the scrutiny of our computations allowed us to identify and rectify a minor rounding error in the software employed to produce our microphone data (see below), we find the intense focus on our simulations to be blown a bit out of proportion.

In addition, I have some comments and queries that I hope the authors can address:

1. In the main manuscript, lines 239–240, “streamwise finlet ‘fences’ with a spacing of 2mm...”: This sentence should be revised, as dimensional terms are not typically used to describe an optimal geometrical parameter. The optimal size should be presented non-dimensionally, scaled with a representative boundary layer length scale.

Although we agree in principle with the referee, non-dimensional geometrical parameters would mean little to the intended primary target audience of biologists and palaeontologists. Also, it was never our intention to identify an optimal spacing; rather, we replicate features seen in the fossil. Hence, we have retained the sentence as originally written.

2. In the main manuscript, lines 248–250, the statement that trailing edge serration can reduce drag and alleviate stall is quite extraordinary. I suggest deleting this sentence along with the quoted references.

It is correct that drag usually is either insignificantly changed or even somewhat increased when trailing edge serrations are added to an aero/hydrofoil. For high α , however, there are a few published cases where drag is slightly reduced (e.g., Liu *et al.*, 2016; Llorente & Ragni, 2020). Nevertheless, we have opted to follow the recommendation by Referee #4, and removed the statement that trailing edge serrations can reduce drag and alleviate stall.

References

- Liu, X., Kamliya Jawahar, H. & Azarpeyvand, M. Wake development of airfoils with serrated trailing edges. *22nd AIAA/CEAS Aeroacoustics Conference*, 2817 (2016).
- Llorente, E. & Ragni, D. Trailing-edge serrations effect on the performance of a wind turbine. *Renewable Energy* **147**, 437–446 (2020).

3. In the main manuscript, line 253, the word “experimentally” should not be used to describe the work, as it could be misleading. Since OpenFOAM is used as the research tool, I recommend replacing "experimentally" with "numerically." This suggestion applies throughout the paper.

“Experimentally” has been replaced with “numerically” throughout the revised main text to comply.

4. In the main manuscript, line 272, the Strouhal number is used but without clarification. Could the authors specify the length scale used in the Strouhal number?

To be consistent with the normalisation used for the Reynolds number, we employed the free-stream velocity magnitude (1.5 m/s) and chord length (0.2 m) as the velocity and length scale, respectively. This information has been added to our revised Supplementary Information (lines 467, 468).

5. In the supplementary note, under the "Noise attenuation by passive flow control devices" section, "... Passive flow control devices introduce small disturbances into the flow, which enhance the breakup of larger vortices, in turn causing noise attenuation...": This statement is overly broad and not entirely accurate. For example, serrated trailing edges are not used to break up large-scale vortices.

We respectfully disagree as trailing edge serrations *do* introduce perturbations to the flow (see, e.g., Callender *et al.*, 2004, 2008). Even though these disturbances can be small in some cases, they nonetheless contribute to the destabilisation of large-scale vortices in the wake. Also, this section of our text is meant as a brief introduction to the subject of noise attenuation by passive flow control devices for non-specialists (given the interdisciplinary readership of *Nature*), which naturally necessitates a limited amount of in-depth details.

References

- Callender, B., Gutmark, E. & Martens, S. A PIV flow field investigation of chevron nozzle mechanisms. *42nd AIAA Aerospace Sci. Meeting Exhibit*, 191 (2004).
Callender, B., Gutmark, E. & Martens, S. A near-field investigation of chevron nozzle mechanisms. *AIAA J.* **46**, 36–45 (2008).

6. In the supplementary note, under the "Constructing the flipper geometry" section, replace "experimentally" with "numerically." The same change applies throughout the rest of the notes.

“Experimentally” has been replaced with “numerically” throughout the revised Supplementary Information to comply.

7. In the supplementary note, under the "Constructing the flipper geometry" section, “The height of both surface treatments increased linearly from 0 to 0.8 mm between 5 and 10% of the chord, and decreased again to 0 mm between 90 and 95% of the chord.” What is the context of these heights when scaled with the wall unit? It's important to understand whether the treatments are within the inner layer or protrude into the outer layer.

The figure below shows the distribution of ridge heights in terms of wall units for a serrated flipper geometry at $5^\circ \alpha$. The average value in the region marked by a white box is approximately 50; hence, the ridges are in the overlap between the linear and logarithmic regimes. It is also worth noting that the development of the boundary layer along the flipper surface not necessarily has to follow a theory established for boundary layers over flat plates. For example, if a separation bubble develops near the trailing edge, there will be a region on the surface where the boundary layer theory does not apply.

Ridge heights in terms of wall units for a serrated flipper geometry at $5^\circ \alpha$.

8. The title of the section “Experimental set-up” should be changed to “Numerical set-up.”

The text has been modified to comply.

9. In the supplementary note, under the "Impact of trailing edge serrations" section, “The strongest fluctuations occur on the suction side, in the region where the transition to turbulence takes place, which suggests that the separation point is unsteady.” This sentence is difficult to understand. Did transition actually occur?

According to Eggert & Rumsey (2017), the flow around a NACA 0018 aerofoil at $Re = 400,000$ is transitional, which holds true also for the greater part of the investigated flipper surface. To firmly identify the precise location where the flow becomes fully turbulent, we would need monitoring points along the entire surface. However, this was not the purpose of our computations, and therefore we only have a limited number of monitoring points for the instantaneous pressure and velocity. The figure below shows the monitoring point located closest to the flipper (red dot), together with a spectrum of the turbulent kinetic energy at that point and the theoretical $-5/3$ decay line. Note that the spectrum partially follows the $-5/3$ line, to suggest the existence of turbulence at this location.

***a**, Instantaneous velocity fluctuations for a case with 0.8 mm ridges and trailing edge serrations at $5^\circ \alpha$. The red dot denotes the monitoring point located closest to the flipper section. **b**, Spectrum of the turbulent kinetic energy at the point indicated in **a** together with the theoretical $-5/3$ decay line.*

Reference

Eggert, C. A. & Rumsey, C. L. CFD study of NACA 0018 airfoil with flow control. *NASA/TM-2017-219602* (2017).

10. “For the $10^\circ \alpha$ case, the separation occurs near the leading edge, but the maximum fluctuation levels are lower than in the previous ($5^\circ \alpha$) case.” Does the suction side boundary layer undergo a complete stall at 10° or 15° ? It’s unclear if the trailing edge serration would work in this regime.

Our revised Figure 5b depicts drag and lift coefficients for both unserrated and serrated flipper geometries at $0, 5, 10, 15,$ and $20^\circ \alpha$. Based on our computations, complete stall occurs somewhere between 15 and $20^\circ \alpha$. Notably, there is a minor reduction in drag at $20^\circ \alpha$ when serrations are added to the trailing edge of our digital model.

To validate our computations, we compared our results against values obtained by Eggert & Rumsey (2017) for a NACA 0018 aerofoil, as well as force coefficients listed on the Airfoiltools homepage (see figures below). Collectively, these data demonstrate that our flow computations are accurate.

[REDACTION]

a, Lift coefficient reported by Eggert & Rumsey (2017, fig. 4) for a NACA 0018 aerofoil at a Reynolds number (Re) that is comparable to the one used in our computations. *b*, Lift and *c*, drag coefficients obtained from the Airfoiltools data base (<http://airfoiltools.com/airfoil/details?airfoil=naca0018-il>) for a NACA 0018 aerofoil at $Re = 200,000$ and $500,000$.

Reference

Eggert, C. A. & Rumsey, C. L. CFD study of NACA 0018 airfoil with flow control. *NASA/TM-2017-219602* (2017).

11. “The largest differences in the radiated noise are observed at this α , which can be explained by the trailing edge serrations extending further into the streamlines than at $0^\circ \alpha$.” This sentence is unclear and difficult to interpret.

When conducting the new numerical analyses, we discovered that the write precision setting in OpenFOAM rounds off the time stamps of the microphone data, which introduced a small but systematic error in the resulting spectra. These have now been rectified and the relevant panels of Figure 5 and Extended Data Figures 9 and 10 revised accordingly. Although this action caused some minor–moderate changes to the plotted data in a few line graphs, it is important to note that *the trends remain the same as before*; that is, they demonstrate noise abatement by trailing edge serrations and surface treatments on a virtual geometry that is partially based on SSN8DOR11. The supplementary text has been modified to accommodate these adjustments.

12. In the supplementary note, under the "Noise abatement by surface treatments" section, "It is further noteworthy that the lighter blue colour in the wake (Extended Data Fig. 10d, e) indicates that the velocity fluctuations are smaller in this region when surface treatments are added to our model." The reduction in velocity fluctuations in the wake region due to surface treatments does not necessarily indicate noise reduction benefits.

We agree with the referee that it is generally not possible to draw any specific conclusions regarding potential noise reduction based solely on the velocity fluctuation distribution. Plotting the noise sources would likewise not result in a direct indication of the noise levels because the radiated noise depends on the magnitude, as well as the temporal and spatial variation of these sources (in addition to interactions with surrounding objects). With this being said, however, large pressure fluctuations often lead to stronger emitted noise. Because we solve an incompressible set of Navier–Stokes equations, the pressure fluctuations are directly linked to the velocity fluctuations, and therefore large velocity fluctuations indicate a high probability for stronger radiated noise.

13. "Similar conclusions can be drawn from the results obtained at $5^\circ \alpha$ (Extended Data Fig. 10f–h); that is, surface treatments lead to earlier development of vortices and a reduction of the maximum velocity fluctuation magnitude." What is the mechanism by which the finlet reduces the maximum velocity fluctuations?

We hypothesise that the surface treatments act as trip wires; that is, they create small eddies that speed up the breakdown of larger vortices, thereby decreasing the maximum velocity fluctuations.

14. "This decrease is attributed to the small vortices generated by the ornamentations that trigger flow separation." Does the ornamentation trigger or suppress flow separation?

We thank Referee #4 for pointing out a portion of the text that could be further elaborated. What we meant is that small eddies trigger the breakdown of larger vortices. This has been clarified in our revised Supplementary Information (lines 510–512).

15. "At $0^\circ \alpha$ (Extended Data Fig. 10i), there is a significant (~ 4 – 5 dB) noise reduction in the low-frequency (< 50 Hz, $St = 6.7$) region for serrated flipper sections equipped with either ridges or troughs." Do the authors have noise results for surface treatments alone? It's possible that much of the noise reduction is due to the surface treatment, rather than the serration, as Figure 9 suggests the serration does not significantly reduce noise. Are the authors suggesting that the low-frequency noise reduction is caused by the breakdown of large-scale turbulence in the wake? The noise spectra appear quite tonal.

It may well be that contributions from the surface ornamentations are disproportionate in our computations (as explained above, they are not without their limitations and imperfections); however, given that trailing edge serrations demonstrably are present in SSN8DOR11, and that the purpose of our numerical investigation is to mimic features seen in the fossil, exploring purely hypothetical cases (such as to one proposed by the referee) is beyond the scope of the present study.

Presumably, the recorded noise reduction is due to a faster breakdown of large-scale vortices caused by upstream fluctuations introduced by the ornamentations. Consequently, this process does not only occur in the wake, but also in the vicinity of the flipper surface.

16. “This is presumably because small-scale vortices are present in the separation region surrounding the smooth flipper as well.” The connection between higher frequency noise and the separation bubble requires more evidence.

Again, we thank the referee for indicating a point of clarification. Our intention was only to link high frequency noise with small vortices, not necessarily the separation bubble. This has been changed in our revised Supplementary Information (lines 523–525).

Response to Referee #5

Thank you for the opportunity to review this paper a second round. I find that the authors have responded very thoroughly to my comments, and also to the comments of the other reviewers. All in all, the second round improved a paper that was already good and I believe it can be published without any further changes.

We are appreciative of Referee #5’s constructive assessment, and their recognition that all of the other referees’ comments have been appropriately addressed.

Response to Referee #2

At this point it appears that the authors have made the appropriate changes for the manuscript. My only comment refers to lines 284-285, where it is stated that the large flippers could possibly produce thrust as referenced by Bucholtz (2001). The only place that any mention of the thrust production aside from the caudal fin occurs in Bucholtz is in regard to the “large hind limbs” for various ichthyosaurs (left column, p. 71). If the authors are making a case for the fore (pectoral) flippers to produce thrust and it is indistinct which flippers the authors are referring to, a better reference is Riess (1986), who speculated on this for ichthyosaurs and more particularly by Segre et al. (2017). My justification for including Segre et al. is that there is actual video evidence of a large aquatic predator (humpback whale) using its elongate pectoral flippers for propulsion. Bucholtz and Riess merely speculate without any visual evidence, other than morphological evidence, that the pectoral fins can be used for propulsion. Segre et al. provides real evidence that of the possibility and thus strengthens and supports the author’s assertion.

We thank Referee #2 for acknowledging our compliance with their comments. Please note that we now cite Riess (1986), Buchholtz (2001) and Segre *et al.* (2017) when discussing possible means of flipper propulsion in *Temnodontosaurus* (main text, line 257).

Responses to Referee #4

I would like to begin by thanking the authors for their efforts in addressing my comments during this second round of review. I appreciate that the primary focus of the manuscript relates to the biological science, and I acknowledge the interdisciplinary nature of the work.

We appreciate that the referee acknowledges our efforts to address their previous comments, and for recognising the interdisciplinary nature of our work.

However, I must express my concern regarding the authors' remark that "...we find the intense focus on our simulations to be blown a bit out of proportion." I would reject this characterization. The central premise of the paper is to explore the stealthy predatory capabilities of the ichthyosaur and to link this behaviour to the anatomical and morphological features revealed in fossilized flippers, specifically the serrated trailing edge, chordwise parallel skin ornamentations, and the protracted distal tip functioning as flexible winglets. The numerical simulations serve as the primary tool to validate the aerodynamic and acoustic implications of these features. Therefore, the fidelity of the computational study is not a peripheral detail but an essential component underpinning the novelty and impact of the manuscript, especially in a high impact, multidisciplinary journal such as Nature.

While the authors have stated their intent to target readers primarily from the biology and paleontology communities, I believe this is a narrow view. The potential of this study to inspire cross disciplinary interest, such as among bio-inspired engineers or fluid dynamicists, is significant. For instance, a wind turbine designer could conceivably draw inspiration from these features and seek to apply them to turbine blade designs. In this context, reporting design parameters such as "2 mm fence spacing" in purely dimensional terms is of limited use. Given the large disparity in scale between ichthyosaur flippers and engineered systems, such values must be made dimensionless, for example with respect to the local boundary layer thickness, to facilitate meaningful comparisons. This was a minor request, easily achievable using the existing LES database, and would improve the paper’s rigor.

We can only concur with Referee #4's opinion regarding the far-reaching implications of our results, and their potential to be of interest not only to biologists and palaeontologists but also to a considerably wider group of scholars, including hydrodynamics and acoustics engineers. In fact, our manuscript originally ended with the statement "Our findings show that such features already existed in at least one lineage of ichthyosaurs 183 million years ago, and thus could provide inspiration to help limit the adverse biological effects from anthropogenic input to the marine soundscape." However, following a recommendation by Referee #2, the latter part of this sentence was subsequently removed. Furthermore, we did not intend to diminish the relevance of our numerical simulations; rather, "blown a bit out of proportion" was an ill-fated attempt to put the computations into perspective.

Because "2 mm fence spacing" primarily refers to the work by Fiscaletti *et al.* (2022) and not our study, we have opted to remove this sentence from the revised version of our main text.

Reference

Fiscaletti, D., Luesutthiviboon, S., Avallone, F. & Casalino, D. Streamwise fences for the reduction of trailing-edge noise in a NACA633018 airfoil. *AIAA Pap. 2022-1925* (2022).

I do not believe that issuing a point-by-point rebuttal to every response would be productive or fair, especially given the extended duration of the review process has already taken. Instead, I would like to suggest a single decisive action from the authors. They should either:

(1) Provide a clear and consistent rationale regarding the type of aeroacoustics noise they aim to suppress. The authors indicate in multiple responses that they focus on turbulent broadband noise:

"...we instead focused on the broadband spectrum, and whether... have any impact on the emitted broadband noise."

"...as our focus is on noise in the broadband spectrum..."

However, the simulation conditions used in the study (that is, Reynolds number around 300000, low inflow turbulence, and low angle of attack) do not support this as a dominant noise mechanism. Under such conditions, the primary noise source is more likely to be laminar instability tonal noise, a well-documented phenomenon in airfoil acoustics. For instance, Arcondoulis *et al.* (2010, *Acoustics Australia*, 38, 129–133) provide clear evidence of this, including relevant spectral envelopes (such as in Figure 6), which, although based on a NACA0012 profile, remain broadly applicable to NACA0018.

(For context, the laminar instability tonal noise can be eliminated very easily, such as by applying simple surface roughness to trigger bypass transition and suppress the Tollmien-Schlichting wave. It certainly does not require elaborate ancient animal features to do so).

or

(2) Remove the aeroacoustics analysis section entirely and reshape the manuscript to focus more exclusively on the biological and morphological features. This would align the paper more clearly with the intended audience and ensure that the conclusions remain robust within the relevant disciplinary scope.

We thank the referee for their suggestion, and, accordingly, have added a rationale to our revised Supplementary Information (lines 375–379). Moreover, we also welcome this opportunity to clarify some misconceptions of our work. The aim of the numerical simulations is not, and has never been, to investigate how to dampen a certain type of noise, nor has it been to optimise noise suppression. Instead, the purpose of our computations was to test the hypothesis that certain flipper features (that is, the trailing edge serrations and surface ornamentations) could have provided hydroacoustic benefits. Notably, we did not make any *a priori* assumption regarding the type of noise emitted by the virtual fin geometry, and obtained a noise distribution in the broadband spectrum in our unbiased simulations. It is therefore not entirely meaningful to discuss suppression of tonal noise when such single-frequency sound does not appear in our analyses. Referee #4 claims (in a rather sweeping manner) that just because a NACA 0012 generates tonal noise at the Reynolds number (Re) employed in our simulations, this phenomenon will occur also when a NACA 0018 is used. To support this assertion, they cite the work by Arcondoulis *et al.* (2010). Notably, though, these authors do not propose that their findings are applicable to thicker aerofoils, such as a NACA 0018. Rather, what they state after comparing their NACA 0012 geometry to thinner profiles is “Thus it can be deduced that the airfoil profile influences the nature of the acoustic feedback mechanism” (Arcondoulis *et al.*, 2010, p. 132). Importantly, this indicates that the tonal envelope for a NACA 0018 could be dissimilar to that of a NACA 0012. Indeed, when Huanhuan *et al.* (2019) examined four NACA aerofoils (0008, 0012, 0015, and 0018) at 0° angle of attack (α) and a Reynolds number of 200,000, they noticed that in the 0008, 0012 and 0015 cases, a distinct frequency—close to the wake-shedding frequency—could be seen in the sound pressure level spectra. However, for NACA 0018, no such discrete frequency noise could be detected. It is therefore likely that the conditions produced at $0^\circ \alpha$ and $Re = 200,000$ are outside of the tonal envelope for this geometry. Also, in the work by Zhu *et al.* (2022), a NACA 0018 profile exhibited ‘typical’ broadband noise behaviour at $0^\circ \alpha$ and a Reynolds number of 263,000. An experimental study by Nakano *et al.* (2007) further revealed that it is possible to have tonal noise radiating from a NACA 0018 profile, but at a Reynolds number of 160,000. Tonal noise can admittedly be emitted also from a NACA 0018 aerofoil at Reynolds numbers close to 300,000, but only if end effects (that is, finite-span wing) are included in the analysis (Geyer & Moreau, 2021); notably, though, this is *not* the case in our study.

We would also like to emphasise that the hydrofoil geometry we constructed is based on the flipper of a living Minke whale, *Balaenoptera acutorostrata*, which is similar *but not identical* to a NACA 0018. Importantly, even such small geometrical differences could have a great impact on the flow characteristics, potentially resulting in an amplification of the mechanisms that suppress tonal noise. Finally, we did set up a simulation to test if our numerical approach is capable of capturing tonal noise (using the $0^\circ \alpha$, $Re = 160,000$ case in Nakano *et al.* (2007) discussed above), and it did indeed predict the appearance of such single-frequency noise (see our initial rebuttal letter).

References

- Arcondoulis, E. J. G., Doolan, C. J., Zander, A. C. & Brooks, L. A. A review of trailing edge noise generated by airfoils at low to moderate Reynolds number. *Acoustics Australia* **38**, 129–133 (2010).
- Geyer, T. F. & Moreau, D. J. A study of the effect of airfoil thickness on the tonal noise generation of finite, wall-mounted airfoils. *Aerospace Sci. Technol.* **115**, 106768 (2021)
- Huanhuan, F., Yong, L., Qi, W. & Sen, Z. Numerical study on tone noise of different thickness airfoils. *IOP Conf. Ser., Mater. Sci. Eng.* **538**, 012052 (2019).

Nakano, T., Fujisawa, N., Oguma, Y., Takagi, Y. & Lee, S. Experimental study on flow and noise characteristics of NACA0018 airfoil. *J. Wind Eng. Ind. Aerodyn.* **95**, 511–531 (2007).

Zhu, W. et al. Numerical study on flow and noise characteristics of an NACA0018 airfoil with a porous trailing edge. *Sustainability* **15**, 275 (2022).

In summary, I appreciate the interdisciplinary ambition of the manuscript and commend the authors' efforts to explore a novel biological hypothesis. However, given the current ambiguity surrounding the noise mechanisms and the pivotal role of the simulations in supporting the central claims, further clarification or restructuring is necessary.

We are grateful to Referee #4 for their constructive comments but conclude that we respectfully disagree with their assessment (see the discussion above), which is also at odds with the data produced in our analyses.

Temnodontosaurus trigonodon

SMNS 50000

[length ca. 120 cm just the bones preserved; soft tissue paddle even longer]

SSN8DOR11 - assumed to be *Temnodontosaurus trigonodon*

[length ca. 80-85 cm as preserved from radiale to distal soft-tissue tip (p.3: "The metre-long forelimb...")]

Specimens scaled to same length of scale bar (=10cm)

from Motani 1999

Temnodontosaurus burgundiae

SMNS 15950

Temnodontosaurus burgundiae

SMNS N2A

Temnodontosaurus sp.

SMNS 17980